



# A perturbed biogeochemistry model ensemble evaluated against in situ and satellite observations

Prima Anugerahanti[1], Shovonlal Roy[1,2], and Keith Haines[3]

[1]Department of Geography and Environmental Science, University of Reading, Whiteknights, Reading, RG6 6AB, UK
[2]School of Agriculture, Policy, and Development, University of Reading, Whiteknights, Reading, RG6 6AR, UK
[3]Department of Meteorology, University of Reading, Whiteknights campus Early Gate, Reading, RG6 6BB, UK

**Correspondence:** Prima Anugerahanti (p.anugerahanti@pgr.reading.ac.uk), Shovonlal Roy (shovonlal.roy@reading.ac.uk)

**Abstract.** The dynamics of biogeochemical models are determined by the mathematical structure used for the main biological processes. Earlier studies have shown that small changes in model formulation may lead to major changes in system dynamics, a property known as structural sensitivity. We assessed the impact of structural sensitivity in an intermediately complex biogeochemical model (MEDUSA) by modelling the chlorophyll and nitrogen concentrations at five different oceanographic stations

spanning three different regimes: oligotrophic, coastal, and the abyssal plain over a 10-year timescale. A 1-D MEDUSA ensemble was used with each ensemble member having a combination of tuned function parameterizations that describe the key biogeochemical processes, namely nutrient uptake, zooplankton grazing, and plankton mortalities. The impact is quantified using phytoplankton phenology (initiation, bloom time, peak height, duration, and termination of phytoplankton blooms) and other statistical measures. The spread of the ensemble as a measure of uncertainty is assessed against observations using

the Normalised RMSE Ratio (NRR). We found that even small perturbations in model structure can produce large ensemble spreads. The range of 10-year mean surface chlorophyll concentrations are between 0.14-3.69 mg m$^{-3}$ at coastal stations, 0.43-1.11 mg m$^{-3}$ on the abyssal plain, and 0.004-0.16 mg m$^{-3}$ at the oligotrophic stations. Changing mortality and grazing functions have the largest impact on chlorophyll concentrations. The in situ measurements of bloom timings, duration, and terminations lie mostly within the ensemble range and using the ensemble properties such as the mean and median, the errors

are mostly reduced compared to the default model output. The NRRs for monthly variability suggest the ensemble spread is generally narrow (NRR 1.21-1.39 for nitrogen and 1.19-1.39 for chlorophyll profiles, 1.07-1.40 for surface chlorophyll, and 1.01-1.40 for depth integrated chlorophyll). Among the five stations, the most reliable ensembles are obtained for the oligotrophic station ALOHA (for the surface and integrated chlorophyll 10-year time series and bloom peak height), coastal station L4 (for inter-annual mean), and abyssal plain station PAP (for bloom peak height). Overall our studies provided a novel way to

generate ensemble spread by perturbing the model structure/parameterizations, and reliable ensemble means and spreads may be generated.





## 1    Introduction

Major changes in ocean biogeochemistry have been driven by anthropogenic activities, leading to ocean acidification, eutroph-ication, and increased levels of dissolved inorganic carbon (Gehlen et al., 2015; Bopp et al., 2013; Doney, 2010). To understand how the ocean ecosystem responds to these changes, marine biogeochemical models have been used. The majority of these

models focus on the lower trophic food-webs and explicitly represent dissolved nutrients, phytoplankton and zooplankton (NPZ). These models are then coupled with physical general circulation models to address the impacts of climate change (Doney et al., 2012) and forecasting systems (Yool et al., 2013; Butenschon et al., 2016).

Marine biogeochemical model development began with simple NPZ models, and has become steadily more complex with increasing computing power and knowledge of ocean biogeochemistry (Anderson, 2005; Anderson et al., 2015). NPZ models

consist of three compartments: nutrients as the primary resource, phytoplankton as the primary producers, and zooplankton as herbivores or grazers. Such models have been used to investigate the range of possible ecosystem behaviours before coupling them to a physical model (Franks, 2002) and seeking to represent observations at particular sites (Fasham et al., 1990; Robinson et al., 1993). More advanced biogeochemical models represent several processes and feedbacks (Raick et al., 2006), covering much more of the lower-trophic food web (Anderson, 2005). Inclusion of cell size representations (Berelson, 2002; Quèrè

et al., 2005), different phytoplankton functional types, such as calcifiers and dimethyl sulphide producers (Quèrè et al., 2005), and the addition of micronutrients, such as iron to permit spatial variability in phytoplankton concentrations (Yool et al., 2011, 2013), are now part of many biogeochemical models. Moreover, in order to investigate the effect of global climate change and anthropogenic activities in the ocean, marine biogeochemical models are now being embedded into earth system models. For example, the Model of Ecosystem Dynamics, nutrient Utilisation, Sequestration, and Acidification (MEDUSA) (Yool et al.,

2011, 2013) is the chosen biogeochemical component for the UK Earth System Model, as it has high spatial correlation with patterns of pCO$_2$, DIC, and alkalinity (Cox and Kwiatkowski, 2013; Kwiatkowski et al., 2014).

Despite becoming more complex (Anderson, 2005), the overall interactions among nutrients, phytoplankton, and zooplank-ton are still at the heart of all marine biogeochemical models. These interactions are governed by four primary processes, represented in the simplest NPZ models: nutrient uptake, grazing by zooplankton, phytoplankton and zooplankton mortality.

These processes are functions of the state concentrations and can be parameterized by more than one functional form, similar in shape but using different mathematical functions and adjustable parameters. Therefore there are two types of uncertainties that affect biogeochemical models: parametric, associated with the choice of parameter values; and structural, which relates to the underlying model equations (Hemmings and Challenor, 2012). It has been demonstrated in conventional sensitivity analyses that only small perturbations are usually produced even with large variations in parameter values, but much larger changes

in system dynamics can result from changes in the structural process formulations (Wood and Thomas, 1999; Fussmann and Blasius, 2005; Levin and Lubchenco, 2008; Flora et al., 2011; Adamson and Morozov, 2013; Aldebert et al., 2016), a result known as structural sensitivity (Wood and Thomas, 1999; Flora et al., 2011; Adamson and Morozov, 2013).

Structural sensitivity may be less significant in models built on well-tested mechanisms such as those in the physical sci-ences. However, in biogeochemical models, it is rare that a solid mechanistic basis is present, therefore it is uncertain what





is the most appropriate specification of the process functional terms. This is even more problematic if the process itself is not well understood so that theoretical justification for the specific representation is weak (Adamson and Morozov, 2013). Often it is difficult to implement the functional relations that are observed in the laboratory into a large scale ecosystem with heterogeneous populations (Englund and Leonardsson, 2008). It is known from studies of simple predator-prey models that

applying similarly shaped equations often garners completely different stability and oscillatory model dynamics (Fussmann and Blasius, 2005; Roy and Chattopadhyay, 2007). Moreover, a specific functional form may not capture all details of the biological processes, for example, the Michaelis-Menten type function for grazing, commonly known as the 'Holling Type II', fails to correctly describe what happens to grazers' movements when satiation has been reached (Flynn and Mitra, 2016). These studies show that simple biological models are highly susceptible to structural sensitivity. Further, the discrepancies reported

from simple interaction models suggest that the dynamics of complex biogeochemical models need to be tested by altering their default functional forms (Anderson and Mitra, 2010; Anderson et al., 2010)

Some studies have investigated the effects of different process formulations on biogeochemical models, e.g. Yool et al. (2011) has demonstrated that in an intermediately complex model, linear density-dependent mortality produces the most significant differences when applied to diatoms, compared with sigmoidal, quadratic, or hyperbolic forms. The choice of zooplankton

grazing equations can also affect phytoplankton concentration dramatically in a model with five plankton types, PlankTOM5.2 (Quèrè et al., 2005). The Michaelis-Menten (Holling type II) grazing function produces 30% less total surface phytoplankton concentration compared to the sigmoidal (Holling type III) functions, in the North Atlantic and North Pacific (Anderson et al., 2010). However, not all processes give significantly different model outputs. Anderson et al. (2015) also shows that when two similarly shaped photosynthesis-irradiance curves, namely, Smith and the exponential function, were used in an NPZ-detritus

model, the concentration of chlorophyll during the spring bloom was only slightly higher ($0.2$ mg m$^{-3}$) for the exponential function (Anderson et al., 2015).

These studies suggest that when the model formulation is perturbed, each process can give rise to different effects. However, it is still unclear what will happened if formulations of all the core processes, i.e. nutrient uptake, grazing and mortality are perturbed together. Since the individual compartments of models interact with one another, any biological perturbation

is likely to affect the whole ecosystem dynamics. In climate modelling, perturbed physics ensembles have been developed to investigate multiple parameter uncertainty (Murphy et al., 2007; Tinker et al., 2016), and multiple parametrization uncertainties (Subramanian and Palmer, 2017). By adopting an approach similar to that in climate modelling, here we generate a perturbed biogeochemical ensemble where model equations are varied by embedding different functional forms to describe the core processes, similar to the multi-parameterization ensembles in physical models. We implement this framework in MEDUSA

model (Yool et al., 2011, 2013), which is a lower trophic level model with two phytoplankton functional types, distinguished as large diatoms and small non-diatoms, two zooplankton types represented by mesozooplankton and microzooplankton, and three nutrients: silicic acid, iron, and inorganic nitrogen. Nitrogen is the primary currency of the model, similar to NPZ models, but MEDUSA allows phytoplankton to have different C:N ratios and Si:N ratios for diatoms. Diatoms utilise the silicic acid and can only be grazed by mesozooplankton. MEDUSA also includes an iron submodel developed by (Parekh et al., 2005) based





on (Dutkiewicz et al., 2005), in which iron is separated into "free" iron and iron bound to organic ligands. Iron is removed by scavenging and added to the ocean by aeolian deposition.

We assess of the uncertainty arising from the MEDUSA model's equations from ensemble outputs generated using all possible functional combinations within the NPZ compartments. For simplicity we use a 1-D version of MEDUSA-1.0 model (Yool et al., 2011; Hemmings et al., 2015), and produce results for five oceanographic stations covering abyssal plain, oligotrophic, and coastal regimes. Apart from the model outputs on concentration of nutrients and chlorophyll, we also examine the emergent properties of phytoplankton using phytoplankton phenology metrics. The performance of the ensemble mean, median, and the default MEDUSA run are compared with monthly and inter-annual values from in situ observations at those stations. We assessed the spread of the ensemble using the Normalised RMSE Ratio (NRR) which assesses the likelihood of the observations fitting the ensemble range. Section 2 describes the equations used and how the ensemble is run. The assessment of the uncertainty in terms of chlorophyll concentrations, phytoplankton phenology, and comparisons with the observations are described in section 3, and are further discussed in section 4.

## 2 Method

In models, the key processes can be represented by a variety of functional forms, which are comparable in shape but different in their mathematical representation. To explore this structural uncertainty we first attempt to make the functional forms representing key processes more equivalent to each other. For each process functional form we optimise the shape-defining parameters to make the functions equivalent to each other. For example, for Holling type II and Holling type III, we fix the maximum rates of each process, and implement a non-linear least squares method to optimize the half saturation coefficients so that the overall shapes are as similar as possible. A similar approach is used for nutrient uptake (4 functional forms), phytoplankton mortality (4 functional forms), and zooplankton mortality (4 functional forms). These are described in the subsections below.

### 2.1 Nutrient uptake

Alongside light, nutrient concentration also limits the growth of phytoplankton. In MEDUSA the standard hyperbolic monod, hereafter $U_1$, function is the default. The growth of cells monotonically increases with ambient nutrient concentration, and halts when nutrients become scarce. If nutrient concentrations are high, the rate of uptake saturates. Other mathematical functions show similar properties including (i) Sigmoidal (Fennel and Neumann, 2014), similar to Holling type III, $U_2$, (ii) the exponential (Ivlev, 1961), $U_3$, and (iv) trigonometric functions (Jassby and Platt, 1976), $U_4$. All these functions include a shape defining parameter, $k$, which for monod and sigmoidal can be interpreted as a half saturation constant, with the maximum uptake rate, $V_{p^T}$, which is a function of temperature (Eppley, 1972): $V_{p^T} = V_p 1.066^T$, where $V_p$ is the maximum growth rate when temperature is at 0°Celsius. MEDUSA has silicon and iron nutrients, and two phytoplankton types: diatoms and non-diatoms. The uptake function of different phytoplankton types and nutrients use similar functions but different parameter values for $k$, as summarised in Table 1. Values for $k$ are obtained from minimising the sum squared difference between the default and other



uptake forms. The nutrient uptake functions after optimization are shown in Fig. 1(a). The difference in shape of the optimised functional forms are more obvious at low nutrient concentrations.

## 2.2   Zooplankton grazing

In MEDUSA, both phytoplankton and zooplankton are grouped into "small" and "large" categories. The small microzooplankton graze on smaller phytoplankton and slow sinking detritus. The more nutrient rich, and therefore higher quality, non-diatoms are preferred over detritus. Larger mesozooplankton have a broader range of prey types, including both microzooplankton and diatoms which are regarded as higher quality food sources. When describing multiple grazing functions, the zooplankton grazing rate is often defined using either the hyperbolic Michaelis-Menten (Holling type II hereafter, $G_2$) or sigmoidal (Holling type II hereafter, $G_1$) expression, with maximum grazing rate $g_m$, and a weighted preference on the different food sources $p_n$ (Fasham et al., 1990). Since zooplankton preferences will change throughout the year, the assigned preference changes as a function of the food ratio. Holling type II and Holling type III, grazing on prey $Pa$ are described in table 1. In MEDUSA, the default multiple grazing parameterisation is based on the sigmoid Holling type III (Ryabchenko et al., 1997) function. Apart from the weighted preference, both of these functions also include a half saturation constant $k_x$, where $x$ is the zooplankton type.

These functions have similar behaviours where grazing rates saturate and become constant at a maximum grazing rate. At low zooplankton concentrations the sigmoidal response has lower grazing rates than the hyperbolic, and therefore, the sigmoidal curve has a more rapid increase in predation rate before becoming saturated (Edwards and Yool, 2000), shown on Fig. 1(b). Preferences for food types are kept the same as MEDUSA's default parameters, with terms summarized in Table 1.

## 2.3   Plankton mortality

Apart from grazing, plankton loss is caused by mortality. MEDUSA has two mortalities for all the phytoplankton and zooplankton types: density-independent and density-dependent. Density-independent loss terms are modelled by a linear function representing plankton metabolic loss which was kept unchanged. Density-dependent loss includes processes such as higher-trophic grazing and disease. In MEDUSA these processes are modelled using the hyperbolic function of plankton concentration (Fasham et al., 1993). As it is unclear which density dependent loss is the best choice, MEDUSA allows the option to include alternative functions to describe the density-dependent mortalities. We use the combinations of hyperbolic ($\rho_1, \xi_1$), linear ($\rho_2, \xi_2$), quadratic ($\rho_3, \xi_3$), and sigmoidal ($\rho_4, \xi_4$) functions to describe the phytoplankton ($\rho$) and zooplankton ($\xi$) mortalities (equations and abbreviations are shown on Table 1). Similar to grazing and nutrient uptake, the functional forms have different maximum rates for each plankton type. These maximum rates have made the same for all the different functions.

Of the four different mortality functions, linear and quadratic functions show the most distinct trends. Using the linear term is similar to a change in the value of maximum mortality rate, $\mu$. To make the linear function most similar to the sigmoidal and hyperbolic functions, the maximum grazing rate is set so that the total loss integrated over the range of prey density (calculated as the area below the line representing the total loss) is similar to that for the hyperbolic curve. The quadratic term, instead of asymptoting, continues to grow with plankton abundance. In order to keep this as 'similar' to other forms after reaching a certain



concentration, the mortality function is switched to linear, so that the rate reaches a plateau at high abundance. For sigmoidal mortality, the default $\mu$ are not changed but the half saturation constant, $k_M$ is optimised. The optimised mortality functions are shown in Fig. 1(c). A distinctive feature in the shapes of these functional forms after optimisation is that the quadratic mortality rate remains low until phytoplankton concentration reaches 10 mmol N m$^{-3}$, and the linear function always shows constantly

high plankton mortality rate (Fig. 1(c)).

## 2.4   Model Parameters

Apart from sinking rate, maximum growth, and grazing rates, parameters that are not listed in Table 1 are kept at their respective default values used in the MEDUSA model (Yool et al. (2011) shown on table 1-4). From a previous 3-D MEDUSA run, in the oligotrophic regions MEDUSA shows a low 'background' chlorophyll concentration (Yool et al., 2011). In order to raise this

concentration a higher maximum growth rate and lower grazing rate has been used. We chose the value for maximum uptake rate, $V_p$, as 0.8 day$^{-1}$, similar to that in the HadOCC model (Palmer and Totterdell, 2001). For zooplankton grazing, similar to NPZD models (Fasham et al., 1990; Fasham, 1995; Anderson et al., 2015) we use 1 day$^{-1}$ as the value for maximum grazing rate, $g_m$. MEDUSA also contains both slow and fast detritus sinking factors. It is assumed that the latter sinks rapidly relative to the model time-step, and remineralisation of the detrital nitrogen and silicon is done implicitly. In the default model 3 m

day$^{-1}$ is used for the slow sinking detritus, however over long runs we found this leads to downward loss of nutrients from the euphotic zone. Earlier studies have used lower detrital sinking rates (Steele and Henderson, 1981; Fasham et al., 1990; Lacroix and Gregoire, 2002; Raick et al., 2006), between 0 to 1.25 m day$^{-1}$. Therefore we chose a lower sinking rate of 0.1 m day$^{-1}$ to prevent depletion of state variables particularly at the shallower stations.

## 2.5   Running the Model and Generating the Ensemble

MEDUSA is run in the Marine Model Optimization Testbed (MarMOT-1.1) (Hemmings and Challenor, 2012; Hemmings et al., 2015), a site-based mechanistic emulator, where simulations are run in 1-D. MarMOT was developed to investigate the effect of sensitivity in plankton model simulations, especially in regard to parameter and environmental inputs (Hemmings and Challenor, 2012). Despite some uncertainties associated with the differences in physical forcing, fluxes, and initial values of biogeochemical properties, using 1-D simulations to approximate 3-D model behaviour for calibrating models based on

specific sites has improved the 3-D models' predictive skill (Oschlies and Garçon, 1999; Kane et al., 2011; McDonald et al., 2012). Physical and biogeochemical information are needed as input data in order to run MEDUSA. The 1-D MEDUSA is run at five oceanographic stations: PAP, ALOHA, BATS, Cariaco, and L4 shown in Fig. 2. These are chosen as they represent different oceanographic regimes: abyssal plain (PAP), oligotrophic (ALOHA, BATS), and coastal (Cariaco, L4).

At each oceanographic station, all combinations of the optimized functional forms (as described in subsection 2.1, 2.2,

and 2.3), are then embedded into the 1-D MEDUSA code. The same process function is always used for both diatoms and non-diatoms, or mesozooplankton and microzooplankton. The ensemble model at each station is initialized using the in situ measurements such as chlorophyll, nitrogen, silicic acid, and iron, and the ensemble is run over 10 years starting from January



1998. This provides a total number of 128 combinations, arising from 4 types of nutrient uptake functions, 4 types phytoplankton mortalities, 2 types of zooplankton grazing, and 4 types of zooplankton mortalities.

### 2.5.1 Physical input

Physical input files consist of gridded values of vertical velocity (m day$^{-1}$), vertical diffusion coefficient (m$^2$ day$^{-1}$), and temperature (°C), which are applied at each depth level. Additionally, time series of downwelling solar radiation (W m$^{-2}$) and mixed layer depth (m) are also used as input. These data are obtained from the 5-day mean output of the Nucleus for European Modelling of the Ocean (NEMO) model, using the Met Office Forecast Ocean Assimilation Model (FOAM). The FOAM-NEMO system assimilates *in situ* satellite SST, sea-level anomaly, sea-ice concentration, temperature, and salinity profile data, in order to make the physical system more realistic (Storkey et al., 2010). However, assimilating physical data directly into a coupled physical-biogeochemical model often does not improve the simulation of the ecosystem. For example a study by Ourmières et al. (2009) using the LOBSTER model, showed that although assimilating physical data improved the primary production in the Labrador Sea (due to increasing eddy activity), it does not improve the match to SeaWIFS derived chlorophyll-$a$. When assimilation is used in the 3-D HadOCC model it overestimates the nutrient concentrations due to spurious vertical velocities (Ford et al., 2012; Ourmières et al., 2009).

In our work the vertical velocities taken from the assimilated FOAM system were capped at the 90[th] and 10[th] quantiles, and the 10-year mean of the vertical velocity is also removed. This means that there is no time mean vertical velocity, and these adjustments are found to give a better long-term vertical structure to the nutrient and other distributions. Since input data on the vertical diffusivity coefficient was not stored from the assimilation run, we used values from NEMO ORCA025-N102 output from January 1998-December 2001 and ORCA0083-N01 from January 2002-December 2007, both were obtained from the CEDA Group workspace web (http://gws-access.ceda.ac.uk/public/nemo/#_top). Similar to the other physical inputs, vertical diffusivity coefficient from these NEMO outputs are 5-day averaged, common to 3-D MEDUSA (Hemmings et al., 2015).

### 2.5.2 Biogeochemical input and validation data

The input for the biogeochemical environment are the initial conditions for the 11 primary tracers (state variables) including; dissolved organic nitrogen (DIN), non-diatom, diatom, silicon in diatom, silica, detritus, microzooplankton, mesozooplankton, non-diatom chlorophyll, diatom chlorophyll, and iron (mmol m$^{-3}$), along with the model parameter values. Initial concentrations and therefore in situ data are taken from: (1) station ALOHA in the Pacific ocean (22°45'N 158°00'W) part of the Hawaii Ocean Time Series (HOTS), downloaded from http://hahana.soest.hawaii.edu/hot/hot-dogs/interface.html, (2) Bermuda Atlantic Time Series (BATS) in the subtropical North Atlantic (32° 50'N, 64° 10'W) available at http://bats.bios.edu/, (3) the Cariaco basin ( 10°30'N, 64°40'W ) obtained from http://imars.marine.usf.edu/cariaco, (4) Porcupine Abyssal Plain Sustained Observatory (PAP) in the Northeast Atlantic (Hartman et al., 2015) located in 49° N, 16.5° W, taken from http://projects.noc.ac.uk/pap/data, and (5) station L4, part of the Western Channel Observatory located at 50° 15'N, 4° 12.3'W (Smyth et al., 2010, 2015), and the data is available at http://www.westernchannelobservatory.org.uk/data.php. These stations are shown in Fig. 2. After initialization, in situ data from these stations are used to validate the model results. For station





PAP, we use SeaWIFS-derived chlorophyll-a data with 9 km spatial resolution and 8-day averaged provided by GlobColor (http://hermes.acri.fr/) for validating the surface chlorophyll.

At these stations, the DIN consists of ammonia, nitrate, and nitrite, however at oligotrophic stations like ALOHA the ammonium is below the detection limit (Hawaii Ocean Time Series), and therefore DIN only consists of nitrate and nitrite. At

station PAP, we use the initial condition from one of MarMOT's test stations, located at 50° N, 20° W (Hemmings et al., 2015), since the nitrate data was only collected between 30-400 m. At station L4, chlorophyll and nitrogen data were collected from the surface from 1999-2008. Therefore the initial concentration for chlorophyll and nitrogen are the same at every depth (total chlorophyll = 0.27 mg m$^{-3}$, nitrogen = 6 mmol m$^{-3}$). Other inputs that are not available at the websites mentioned above, such as microzooplankton, mesozooplankton, and detritus were taken from the the nearest test stations. In the oligotrophic

stations, 75 % of total chlorophyll was allocated initially to the non-diatom phytoplankton since these dominated the water column (Villareal et al., 2012). At the other stations, half of the total chlorophyll goes into the diatoms.

For validation of the model, we consider the total chlorophyll-a concentration, instead of separating diatoms and non-diatoms. Simulations are made at 37 depth levels, from 6-1200 m to minimise computational cost, except for coastal stations where the overall depths are shallower (up to 500 m for station Cariaco and 50 m for stations L4). The depth levels are similar

to that in ORCA025 NEMO model output. At the lowest level, vertical velocity and diffusion are set to zero and this level is used as the sink for detritus. Additionally, apart from the physical input files, a time series for soluble iron flux from dust deposition is applied but this is kept constant using the average value from (Mahowald et al., 2009).

## 2.6   Model Metrics

We use a list of statistical metrics, such as, correlation coefficient, root-mean squared error (RMSE), bias, ensemble range,

and 10-year mean, which averages the whole 10-year time-series at which both in situ data and ensemble results are available for that particular time, for the depth profiles of nitrogen and chlorophyll and integrated chlorophyll. For surface chlorophyll, apart from the metrics mentioned above, we used inter-annual mean, which averages the chlorophyll abundance at each year in order to see inter-annual variability, and monthly abundance, to observe the seasonal dynamics of chlorophyll. These statistical metrics are used to compare it with in situ data. Additionally, to capture the emergent properties of phytoplankton dynamics,

we consider the phenological aspects of the phytoplankton spring bloom, which are useful ecological indicators for detecting natural and anthropogenic impacts on the pelagic ecosystem (Platt and Sathyendranath, 2008). We consider seven phenology indicators as metrics to investigate how structural sensitivity affects the model simulations. These indicators are centered around the phytoplankton blooms. Before the blooms peak, we consider an initiation time where the chlorophyll concentration exceeds a certain threshold, in this case half the concentration of the bloom peak. When the bloom concentration starts to diminish, we

derived a termination time, where bloom concentration falls below the same threshold. The number of days when chlorophyll concentration is higher than the threshold is taken as the bloom duration. The concentration at the bloom peak and the date it takes place, are also included as indicators. Additionally, we also noted the amplitude of the bloom, which is half of the peak height minus the minimum chlorophyll concentration. The indicators are derived using the method described in appendix A, and applied to all ensemble outputs for each year.




In an ensemble forecast system, an ensemble with good reliability is the one that is statistically consistent with the observations, such that the observation is statistically indistinguishable from the ensemble members. In order to assess the value of the ensemble probability distribution we must assess the consistency of the ensemble spread as well as the ensemble mean error (Moradkhani and Meskele, 2010). A simple method is discussed by Anderson (2001) which takes the ratio $R_a$ of RMSE of the

ensemble mean and the mean RMSE of all the ensemble members which has the expectation value $E[R_a] = \sqrt{\frac{(n+1)}{2n}}$, where $n$ is the number of ensemble members. This is called the Normalised RMSE Ratio (NRR= $R_a/E[R_a]$) where the desirable ensemble spread is expected to have NRR=1. If the NRR >1 then the spread is too small, and NRR <1 indicates that the ensemble spread is too large. We may expect different NRR values for different metrics and also for variability on different timescales, such as monthly or inter-annual data. This method has previously been used to set the number of ensemble members

in data assimilation (Moradkhani et al., 2006; Roy et al., 2012).

## 3  Results

A selection of ensemble results are presented in order to provide a summary of the effect of perturbing model formulations of nutrient uptake, zooplankton grazing, and mortality simultaneously. These have been done at the five oceanographic stations which can be classified into three regional types: abyssal plain (PAP), oligotrophic (BATS and ALOHA), and coastal (Cariaco

and L4). First the ensemble range and mean are compared with the observational fields (described in the method section), followed by the error statistics calculated for the ensemble mean/median, the default run, and the ensemble range in order to assess whether the ensemble spans the observational data. Then variability from the default run and the ensemble are compared with the in situ data, followed by comparing the NRR to assess the ensemble spread, and phytoplankton phenology, in order to investigate whether the ensemble range has captured the events that lead to phytoplankton bloom initiation and termination.

### 3.1  Abyssal Plain

In this station, in situ nitrate was only measured from mid 2002 to mid 2004 with a maximum depth of 300 m and chlorophyll from mid 2003 to mid 2005 with maximum depth of 200 m. Surface chlorophyll is derived from SeaWIFs (8-day averaged) and is available for the 10-year time series (see supplementary Fig. S5).

Distinct seasonality has been simulated by the ensemble mean. High nitrate concentrations at the surface occur during the

winter (December-April) and decline in the summer. However, below 400 m a mostly continuous high ($> 10$ mmol m$^{-3}$) nitrate concentration is present, shown on Fig. 3(d). Chlorophyll concentration starts to decline at a depth of $\sim 50$ m, which also corresponds to the decline in the chlorophyll inter-quartile (between 25$^{th}$ and 75$^{th}$ percentile) range shown on Fig. 3(b). Chlorophyll also shows seasonality, similar to that in nitrogen. In the in situ, high concentrations of chlorophyll are recorded during May-June, in the top 70 m coinciding with the shallowing of the mixed layer depth. However in the model this occurs

earlier in spring (between end of April to May), and slightly deeper, to 100 m, as summarised in Fig. 3(a) and 3(c).

From table 2 chlorophyll profiles correlate better than nitrogen. Chlorophyll and nitrogen profile 10-year means are also within the ensemble range, although its ensemble spread is narrow as their NRR values are 1.20 and 1.25 for chlorophyll and




nitrogen respectively. In terms of chlorophyll and nitrogen profiles, the ensemble median shows the highest correlation and lowest RMSE and bias, compared to ensemble mean and default. High RMSEs in nitrogen occur from ensemble members that contain $U_2G_2$, $U_3G_2$, and $U_4G_2$ combinations, as shown in Fig. 7(t), which also correspond to high nitrogen mean ($< 5$ mmol m$^{-3}$), apart from ensemble members that contain $\rho_2\xi_3$, $\rho_1\xi_2$, $\rho_3\xi_3$, and $\rho_1\xi_4$ combinations. High chlorophyll RMSEs ($>0.62$)

are produced from members that combine $G_1$ with $\rho_1\xi_2$, $\rho_3\xi_3$, and $\rho_1\xi_4$ combinations. Similar to nitrogen, this coincides with high chlorophyll mean ($> 0.7$ mg m$^{-3}$). Surface chlorophyll 10-year mean and RMSEs ($>2$) are notably high when combining $U_2$ with $\rho_2\xi_3$, $\rho_1\xi_3$, and $\rho_1\xi_4$, as summarised in Fig. 8(e) and (j).

When compared to satellite-derived chlorophyll-a, the surface chlorophyll at this station has higher correlation than other regions, especially using ensemble median output, which also have lower RMSEs compared to the ensemble mean and default

run. In some years the satellite-derived chlorophyll is not within the ensemble range, due to the ensemble range overestimating the satellite-derived chlorophyll (supplementary material Fig. S5). Additionally, in terms of inter-annual mean, only in certain years (1998, 1999, and 2001) is satellite-derived chlorophyll within the ensemble inter-quartile range, although in other years they are within the full range, but outside the inter-quartile range, summarised in Fig. 4(a). This gives a "too narrow" ensemble spread, with NRR of 1.26. A decline in surface chlorophyll over the time has also been recorded in the satellite observations

(r= -0.21 p $< 0.05$), however only six ensemble members capture the decline in surface chlorophyll, with weaker correlations (r= -0.14 ($\pm 0.06$), p $< 0.05$). For monthly data, the satellite-derived surface chlorophyll concentrations are mostly within the ensemble range, and closer to the ensemble median, as shown on Fig. 5(a). In low chlorophyll months ($< 0.5$ mg m$^{-3}$) from November-March, the satellite-derived chlorophyll is within the 75$^{\text{th}}$ and 25$^{\text{th}}$ quartiles. Although in the time series the satellite-derived chlorophyll sometimes fall outside of the ensemble range, the overall ensemble monthly means show the

highest monthly mean surface chlorophyll concentration occurring between May and June, similar to the satellite-derived chlorophyll, shown on Fig. 5(a).

### 3.2 Oligotrophic Ocean

In oligotrophic regions nutrients are expected to be scarce at the surface but may be plentiful at deeper depths (Dave and Lozier, 2010; Lipschultz, 2001). All ensemble members represent this distribution well for ALOHA, as seen in Fig. 6(e), with nitrogen

levels $> 1.0$ mmol m$^{-3}$ only at $\sim 150$ m depth. However at BATS, from January 1999 the nitrogen concentration in the top 200 m is clearly overestimated, Fig. 6(k), with nitrogen levels $> 1.0$ mmol m$^{-3}$ at $\sim 10$ m (with some members occasionally showing such concentrations at 3 m). Higher ensemble inter-quartile ranges are found between 3-50 m, and this range decreases with depth, shown in Fig. 6(d) and 6(j). Mean nitrogen concentration is overestimated as indicated by the positive bias from the ensemble mean, as shown in Table 2.

Another feature of the oligotrophic ocean is a deep chlorophyll maximum (DCM) that occurs below the mixed layer (Fennel and Boss, 2003). In Fig. 6(b) and 6(h), high chlorophyll concentrations are simulated by the ensemble mean between 70-90 m in BATS and up to 150 m in ALOHA. A DCM occurs when lower chlorophyll is detected at the surface, which roughly matches with the in situ profiles at ALOHA (see Fig. 6(c) and Letelier et al. (2004)) and BATS (Fig. 6(i)) although the depth of the DCM is slightly shallower than in situ (down to 150 m). The high subsurface chlorophyll coincides with a higher ensemble



range, with the range decreasing with depth. However neither BATS nor ALOHA show a continuous DCM as seen in the in situ profiles, Fig. 6(c) and 6(i).

The majority of ensemble members underestimate in situ 10-year mean chlorophyll concentrations, especially at BATS where all ensemble members show positive bias towards both surface and integrated chlorophyll profiles. This in turn results

in NRR >1, showing that the ensemble spread is too narrow. At station ALOHA, in situ chlorophyll 10-year means (surface, profile, and integrated) are always within the ensemble range. In contrast, the modelled 10-year mean nitrogen from the ensemble mean and median are more than twice the in situ observations, also leading to a narrow ensemble spread with the ALOHA NRR value for nitrogen being the largest, summarised in Table 2. At BATS some members show a very low chlorophyll mean ( $< 0.015$ mg m$^{-3}$) and high nitrogen concentrations ($> 0.34$ mmol m$^{-3}$), see Fig. 7(b), 7(f), 8(l), and 8(q), resulting in high

RMSEs for both variables. Most of these members use functional combinations $G_2$, $\rho_2\xi_2$, $\rho_2\xi_3$, $\rho_2\xi_4$, $\rho_3\xi_1$, and $U_3$. The low chlorophyll concentrations, coinciding with high RMSEs, also come from the same ensemble members as for station ALOHA, except for $U_3$. Ensemble members that use $U_1G_1$ and $U_4G_1$ show higher profile 10-year mean concentration of chlorophyll at both stations, Fig. 7, although when paired with $\rho_3\xi_3$ and $\rho_1\xi_4$, the RMSEs increase. High nitrogen concentrations are almost always observed when $U_3$ and $U_2$ were used in the oligotrophic regions.

Surface chlorophyll at ALOHA (supplementary Fig. S2) has lower RMSEs and higher 10-year mean concentration compared to those at BATS, summarised in Table 2. Low chlorophyll with high RMSEs ($> 0.1$) have not been observed in station ALOHA. Ensemble members with lower surface chlorophyll concentrations were similar to the observation profiles, and high surface chlorophyll RMSEs coincide with high surface concentrations, summarised in Fig. 8(a) and (f). The low RMSEs for surface chlorophyll at ALOHA is also reflected in the NRR, with a value (NRR= 1.07) close to unity, although slightly narrow,

and the ensemble almost always encompasses the in situ observations. During low chlorophyll months (June-September), most of the ensemble members still underestimate the in situ monthly mean, summarised in Fig. 5(b), and not all peaks are covered by the ensemble.

Figure 4(b) and (c), shows that there is no distinct inter-annual variability at either ALOHA or BATS. Figure 5(b) and (c) shows that the highest mean of in situ chlorophyll concentration is usually found in December (0.13 mg m$^{-3}$) and April (0.28

mg m$^{-3}$), at station ALOHA and BATS respectively, and these are within the ensemble range. At station BATS in 2004, high in situ chlorophyll mean was recorded (0.65 mg m$^{-3}$) that was not captured by all of the ensemble members, see the supplementary material Fig. S1 and 4(c). Since model outputs at BATS have a lower 10-year mean chlorophyll than in situ data, most of the ensemble members underestimate the surface inter-annual means, therefore making the ensemble spread too narrow both in the 10-year mean and the inter-annual mean, shown on Table 2 and 3. This is also reflected in the monthly mean,

whereby in the months of low concentration in the ensemble and in situ ($< 0.1$ mg m$^{-3}$ occurring in July-October), the in situ concentrations are above the ensemble range. At ALOHA, the inter-annual mean, the in situ data are mostly within the 75[th] and 25[th] quartiles. This is also shown in the monthly means, especially during high in situ ($> 0.1$) surface chlorophyll months from November-January, summarised in Fig. 5(b). However, the range of the ensemble for inter-annual mean at station ALOHA is seen to be too wide, with NRR of 0.84, as the in situ inter-annual means are mostly closer to the 75[th] quartile, making the mean

RMSE of the ensemble higher than the ensemble mean's RMSE (0.043 and 0.025 respectively).



At station ALOHA the ensemble mean and median produce smaller errors for both chlorophyll and nitrogen. In the depth profiles, bias in the default run is smaller than for the ensemble mean and median. However the surface and integrated chlorophyll show that the ensemble mean and median produce lower bias than the default concentrations. This is the opposite for BATS where both RMSEs and correlation coefficient are higher for the default run compared to ensemble mean and median, as well as for the biases. At both stations, integrated chlorophyll from ensemble mean and median shows smaller RMSEs and a better correlation coefficient, compared to the default run. At ALOHA, NRR for the integrated chlorophyll is closer to 1 compared to either the surface and profiles. However the default run in oligotrophic regions generally produces higher chlorophyll and lower nitrogen concentrations compared to the ensemble mean and median. This also matches better with in situ patterns as the correlation coefficient, $r$ is higher. This is because using $U_1G_1$ gives rise to higher chlorophyll concentrations.

## 3.3 Coastal

In the coastal stations, in situ observations show strong seasonality, shown on Fig. 9(c), (f), (g), and (h) . In terms of the inter-annual mean, the ensemble range at stations Cariaco and L4 always includes the observations (Fig. 4(c) and (d)), despite the ensemble spread mostly being quite narrow, as described by the NRR values in Table 2.

The in situ profiles at Cariaco show high chlorophyll concentrations ($>1$ mg m$^{-3}$) within the upper 30 m occurring between January to February (see Fig. 9(c)). This coincides with the rise of nitrogen from deeper depths to $\sim$30 m, as seen on Fig. 9(f), increasing the nitrogen concentration to $\sim 5$ mmol m$^{-3}$. However this is not captured by the ensemble mean, with chlorophyll concentration almost constant above 0.7 mg m$^{-3}$ in the upper 30 m, as shown in 9(a) and the surface (supplementary Fig. S3). A decline of chlorophyll has been recorded in station Cariaco from 2004 (Taylor et al., 2012), and this has been captured by the ensemble mean, median, and default (r= -0.72, p< 0.05, r= -0.66, p< 0.05, and r=-0.35, p< 0.05 respectively). For nitrogen the seasonal upwelling is not captured, although in 2001, and between 2005-2006, downwelling of nutrients are reproduced, summarised in Fig. 9(d). Figure 9(e) shows the inter-quartile range for nitrogen increasing below $\sim$40m and then decreasing again at $\sim$100 m. Similarly on Fig. 9(b) chlorophyll interquartile range is high at depths where chlorophyll is plentiful.

At station L4 in situ and ensemble means both show seasonality of nitrogen with high concentration ($>8$ mmol m$^{-3}$) occured during November to February, and close to zero ($> 0.1$ mmol m$^{-3}$), during summer months consistent with the observation from Smyth et al. (2010). For chlorophyll Figure 9(g) shows sharp peaks in spring time (March-April) and fall (September) for in situ data, and the ensemble means peak around one month later (May-June), without a distinct secondary peak, similar to the typical North Atlantic spring bloom (Siegel et al., 2002; Behrenfeld et al., 2013). Observed chlorophyll concentrations generally range from 0.09-2 mg m$^{-3}$, apart from the sharp increases during bloom events (up to 6.41 mg m$^{-3}$), yet during non bloom period, ensemble range from 0.28-3.13 mg m$^{-3}$ and during bloom events, the highest peak is 5.95 mg m$^{-3}$, therefore the surface chlorophyll is not fully captured by the ensemble. This is reflected by the high NRR value of 1.31, indicating a narrow spread.

Both stations show weak positive correlations of surface chlorophyll from the ensemble mean, summarised in Table 2. The ensemble mean and median show better correlation and smaller RMSEs compared to the default run, apart from nitrogen at station L4. Chlorophyll is biased at both stations, for the ensemble mean at Cariaco, and for the ensemble median at L4.



Integrated chlorophyll shows better correlation with in situ observations at station Cariaco compared to both surface and chlorophyll profiles. Nonetheless, compared to other oceanic regions, Cariaco still has the highest RMSE for both chlorophyll profile and surface values. At L4, the ensemble mean shows high RSME for surface nitrogen, but low RMSE for surface chlorophyll, see Table 2.

Although from Table 2, in situ surface chlorophyll concentrations are slightly overestimated by the ensemble mean, most of the ensemble outputs at Cariaco are underestimated, except for ensemble members that use the combinations $\rho_2\xi_3$, $\rho_1\xi_2$, $\rho_3\xi_3$, and $\rho_1\xi_4$. This in turn makes the ensemble spread narrow, as indicated by the NRR value. Unlike the oligotrophic regions, these high chlorophyll concentrations also coincide with higher RMSE ($> 1.7$). Higher nitrogen concentrations ($>1.2$ mmol m$^{-3}$) with high RMSEs ($> 1.5$) are also associated with the same ensemble members. Despite this, these members show relatively

low nitrogen concentration ($>7$ mmol m$^{-3}$) at station L4. The chlorophyll mean at L4 shows that high concentrations ($>$ 0.9 mg m$^{-3}$) are produced when $G_1$ is paired with $\rho_1\xi_2$, $\rho_3\xi_3$, and $\rho_1\xi_4$. These also coincide with high RMSEs, especially in members which pair $U_1$ and $\rho_1\xi_2$, $\rho_3\xi_3$, and $\rho_1\xi_4$. Low chlorophyll concentrations and RMSEs at the coastal stations are produced from $U_2G_2$ and $U_3G_2$, and additionally $U_3G_1$ in station L4. High nitrogen concentrations ($> 9$ mmol m$^{-3}$) are produced by $U_4G_2$, with correspondingly high RMSE. Surface chlorophyll at coastal stations has a higher relative range than

other stations, with L4 showing the higher range compared to Cariaco, summarised in Table 2. Despite having lower range than L4 in terms of surface 10-year mean, in the annual mean (Fig 4(d) and (g)), the NRR value for Cariaco is too small (0.78), indicating the ensemble spread is wider than necessary.

At station L4, the in situ inter-annual means are closer to the ensemble median, indicated by the smaller bias and RMSE compared to both the default and ensemble mean, shown on Table 2 and Fig. 4(e). Despite the narrow ensemble range in the

overall mean of surface chlorophyll, the spread for the overall annual mean is reliable (NRR= 1.001) and in situ means are almost always close to the ensemble mean and median. In the monthly means, shown in Fig 5 from September-April, in situ observations are within the ensemble range, however, in the summer months when chlorophyll starts to decline (May-August) due to the exhaustion of nutrients, in situ monthly means are below the ensemble range. This in turn indicates that seasonally, the ensemble does not always cover the in situ observations, making the spread for the overall 10-year mean too narrow. The

highest in situ monthly mean chlorophyll concentration occurs in April, yet the ensemble mean and median show this peak in June, and the default run in May. There are also two peaks that occur in the in situ monthly means, one in April, and the other in September. If only diatom chlorophyll concentrations is shown, the two bloom events are shown better, especially in the default run (see supplementary material Fig. S4).

### 3.4 Phytoplankton Phenology

At most stations, the phenology metrics are covered by the ensemble range. There are differences in the timing of phenological events between the ensemble mean, median, and default run, ranging from a couple of days to a couple of weeks, as shown in Table 3. The timing of initiation, bloom peak, and termination show wide interquartile ranges for all stations and can lie between ~20 and 100 days earlier than the in situ timing, apart from stations PAP and ALOHA, see Fig. 10(b). At stations PAP and ALOHA the inter-quartile range is at least ~40 days too early. However, the ensemble mean and median at station L4





and Cariaco are later than in situ timings. For initiation both stations are two months late and are within the ensemble range. In terms of the timing of the bloom peak and termination, they are up to 3 months late 120 days too late respectively.

BATS has the largest range of phenological timings, especially in termination time. In terms of initiation, the in situ timing is within the interquartile range and only three days earlier than the ensemble median. However, in ALOHA the initiation time shows more inter-annual variability (supplementary Fig. S6) eg. in some years bloom initiation may occur in June, August and October, as well as December and January. This causes the mean observed initiation time to become in May. From Fig. 10(a), the ensemble run shows a mean initiation time between late January and April instead and so the observations fall outside the ensemble range. Due to this variable initiation, although bloom time is within the ensemble range at ALOHA, the timing is outside the 75$^{th}$ and 25$^{th}$ percentile range, making the ensemble spread too narrow (NRR=1.35). The peak chlorophyll at ALOHA shown in Fig. 5(b), where high ($> 0.1$ mg m$^{-3}$) chlorophyll monthly means are recorded in June, August, and September as well as December and January, yet the ensemble mean and median show highest concentrations only in January and February, also placing the bloom timing outside the inter-quartile range, summarised in Fig. 10(b). At BATS the earliest initiation is mid January in the ensemble, but the earliest in situ initiation occurs in February. Therefore, peak bloom time from the ensemble at BATS are usually later than in situ. However, ensemble estimates of bloom peaks for 30°N, where BATS is located, agree with a study by Racault et al. (2012), who identify early April as the peak time.

Both coastal stations show in situ initiation typically happens in mid-March, and these are within the ensemble range, which spans for 100 days (between the end of February and late June). The ensemble means show later initiation, with the 75$^{th}$ and 25$^{th}$ spanning mid April to end of May for Cariaco, and between early and mid May for L4. This later timing is also clear in peak bloom times, shown on Fig. 10(b). Figure 5(e) shows the in situ bloom at L4 is one to two months overestimated by the ensemble. Cariaco is the only station with peak bloom time, duration, and termination outside the ensemble range, due to the lack of chlorophyll seasonality, as explained in section 3.3. This results in the timing of initiation, bloom peak, duration, and termination having high NRR values.

Initiation timing is captured best at station PAP, with the ensemble median's initiation only averaging eight days earlier than for the satellite-derived chlorophyll, resulting in NRR for initiation closest to one (1.14) compared to other stations. A typical North Atlantic bloom happens during spring (Raymont, 1980), however most blooms at PAP occur in late May-early June, as shown in Table 3 and Fig. 5. Later blooms are recorded from satellite-derived chlorophyll-a in 2005, three months later than the average and much later than the ensemble mean and median, although the bloom timing is still within the ensemble range, although the range itself is still narrow, according to the NRR value (1.31). At L4, also in the North Atlantic, the spring bloom is in April, but most ensemble members show later initiation and peak bloom time, mostly in June. Due to this delay the NRR values at L4 indicate that the ensemble range is too narrow, although still within the full ensemble range. Ensemble mean and median at PAP show good agreement with in situ termination date. Although other station termination times are also within the ensemble range, most are later than the inter-quartile range. However, at ALOHA, located at 22°N, the ensemble median for termination at the end of August falls close to the observations from Racault et al. (2012).

Compared to running only the default MEDUSA, where only a single mean peak value is produced, the ensemble range mostly encompasses the in situ peak amplitudes, shown on Fig. 10(c). Only at BATS are the in situ peak height and amplitude





outside the ensemble range, resulting from the narrow ensemble range seen from the NRR value. This is expected since most of the ensemble members underestimate in situ chlorophyll. At Cariaco, in situ peak heights are within the ensemble range, but observed peaks are higher (mean= 3.5 mg m$^{-3}$, maximum peak= 7.7 mg m$^{-3}$), and the ensemble reaches less than half of the in situ peak (mean= 1.2 mg m$^{-3}$, maximum height= 5 mg m$^{-3}$). This underestimates the peak and consequently also the

amplitude, resulting NRR of 1.40 and 1.39 respectively. Ensemble members with higher peak and amplitudes are also those with higher chlorophyll biases. Despite the narrow ensemble range, at L4 chlorophyll peaks are within the 75$^{th}$ and 25$^{th}$ range box, and its amplitude is within the full spread. In contrast stations ALOHA and PAP have reliable ensemble spreads according to their NRR values for peak height (see Table 3).

  Similar to peak heights, the bloom durations at most stations are within the ensemble range, apart from station Cariaco,

which shows the narrowest ensemble spread according to its NRR value. The duration at Cariaco is overestimated because the peak is very wide (up to 143 bloom days). This, along with the late initiation of the bloom, results in a three month late termination. At ALOHA, duration is outside the 75$^{th}$ and 25$^{th}$ quartile box, since the peak is also much broader compared to in situ blooms. This results in too narrow ensemble mean according to the NRR value. The opposite is true at BATS where in situ peaks are generally broader, and the ensemble members with lower chlorophyll concentration showing narrower peaks, and a

greater range in bloom durations, which consequently lowers the NRR value.

## 4 Discussion

In this paper we have investigated structural sensitivity of an intermediately complex biogeochemical model by generating its ensemble outputs of chlorophyll and nitrogen and comparing with a single default run, and with in situ observations at five oceanographic stations. The ensemble consists of 128 ensemble members, each with different process function combinations.

Following the work of Fussmann and Blasius (2005), these functions have been previously calibrated, using non-linear least squares, and keeping the maximum process rates fixed in order to maintain phenomenological similarity. We have chosen nutrient uptake, zooplankton grazing, and plankton mortalities to vary, as these are the core processes of every biogeochemical model, from the simplest to the most complex.

  Most current biogeochemical models are run in a deterministic, rather than a probabilistic, manner, even though data from

observations contain many uncertainties, eg. in satellite-derived chlorophyll. For physical models, perturbed parameter ensembles have been explored and utilized to quantify climate change uncertainties in a probabilistic sense (Murphy et al., 2007; Tinker et al., 2016). Here we provide a perturbed biology ensemble conditioned upon process structural uncertainties. Applying structural sensitivity in the 1-D framework has also allowed a large parameter space of concurrent variations to be explored, for several different oceanographic regions, and with minimal computational cost. From these assessments, we find that small

perturbations in model structure can produce a wide range of results, particularly regarding phytoplankton phenology. Apart from the assessment of uncertainties arising from the structural sensitivity and the reliability of the ensemble spread, we have also compared the RMSEs against observations for the ensemble mean and median, and for the deterministic model default run.





Our findings reveal that in all regions the Holling Type II ($G_2$) grazing function decreases the chlorophyll concentration, and pairing it with linear phytoplankton mortality ($\rho_2$) lowers the concentrations even further. The nutrients respond in the opposite direction with enhanced nitrogen concentrations. Yool et al. (2011) has similarly shown that using a linear mortality causes the biggest changes, and Anderson et al. (2010) show that type III or sigmoidal grazing depletes less phytoplankton at

low concentrations compared with hyperbolic grazing. It is therefore consistent that the lowest chlorophyll concentrations are observed from the combination of these functional forms. We found that default phytoplankton mortality ($\rho_1$) and sigmoidal zooplankton mortality ($\xi_4$) produce the highest chlorophyll concentrations in all regions, similar to the experiment from Yool et al. (2011). Linear zooplankton mortality ($\xi_2$) produces enhanced chlorophyll concentrations due to higher zooplankton mortalities and lower phytoplankton mortalities, especially when combined with the default hyperbolic phytoplankton mortality

($\rho_1$). In terms of uptake, the exponential ($U_3$) and sigmodal ($U_2$) functions show low chlorophyll and high nitrogen concentrations, especially in the oligotrophic regions. Figure 1 shows that in low nitrogen regions, uptake rates using $U_3$ and $U_2$ are lower than those using the default michaelis-menten ($U_1$), or trigonometric ($U_4$), functions. Yet, the differences when using $U_3$ and $U_2$ compared to using $U_4$ and $U_1$ are not as large as using type II grazing and linear mortality on chlorophyll concentrations at the oligotrophic stations. Another example is found at station L4; when pairing $U_4$ uptake and $G_2$, the ensemble produces

high nitrogen concentrations but low chlorophyll concentration is not seen, Fig. 8(n). This is because when phytoplankton concentration is >1 mg m$^{-3}$, $G_2$ depletes less phytoplankton compared to $G_1$, and combining this with $U_4$ lowers the uptake rate, as summarised in Fig. 1, leaving higher nitrogen.

As an additional metric for the ensemble spread we computed the Normalised RMSE Ratio (NRR) (Anderson, 2001) to measure whether the ensemble has a reasonable spread, which could then be regarded as an uncertainty when using the results

of the model simulations. The NRR values for the five oceanographic stations indicate that the spread is usually too narrow, especially for nitrogen profiles, with the smallest NRR station L4, and the largest at station ALOHA, shown in Table 2. Nitrogen in situ mean concentrations are generally overestimated by ensemble especially in the oligotrophic region. But the model well represents days when concentrations are high, and for chlorophyll at the oligotrophic stations, only very low chlorophyll concentrations fall within the ensemble range (see supplementary material Fig. S1 and S2).

Overall, station Cariaco shows the highest chlorophyll RMSEs. This station also produces model outputs in which high chlorophyll concentrations coincide with high nitrogen. Consistently high chlorophyll concentrations are seen throughout the time series, while the observations only show a phytoplankton bloom once per year, leading to the higher chlorophyll RMSE than at other stations. The observed timing of the bloom, duration, and termination are outside the ensemble range, and consequently the ensemble range is assessed as too narrow according to its NRR value (>1.39). At this station, chlorophyll

is mostly driven by the upwelling of nutrients caused by the trade winds (Lorenzoni et al., 2013). However, the upwelling is not captured well by the assimilated vertical velocity we used. Instead of chlorophyll peaks occurring between December-January throughout the time-series, the ensemble produces a constantly high chlorophyll concentration, summarised in Fig. 5(d). Despite these discrepancies, the chlorophyll profile at Cariaco has an NRR value closest to 1, but not for the surface annual mean. Although the in situ inter-annual mean concentrations are almost always within the inter-quartile range (see Fig.




4(d)), some of the means are either closer to the upper quartile or the lower quartile. This in turn widens the ensemble spread (NRR=0.78).

The NRR for chlorophyll (mean, surface, and integrated) at station BATS is the highest and BATS is the only station at which the default run performs better than the ensemble mean for RMSEs of chlorophyll and nitrogen concentration. The

low concentrations modelled at BATS give the high NRR values for surface and profile integrated chlorophyll, indicating too narrow ensemble spread. Only the lower or 'background' chlorophyll are inside the ensemble range, but not the higher bloom peaks. Peak height and amplitude in the phenology are therefore underestimated. However in the monthly means (Fig. 5(b)) from December to April, the months where modelled chlorophyll means are high, in situ means are within the ensemble range. This is also observed at ALOHA, showing that in oligotrophic regions during higher chlorophyll concentrations, the ensemble

range increases to encompass the in situ concentrations. The higher in situ chlorophyll concentration at BATS compared to ALOHA may be because the station is in the mode-water region of the subtropical gyre and has deeper mixed layers allowing deep nutrients to reach the surface and supply the growth of phytoplankton. In the ensemble run, the mean chlorophyll profile concentrations are very similar between ALOHA and BATS, shown on Table 2. At the surface, ALOHA shows higher mean chlorophyll than BATS, perhaps due to high concentrations ($\sim 0.24$ mg m$^{-3}$) in 1999, simulated at the surface at ALOHA,

contributing to the overall mean, while at BATS steady low chlorophyll concentrations are found throughout the ensemble. Lower modelled concentrations of chlorophyll, compared to the in situ, in the subtropical gyre have also been observed in the 3-D default MEDUSA model itself (Yool et al., 2011).

Although also oligotrophic, station ALOHA shows the lowest RMSE for both nitrogen and chlorophyll, compared to other stations, especially for surface chlorophyll. At this station, the surface and integrated chlorophyll, peak height, and amplitude

have the NRR close to 1 (see Table 2 and 3). However for bloom initiation ALOHA in situ timing is outside the ensemble range. In some years, the observed bloom initiation has been recorded from June to August and these patterns are captured by the ensemble. However in most years, modelled bloom initiations are between December and January. The summer chlorophyll bloom that occurs in the north pacific subtropical gyre consists of picoplankton (White et al., 2015) and may be caused by the addition of nitrogen from nitrogen-fixing organisms (Dore et al., 2008) which have not been explicitly represented in

MEDUSA. This may explain the discrepancy in bloom timing. At BATS in the Atlantic subtropical gyre, phytoplankton surface blooms occur in the spring time. The initiation, bloom time, and duration are within the ensemble range, summarised on Table 3. The large range is due to the initiation time varying strongly with concentration, therefore also affecting the duration. For example, high concentration chlorophyll produces earlier initiation times (February-April), and years with lower concentrations show later initiation times (May-June).

Station PAP shows the best match between the observed bloom and the ensemble. At PAP, seasonality is very well defined with both nitrogen and chlorophyll concentrations affected greatly by the mixed layer depth. Phytoplankton dynamics in regions like the North Atlantic is dictated by the mixed layer deepening and the enrichment of nutrients in the surface layer (Behrenfeld and Boss, 2014). In terms of peak height and amplitude, although within the ensemble range with NRR values close to unity, they are still larger than 1, indicating a slightly narrow spread (see Table 3). In some years (eg. 2002 and 2006), the peak height





is overestimated by most the ensemble, and in the monthly means, the month where the ensemble concentrations peak, is later, summarised in Fig. 5.

At station L4, the annual mean of surface chlorophyll has a reliable spread. In situ initiation, bloom timing, and duration are earlier than in most of the ensemble members, although still lying within the ensemble range, despite this being narrow
by the NRR. Some ensemble mean timings are similar to the satellite observations at this latitude (Racault et al., 2012), such as termination and peak bloom time. When in situ chlorophyll is fitted with a smooth curve, the highest peak mostly occurs during spring (March-April). But model metrics, including ensemble mean and median, are noisy, and peaks mostly fall in the summer (May-July), although the peak height is usually within the ensemble range. At L4, distinct phytoplankton blooms occur twice a year: in spring and the second in fall (Smyth et al., 2010). These blooms are sometimes well simulated, eg. in Fig.
9(g) and 5(d), but are not as distinct as in situ measurements because of the variability of the model. However, the difference in peak timing does not affect the duration of the blooms, and the in situ duration is well within the ensemble inter-quartile range.

At some stations, we have observed different NRR value for the inter-annual means (shown on Table 3 and Fig. 4) and the monthly data (shown on table 2 and Fig. 5). The discrepancy may be because in the monthly data, not all in situ peaks and troughs are being covered by the ensemble spread, but the inter-annual in situ means are almost always within the ensemble
range. The most notable difference is in the coastal stations where for station Cariaco, the inter-annual in situ means are mostly in the upper quartile, summarised in Fig. 4(d). The addition of the ensemble members that produced lower than average concentration therefore widens the overall ensemble spread, reducing the NRR. On the other hand, at station L4, the NRR for annual mean is close to unity because most of the year, the in situ data is very close to the ensemble median, see Fig. 4(e), with $R_a$ value of 0.711. In terms of phytoplankton phenology, BATS ensemble has the highest range of peak bloom time (174
days), however the NRR suggests too narrow range (1.17). This is because most of the ensemble members produce blooms between April and May, and the in situ timing occurred in the 29 March and so the in situ mean bloom time is still outside the interquartile range. The large range is caused by some ensemble members blooming much later. However, since the in situ timing is earlier, it is not within most of the ensemble range, so the overall ensemble spread is deemed narrow.

We have chosen phytoplankton phenology to define model metrics because of its importance to marine ecosystem produc-
tivity, eg. Cushing (1990) show that the survival of zooplankton and fish larvae is affected by the timing of phytoplankton blooms. The timing of the blooms have also been shown to control the variability of $pCO_2$ in the sub polar region (Bennington et al., 2009). Despite the importance of bloom timing, discrepancies in predicting bloom timing by large-scale biogeochemical models is reported in many studies, e.g., Henson et al. (2017) and Kostadinov et al. (2017). Henson et al. (2017) shows that the 3-D MEDUSA 2.0 (Yool et al., 2013) model initiates spring bloom start dates ∼50 days early, and in the southern hemisphere
it initiates them ∼50 days late. By generating an ensemble of 7 CMIP5 models, Kostadinov et al. (2017) highlighted that the difference in bloom timing between the model ensemble and satellite-derived chlorophyll can be more than one month over most of the ocean. This agrees with our study (see, Table 3), as most of our ensemble members have earlier bloom initiation dates, and the difference between the ensemble mean and in situ timing of bloom, eg. PAP and L4, are more than one month. However, the use of the whole ensemble range, can help to provide an uncertainty range for the timing of phytoplankton
blooms. By utilising the ensemble, start date differences may be reduced. The ensemble range almost always encompasses the



observed annual mean, peak height, and amplitude. Therefore it may be suitable to use the ensemble model in order to forecast these phenological aspects.

## 5 Conclusions

Our study highlights that it is important to conduct structural sensitivity analyses in addition to parameter sensitivity analyses. It
is crucial to include mathematical functions that can capture sufficient information of the key biogeochemical processes known from experimental studies. However, none of the deterministic functions can capture all details of these processes (Anderson et al., 2010), therefore we have introduced a method whereby instead of having only one default model output, we have an ensemble generating a range of possible outcomes arising from alternative model structures. We have explored the structural sensitivity of the 1-D version of MEDUSA, the ocean biogeochemistry component of UK-ESM1, to reduce the errors between
in situ and model outcome. This study emphasises that small perturbation in MEDUSA process structure can produce very different model results.

    Linear phytoplankton mortality and hyperbolic (Holling type II) grazing generally produces lower chlorophyll concentrations, thereby reducing the ensemble mean and median chlorophyll concentrations. In regions of low nitrogen, sigmoidal and exponential uptake produce high nitrogen concentrations. The spread of the ensemble for chlorophyll and nitrogen profiles is
the widest at coastal stations (NRR between 1.19-1.31), and narrowest at the oligotrophic stations (NRR 1.29-1.40). However, a reasonable range of ensemble is produced at oligotrophic station ALOHA, with NRR values of 1.01 and 1.07, for 10-years of integrated and surface chlorophyll time series respectively, and L4 for surface inter annual means, with NRR value of 1.001. The ensemble range mostly covers the in situ observations (particularly the surface overall annual mean) of chlorophyll and nutrients at all stations. At oligotrophic stations, the ensemble mean and median tend to simulate lower chlorophyll concentra-
tions and higher nitrogen concentrations compared with the default run and the in situ observations. However, at coastal and abyssal stations, the ensemble mean and median tend to overestimate chlorophyll, particularly during low chlorophyll months (June to September) in the North Atlantic. This in turns means that the spreads are wider for the inter-annual means at most of the stations, since the in situ inter-annual means are within the ensemble range, whereas in the monthly data the ensemble often doesn't capture the peak, making the NRR narrow. For phytoplankton phenology, the observed bloom initiation, peak
bloom time, and termination, are all within the ensemble range at most stations. Although the ensemble members mostly show late blooms compared with in situ, the bloom durations are still within the 75[th] and 25[th] percentiles of the ensemble.

    Our study shows promise that an ensemble biogeochemical model resulting from perturbing the model structure, can produce a meaningful range of chlorophyll and nitrogen. However, our study is based on 1-D simulation, and further study with a 3-D biogeochemical model would help extend results to the global ocean. It may also be possible to further minimise the
computational costs by systematically reducing the number of ensemble members whilst retaining a realistic ensemble range. Further studies could include varying weighting of ensemble members, or reducing the number of model combinations to improve the ensemble range and to assess properly different plankton functional types and dissolved inorganic carbon. Such a perturbed biology ensemble may also be used for data assimilation eg. with satellite-derived chlorophyll.





*Data availability.* The raw model outputs will be available at Pangaea after the manuscript has been published and upon request from the authors (p.anugerahanti@pgr.reading.ac.uk, shovonlal.roy@reading.ac.uk)

**Appendix A: Determining phytoplankton phenology**

Before determining the initiation time, bloom timing must be identified. This is done by taking the ten years of surface chloro-

phyll output and breaking it down into individual years. These are then rearranged into two datasets: January-December and June-May, and the date of maximum chlorophyll concentration in each year is determined. If the peak timing occurs mostly towards the end or the beginning of the year, June to May datasets are used instead of the former. The timing is then adjusted if the calendar year has changed.

The initiation is determined by the day that chlorophyll concentration exceeds a given threshold. However, since in situ

chlorophyll has some data gaps and modelled chlorophyll is not smooth, some studies have fitted a function or model to the datasets to make the chlorophyll data smoother (Platt et al., 2009; Sapiano et al., 2012; Brody et al., 2013). Here we use a 5th order polynomial curve to get a smooth fit of the bloom peaks in the data (Fig. A1), from which phenology metrics are calculated. After being fitted, a threshold of half the bloom peak concentration is chosen. To find the **peak time**, the date at which maximum chlorophyll concentration is achieved in the fitted curve is determined, and this date is used as a reference to

calculate other metrics. **Amplitude** is then calculated as half of the highest peak minus the minimum concentration. **Initiation** is the day when chlorophyll concentration goes just above the threshold towards the maximum (Brody et al., 2013). **Termination** of the bloom is defined when concentration falls below the threshold (Racault et al., 2012). If two peaks are detected the termination of the spring bloom is determined when the first bloom reduces to its minimum, just before the second bloom starts (in the first valley). **Duration** of the bloom is simply the total number of days on which chlorophyll concentration is

above the threshold or termination minus initiation.

This phenology is useful to see how the bloom develops and terminates, whether the concentration increases rapidly and decreases slowly or vice versa. The phenology is summarised in Fig. A1. The curve fitting method is only applied if the data shows potential outliers especially in higher concentrations. If there is only one prominent bloom each year, as at stations ALOHA and BATS, and the data is smooth, the regular threshold method (when the concentration is above 50% of the max-

imum bloom, and the associated initiation and termination times), without fitting the data with a curve is applied. To avoid results being affected by how bloom phenology is determined, the same method is used for determining the metrics from both in situ and model output.

*Competing interests.* The authors declare that they have no conflict of interest





*Acknowledgements.* This study was funded by the Bakrie Center Foundation (grant no. 1307/BCF-SK/RSCH/VII/2015), Indonesia. The authors would like to thank Kevin White, for his advise on this study and John Hemmings, for providing the most recent version of the MarMOT code. We would like to thank Ruth Airs and Denise Cummings for providing the L4 chlorophyll and nutrient measurements. This study uses various oceanographic station data and we would like to thank all crews and scientists involved in collecting, processing, and making the data publicly available.



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





## Tables

**Table 1.** Parameter values for resource uptake ($U$), zooplankton grazing ($G$), and plankton mortalities ($\rho$ and $\xi$ for phytoplankton and zooplankton respectively), described using similar functional forms (shown in Fig. 1). In grazing equation, $g_m$ represents maximum grazing rate, $P_a$ is the prey, and $p_a$ denotes the grazing preference. Starred equations are the default functional responses in MEDUSA.

| Process/ Plankton type | Symbol | Meaning | Parameter value (mmol m$^{-3}$) | | | |
|---|---|---|---|---|---|---|
| **Nutrient Uptake** ($U$) | | | Monod* | Sigmoidal | Exponential | Trigonometric |
| | | | ($U_1$) | ($U_2$) | ($U_3$) | ($U_4$) |
| | | | $\frac{n}{n+k}$ | $\frac{n^2}{n^2+k^2}$ | $1 - \exp(-\frac{n}{k})$ | $\frac{2}{\pi}\arctan\left(\frac{n}{k}\right)$ |
| Non-diatom | $kN_{nd}$ | shape defining constant for nitrogen | 0.5 | 0.74 | 1.12 | 0.60 |
| | $kFe_{nd}$ | shape defining constant for iron | 0.33 $\times 10^{-3}$ | 0.49 $\times 10^{-3}$ | 0.74 $\times 10^{-3}$ | 0.40 $\times 10^{-3}$ |
| Diatom | $kN_d$ | shape defining constant for nitrogen | 0.75 | 1.12 | 1.68 | 0.91 |
| | $kSi_d$ | shape defining constant for silicon | 0.75 | 1.12 | 1.68 | 0.91 |
| | $kFe_d$ | shape defining constant for iron | 0.67 $\times 10^{-3}$ | 0.99 $\times 10^{-3}$ | 1.50 $\times 10^{-3}$ | 0.81 $\times 10^{-3}$ |

| Process/ Plankton type | Symbol | Meaning | Parameter value (mmol m$^{-3}$) | |
|---|---|---|---|---|
| **Grazing** ($G$) | | | Holling type III* | Holling type II |
| | | | ($G_1$) | ($G_2$) |
| | | | $g_m \frac{p_a Pa^2}{k_g^2 + p_a Pa^2 + p_b Pb^2}$ | $g_m \frac{p_a Pa^2}{k_g(p_a Pa + p_b Pb) + p_a Pa^2 + p_b Pb^2}$ |
| Microzooplankton | $k_{mi}$ | half saturation constant | 0.80 | 0.46 |
| | $pmi_{nd}$ | grazing preference for non-diatom | 0.75 | 0.75 |
| | $pmi_{det}$ | grazing preference for detritus | 0.25 | 0.25 |
| Mesozooplankton | $k_{me}$ | half saturation constant | 0.30 | 0.17 |
| | $pme_{nd}$ | grazing preference for non-diatom | 0.15 | 0.15 |
| | $pme_{det}$ | grazing preference for detritus | 0.15 | 0.15 |
| | $pme_d$ | grazing preference for diatoms | 0.35 | 0.35 |
| | $pme_{mi}$ | grazing preference for microzooplankton | 0.35 | 0.35 |

| Process/ Plankton type | Symbol | Meaning | Hyperbolic* | Linear | Quadratic | Sigmoidal |
|---|---|---|---|---|---|---|
| **Mortality** ($\rho, \xi$) | | | ($\rho_1, \xi_1$) | ($\rho_2, \xi_2$) | ($\rho_3, \xi_3$) | ($\rho_4, \xi_4$) |
| | | | $\mu \frac{P}{P+k_M} P$ | $\mu P$ | $\mu P^2$ | $\mu \frac{P^2}{P^2 + k_M^2} P$ |
| Non-diatom | $\mu_{nd}$ | maximum rate (day$^{-1}$) | 0.10 | 0.10 | 0.05 | 0.10 |
| | $k_{Mnd}$ | half saturation constant | 0.50 | - | - | 0.74 |
| Diatom | $\mu_d$ | maximum rate (day$^{-1}$) | 0.10 | 0.10 | 0.05 | 0.1 |
| | $k_{Md}$ | half saturation constant | 0.50 | - | - | 0.74 |
| Microzooplankton | $\mu_{mi}$ | maximum rate (day$^{-1}$) | 0.10 | 0.10 | 0.05 | 0.10 |
| | $k_{Mmi}$ | half saturation constant | 0.50 | - | - | 0.74 |
| Microzooplankton | $\mu_{mi}$ | maximum rate (day$^{-1}$) | 0.20 | 0.20 | 0.07 | 0.20 |
| | $k_{Mmi}$ | half saturation constant | 0.75 | - | - | 1.12 |

**Table 2.** Error statistics, 10-year mean, and NRR of chlorophyll (mg m$^{-3}$) and nitrogen (mmol m$^{-3}$) concentration at five stations for the default run, ensemble mean, ensemble median, and the ensemble range (ensemble maximum - ensemble minimum). These are calculated from surface to 200 m depth, starting from January 1998 to December 2007. Bias is (model output) – (in situ observation). Bold text indicate the smallest RMSE. At Station L4 error statistics and mean are taken from the surface and starts from January 1999 for chlorophyll and June 2000 for nitrogen. For station PAP, error statistics are taken from 2002-2004 since in situ data is only available during that time.

| Stations | | Nitrogen profile | | | | Chlorophyll profile | | | | Surface chlorophyll | | | | Integrated chlorophyll | | | |
|---|---|---|---|---|---|---|---|---|---|---|---|---|---|---|---|---|---|
| | | r | RMSE | Bias | Mean | r | RMSE | Bias | Mean | r | RMSE | Bias | Mean | r | RMSE | Bias | Mean |
| PAP | Ens mean | 0.23 | 3.26 | 0.61 | 6.59 | 0.42 | 0.32 | 0.06 | 0.48 | 0.45 | 0.51 | 0.22 | 0.66 | | | | |
| | | (±0.07) | (±2.57) | (±5.13) | (±5.24) | (±0.37) | (±0.73) | (±0.68) | (±0.75) | (±0.38) | (±0.73) | (±0.68) | (±0.76) | | | | |
| | Ens median | 0.23 | **3.16** | 0.54 | 6.38 | 0.49 | **0.29** | 0.003 | 0.42 | 0.54 | **0.46** | 0.15 | 0.60 | | | | |
| | Default run | 0.21 | 3.32 | -0.20 | 5.64 | 0.28 | 0.40 | 0.18 | 0.59 | 0.36 | 0.57 | 0.30 | 0.74 | | | | |
| | In situ | | | | 5.83 | | | | 0.42 | | | | 0.44 | | | | |
| | NRR | | 1.25 | | | | 1.20 | | | | 1.29 | | | | | | |
| ALOHA | Ens mean | 0.77 | **1.06** | 0.67 | 1.20 | 0.22 | **0.10** | -0.06 | 0.06 | 0.22 | **0.05** | -0.01 | 0.10 | 0.69 | **2.73** | -0.72 | 3.80 |
| | | (±0.03) | (±0.19) | (±0.39) | (±0.39) | (±0.49) | (±0.04) | (±0.11) | (±0.11) | (±0.47) | (±0.09) | (±0.13) | (±0.14) | (±0.60) | (±5.49) | (±7.09) | (±10) |
| | Ens median | 0.77 | **1.06** | 0.68 | 1.18 | 0.14 | 0.11 | -0.07 | 0.05 | 0.13 | **0.05** | -0.01 | 0.07 | 0.56 | 3.3 | -1.17 | 3.34 |
| | Default run | 0.77 | 1.09 | 0.61 | 1.10 | 0.28 | **0.10** | -0.03 | 0.09 | 0.27 | 0.07 | 0.03 | 0.11 | 0.70 | 4.71 | 1.25 | 5.77 |
| | In situ | | | | 0.50 | | | | 0.12 | | | | 0.08 | | | | 4.52 |
| | NRR | | 1.39 | | | | 1.29 | | | | 1.07 | | | | 1.01 | | |
| BATS | Ens mean | 0.56 | 1.39 | 1.16 | 1.77 | 0.19 | 0.33 | -0.12 | 0.05 | 0.22 | 0.33 | -0.12 | 0.05 | 0.39 | 52.13 | -19.39 | 6.18 |
| | | (±0.38) | (±0.84) | (±1.00) | (±1.01) | (±0.37) | (±0.05) | (±0.16) | (±0.16) | (±0.58) | (±0.15) | (±0.05) | (±0.15) | (±0.54) | (±9.40) | (±21) | (±14) |
| | Ens median | 0.55 | 1.39 | 1.16 | 1.77 | 0.11 | 0.33 | -0.12 | 0.05 | 0.06 | 0.34 | -0.12 | 0.05 | 0.27 | **23.30** | -17.71 | 4.51 |
| | Default run | 0.58 | **0.73** | 0.62 | 1.35 | 0.23 | **0.31** | -0.07 | 0.10 | 0.28 | **0.31** | -0.07 | 0.09 | 0.43 | 48.58 | -10.77 | 13.14 |
| | In situ | | | | 0.98 | | | | 0.17 | | | | 0.15 | | | | 23.90 |
| | NRR | | 1.38 | | | | 1.39 | | | | 1.40 | | | | 1.40 | | |
| Cariaco | Ens mean | 0.78 | **2.97** | 0.61 | 5.39 | 0.29 | **0.83** | -0.02 | 0.49 | 0.13 | **1.23** | 0.02 | 0.77 | 0.41 | **17.73** | -1.05 | 11.47 |
| | | (±0.08) | (±0.49) | (±2.54) | (±2.54) | (±0.34) | (±0.42) | (±0.93) | (±0.93) | (±0.22) | (±0.33) | (±1.90) | (±0.57) | (±0.40) | (±7.90) | (±17) | (±17) |
| | Ens median | 0.76 | 3.24 | 0.51 | 5.29 | 0.20 | 0.88 | -0.18 | 0.32 | 0.072 | 1.29 | -0.29 | 0.46 | 0.29 | 19.46 | -5.51 | 7.00 |
| | Default run | 0.76 | 3.29 | 0.59 | 5.37 | 0.22 | 0.87 | -0.09 | 0.42 | 0.11 | 1.27 | -0.18 | 0.57 | 0.34 | 18.71 | -3.86 | 8.65 |
| | In situ | | | | 4.78 | | | | 0.51 | | | | 0.76 | | | | 12.52 |
| | NRR | | 1.25 | | | | 1.19 | | | | 1.21 | | | | 1.17 | | |
| L4 | Ens mean | 0.70 | 2.94 | 1.56 | 4.52 | | | | | 0.25 | 1.05 | 0.42 | 1.76 | | | | |
| | | (±0.14) | (±2.13) | (±4.06) | (±4.06) | | | | | (±0.33) | (±1.67) | (±2.61) | (±2.61) | | | | |
| | Ens median | 0.68 | 3.10 | 1.73 | 4.69 | | | | | 0.21 | **1.02** | 0.27 | 1.61 | | | | |
| | Default run | 0.52 | **2.67** | 1.12 | 4.08 | | | | | 0.31 | 1.13 | 0.83 | 2.17 | | | | |
| | In situ | | | | 2.96 | | | | | | | | 1.34 | | | | |
| | NRR | | 1.21 | | | | | | | | 1.31 | | | | | | |



**Table 3.** Surface annual mean and phytoplankton phenology from in situ, ensemble mean, median, and default run.

| Stations | | Annual Mean (mg m$^{-3}$) | Initiation Time | Bloom (mg m$^{-3}$) | Peak Height (mg m$^{-3}$) | Amplitude (mg m$^{-3}$) | Duration | Termination |
|---|---|---|---|---|---|---|---|---|
| PAP | Ens mean | 0.61 | 01 Apr | 07 May | 2.07 | 0.96 | 95 | 26 Jul |
| | Range | ±0.70 | ±51 | ±45 | ±2.98 | ±1.63 | ±99 | ±124 |
| | NRR | 1.26 | 1.14 | 1.31 | 1.08 | 1.09 | 1.42 | 1.60 |
| | Ens med | 0.55 | 12 Apr | 15 May | 2.03 | 0.95 | 87 | 22 Jul |
| | Default run | 0.71 | 03 Apr | 05 May | 2.1 | 0.96 | 99 | 21 Aug |
| | In situ | 0.44 | 20 Apr | 03 Jun | 1.52 | 0.44 | 95 | 24 Jul |
| ALOHA | Ens mean | 0.07 | 21 Mar | 21 Apr | 0.14 | 0.047 | 62 | 15 Aug |
| | Range | ±0.13 | ±89 | ±119 | ±0.28 | ±0.11 | ±95 | ±119 |
| | NRR | 0.84 | 1.35 | 1.29 | 0.97 | 1.19 | 1.56 | 1.28 |
| | Ens med | 0.063 | 26 Mar | 02 May | 0.14 | 0.05 | 85 | 24 Aug |
| | Default run | 0.10 | 14 Mar | 18 Apr | 0.25 | 0.096 | 66 | 10 Aug |
| | In situ | 0.084 | 08 May | 26 May | 0.14 | 0.048 | 47 | 23 Jun |
| BATS | Ens mean | 0.047 | 02 Mar | 12 Apr | 0.1 | 0.043 | 89 | 06 Jul |
| | Range | ±1.53 | ±187 | ±174 | ±0.42 | ±0.19 | ±116 | ±198 |
| | NRR | 1.40 | 1.18 | 1.17 | 1.42 | 1.42 | 1.08 | 1.20 |
| | Ens med | 0.038 | 28 Feb | 06 Apr | 0.08 | 0.033 | 95 | 02 Aug |
| | Default run | 0.091 | 06 Mar | 25 Apr | 0.29 | 0.13 | 65 | 19 Jun |
| | In situ | 0.17 | 25 Feb | 29 Mar | 0.58 | 0.27 | 93 | 28 May |
| Cariaco | Ens mean | 0.61 | 20 May | 22 Jul | 1.09 | 0.38 | 133 | 30 Sep |
| | Range | ±0.53 | ±101 | ±66 | ±2.61 | ±0.86 | ±63 | ±61 |
| | NRR | 0.78 | 1.48 | 1.40 | 1.39 | 1.42 | 1.88 | 1.55 |
| | Ens med | 0.37 | 22 May | 14 Jul | 0.83 | 0.34 | 110 | 06 Sep |
| | Default run | 0.46 | 21 May | 22 Jul | 0.98 | 0.39 | 122 | 19 Sep |
| | In situ | 0.61 | 16 Mar | 21 Apr | 2.39 | 1.15 | 76 | 01 Jun |
| L4 | Ens mean | 1.65 | 13 May | 06 Jun | 3.25 | 1.13 | 64 | 17 Aug |
| | Range | 2.76 | ±100 | ±82 | ±3.12 | ±1.50 | ±78 | ±167 |
| | NRR | 1.00 | 1.49 | 1.42 | 1.32 | 1.48 | 1.22 | 1.19 |
| | Ens med | 1.49 | 18 May | 07 Jun | 3.09 | 1.13 | 70 | 18 Sep |
| | Default run | 2.03 | 19 Apr | 08 Jun | 3.73 | 1.3 | 94 | 11 Aug |
| | In situ | 1.20 | 09 Mar | 11 Apr | 3.58 | 1.64 | 80 | 28 May |





**: Figure Captions**

Figure 1. Nearly identical curves which describes resource uptake (a), zooplankton grazing (b), and phytoplankton mortality (c). Figure (a) shows four uptake functions, which have been optimised to the default uptake function, monod. Figure (b) shows two grazing functional forms, the holling type III and type II functions. Four phytoplankton mortality functions are shown on figure (c), whereby hyperbolic is the default function. The optimisation method is describe in section 2.1, 2.2, and 2.3. Table 1 describes the function's equations and parameters.

Figure 2. SeaWIFs-derived mean 1998 chlorophyll-$a$ (mg m$^{-3}$) overlain with the 5 oceanographic stations time series site (Red dots). These stations are located in different oceanic regions: oligotrophic (ALOHA and BATS), coastal (L4 and Cariaco), and abyssal plain (PAP).

Figure 3. Chlorophyll and nitrogen profiles from ensemble mean ((a) and (d) respectively), in situ observations ((c) and (f) for chlorophyll and nitrogen respectively), and 75$^{th}$ and 25$^{th}$ quartile range of concentrations at each depth ((b) for chlorophyll and (e) for nitrogen) at station PAP. The range are obtained by averaging the concentrations from all ensemble members for 10 years at each depths. Black dots in the second column show the mean concentration of the ensemble mean over the time series (from January 1998-December 2007). White solid line in (a) shows mixed layer depth.

Figure 4. Inter-annual mean of surface chlorophyll from all the study sites ((a)-(e)) and the 10-year annual mean (g), all measured in mg m$^{-3}$. The boxplots show the ensemble annual means. Blue cross is the in situ observation, red open circle, black dot, and blue stars are the ensemble mean, median, and the default run respectively. The blue box is the 75$^{th}$ (top) and 25$^{th}$ (bottom) quartiles. Red line is the median. The whiskers are the ensemble minimum and maximum mean of surface chlorophyll. Annual mean values and NRR are described in Table 3.

Figure 5. 10-year monthly mean surface chlorophyll from all the study sites ((a)-(e)), showing the seasonal dynamics of surface chlorophyll (mg m$^{-3}$). The boxplots show the ensemble annual means. Blue cross is the in situ observation, red open circle, black dot, and blue stars are the ensemble mean, median, and the default run respectively. The blue box is the 75$^{th}$ (top) and 25$^{th}$ (bottom) quartiles. The red line is the median. The whiskers are the ensemble minimum and maximum mean of surface chlorophyll. In station PAP, in situ data for December is not available due to low light and high cloud cover.

Figure 6. Time series (from January 1998-December 2007) of ensemble mean and in situ, and range of chlorophyll and nitrogen concentrations at oligotrophic stations. Station ALOHA is shown on (a)-(f) and BATS is shown on (g)-(l). White solid line in



(b) and (g) represents mixed layer depth. (b), (d), (h), and (j) are the 75[th] and 25[th] percentile range of chlorophyll ((b) for ALOHA and (h) for BATS) and nitrogen ((d) for ALOHA and (j) BATS) over the depth. The range is obtained by averaging the chlorophyll and nitrogen concentrations of each ensemble members over the time series at each depth. Black dots in (b), (d), (h), and (j) are the mean of the ensemble. Ensemble mean chlorophyll profiles (shown on (a) and (g)) and nitrogen ((e) and (k)) are obtained from all of the ensemble members. *In situ* chlorophyll are shown in (c) and (i), and nitrogen are shown in (g) and (l), for ALOHA and BATS respectively.

Figure 7. Chlorophyll profile 10-year means ((a)-(d)) and its RMSEs ((e)-(f)) at four oceanographic station from all of the ensemble members. Station L4 is not included as chlorophyll data is only taken at the surface. These are arranged by the lowest chlorophyll (top left) mean to the highest (bottom right), depending on the oceanographic regions.

Figure 8. 10-year mean and RMSE of surface chlorophyll (mg m$^{-3}$), and nitrogen (mmol m$^{-3}$), at five stations from all ensemble members. The first panel ((a)-(e)) shows surface chlorophyll mean, and the third panel ((k)-(o)) shows nitrogen mean. RMSEs are shown on the second panel ((f)-(j)) and fourth ((p)-(t)) for surface chlorophyll, and nitrogen respectively. Concentrations and RMSEs are arranged by the lowest chlorophyll (top left) mean to the highest (bottom right), depending on the oceanographic regions. For station PAP, the sequence is sorted based on coastal station.

Figure 9. Time series of chlorophyll and nitrogen profile of ensemble mean, their range, and in situ concentrations at the coastal stations Cariaco (a-f) and L4 (g-h) from January 1998-December 2007. (a) and (d) show chlorophyll and nitrogen ensemble mean at Cariaco respectively. White solid line in (a) is the mixed layer depth. (b) and (e) shows the 75[th] and 25[th] percentile of chlorophyll and nitrogen concentrations at each depth. The black dots are the mean of the ensemble. These range are obtained form the 10-year mean concentrations at each depth. Since in situ chlorophyll and nitrogen were taken at the surface in station L4, only surface time series were shown in (g-h). The grey shades on chlorophyll, shown in (g), and nitrogen, shown in (h) time series show 75[th] and 25[th] percentile of the range. Blue and red dots are in situ concentrations for chlorophyll and nitrogen respectively.

Figure 10. Phytoplankton phenology metrics at the five stations. Blue cross is the in situ, red, black, and blue dots are the ensemble mean, median, and the default run respectively. The timings and concentrations are averaged annually from January 1998 to December 2007.

Figure A1. Determining phenology using a combination of threshold method and curve fit at station L4, here the initiation is when the fitted curve is above 50% of the maximum peak, however the termination is on the first valley.



**: Figures**

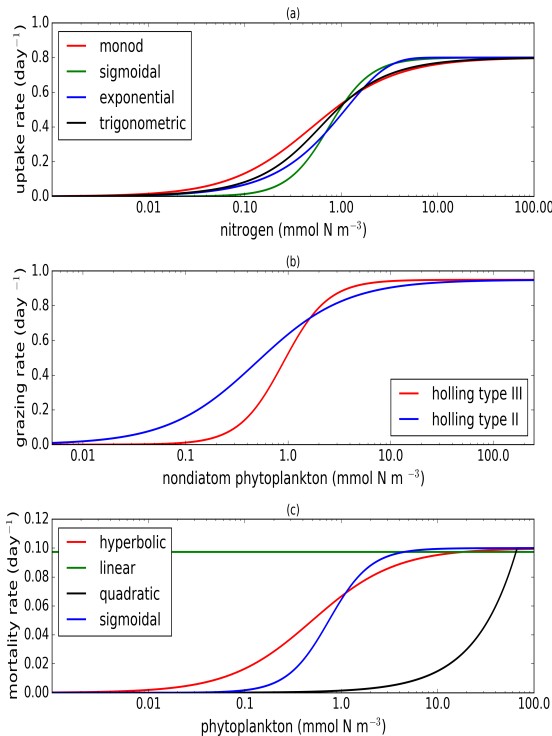

**Figure 1.** Nearly identical curves which describes resource uptake (a), zooplankton grazing (b), and phytoplankton mortality (c). Figure (a) shows four uptake functions, which have been optimised to the default uptake function, monod. Figure (b) shows two grazing functional forms, the holling type III and type II functions. Four phytoplankton mortality functions are shown on figure (c), whereby hyperbolic is the default function. The optimisation method is describe in section 2.1, 2.2, and 2.3. Table 1 describes the function's equations and parameters.





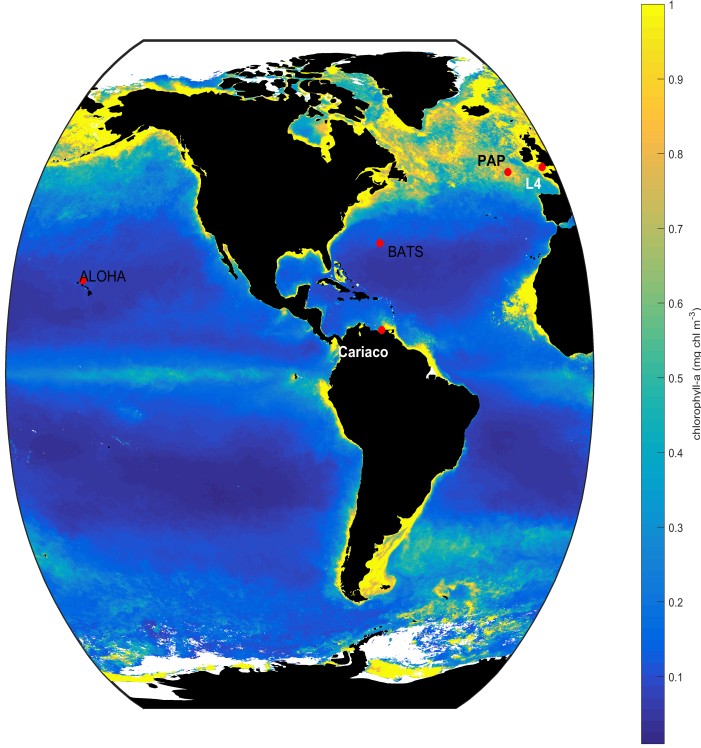

**Figure 2.** SeaWIFs-derived mean 1998 chlorophyll-$a$ (mg m$^{-3}$) overlain with the 5 oceanographic stations time series site (Red dots). These stations are located in different oceanic regions: oligotrophic (ALOHA and BATS), coastal (L4 and Cariaco), and abyssal plain (PAP).





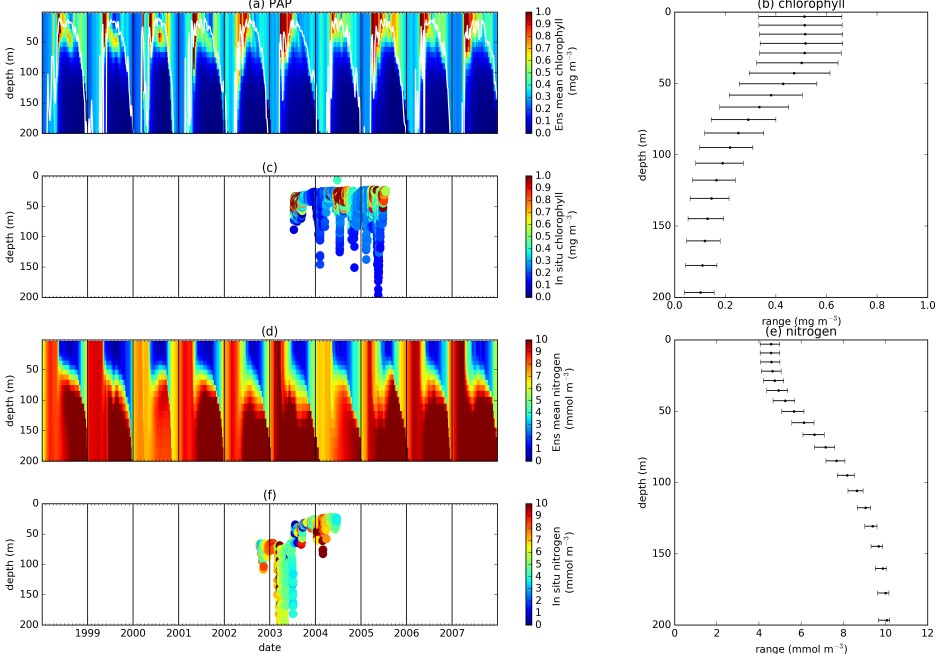

**Figure 3.** Chlorophyll and nitrogen profiles from ensemble mean ((a) and (d) respectively), in situ observations ((c) and (f) for chlorophyll and nitrogen respectively), and 75[th] and 25[th] quartile range of concentrations at each depth ((b) for chlorophyll and (e) for nitrogen) at station PAP. The range are obtained by averaging the concentrations from all ensemble members for 10 years at each depths. Black dots in the second column show the mean concentration of the ensemble mean over the time series (from January 1998-December 2007). White solid line in (a) shows mixed layer depth.



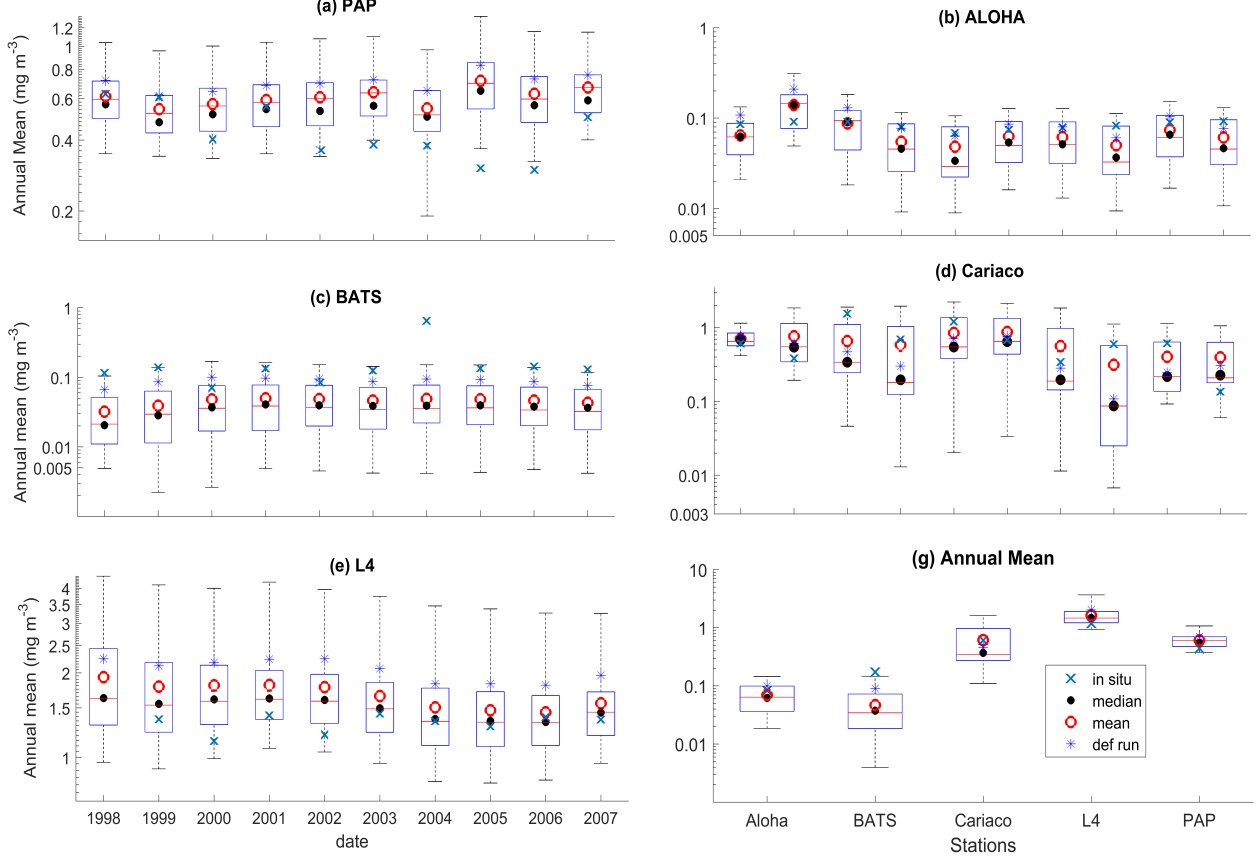

**Figure 4.** Inter-annual mean of surface chlorophyll from all the study sites ((a)-(e)) and the 10-year annual mean (g), all measured in mg m$^{-3}$. The boxplots show the ensemble annual means. Blue cross is the in situ observation, red open circle, black dot, and blue stars are the ensemble mean, median, and the default run respectively. The blue box is the 75$^{th}$ (top) and 25$^{th}$ (bottom) quartiles. Red line is the median. The whiskers are the ensemble minimum and maximum mean of surface chlorophyll. Annual mean values and NRR are described in Table 3.



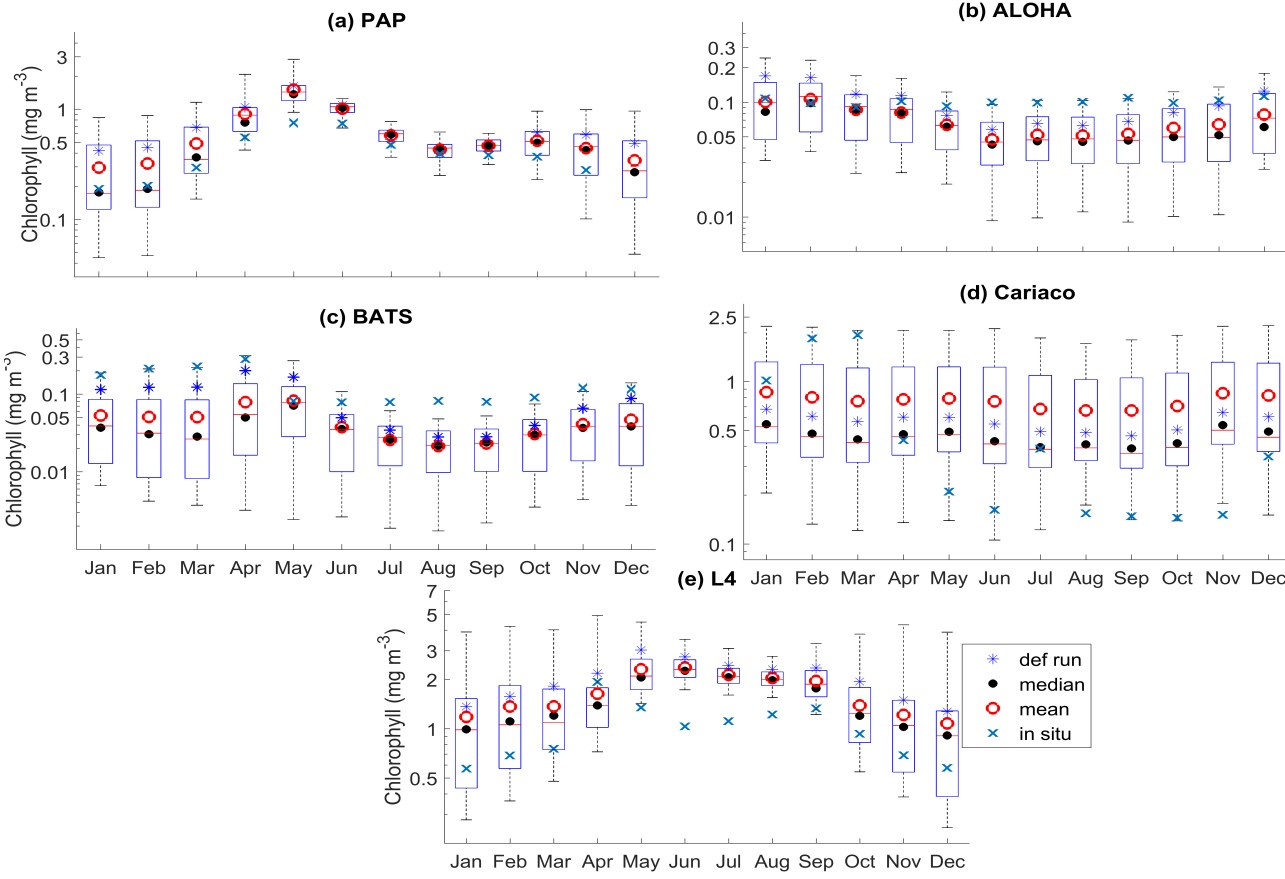

**Figure 5.** 10-year monthly mean surface chlorophyll from all the study sites ((a)-(e)), showing the seasonal dynamics of surface chlorophyll (mg m$^{-3}$). The boxplots show the ensemble annual means. Blue cross is the in situ observation, red open circle, black dot, and blue stars are the ensemble mean, median, and the default run respectively. The blue box is the 75$^{th}$ (top) and 25$^{th}$ (bottom) quartiles. The red line is the median. The whiskers are the ensemble minimum and maximum mean of surface chlorophyll. In station PAP, in situ data for December is not available due to low light and high cloud cover.





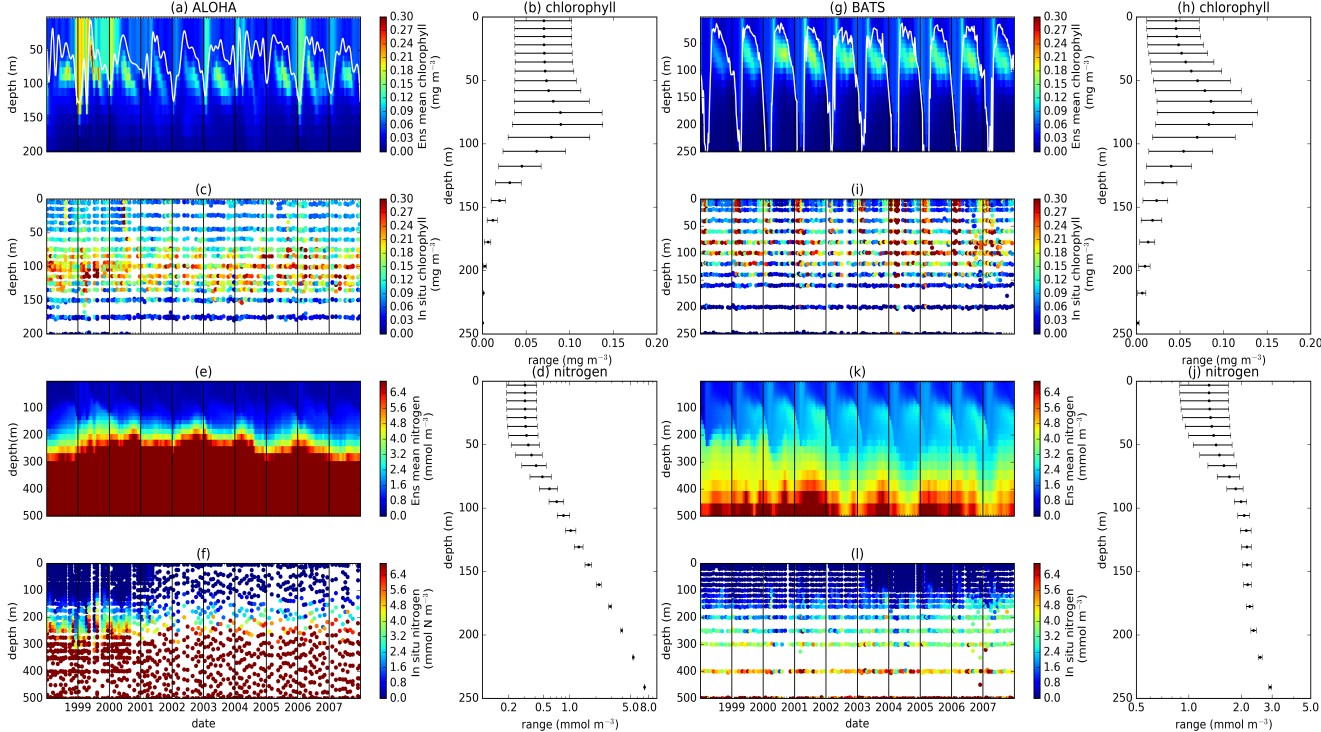

**Figure 6.** Time series (from January 1998-December 2007) of ensemble mean and in situ, and range of chlorophyll and nitrogen concentrations at oligotrophic stations. Station ALOHA is shown on (a)-(f) and BATS is shown on (g)-(l). White solid line in (b) and (g) represents mixed layer depth. (b), (d), (h), and (j) are the 75$^{th}$ and 25$^{th}$ percentile range of chlorophyll ((b) for ALOHA and (h) for BATS) and nitrogen ((d) for ALOHA and (j) BATS) over the depth. The range is obtained by averaging the chlorophyll and nitrogen concentrations of each ensemble members over the time series at each depth. Black dots in (b), (d), (h), and (j) are the mean of the ensemble. Ensemble mean chlorophyll profiles (shown on (a) and (g)) and nitrogen ((e) and (k)) are obtained from all of the ensemble members. *In situ* chlorophyll are shown in (c) and (i), and nitrogen are shown in (g) and (l), for ALOHA and BATS respectively.



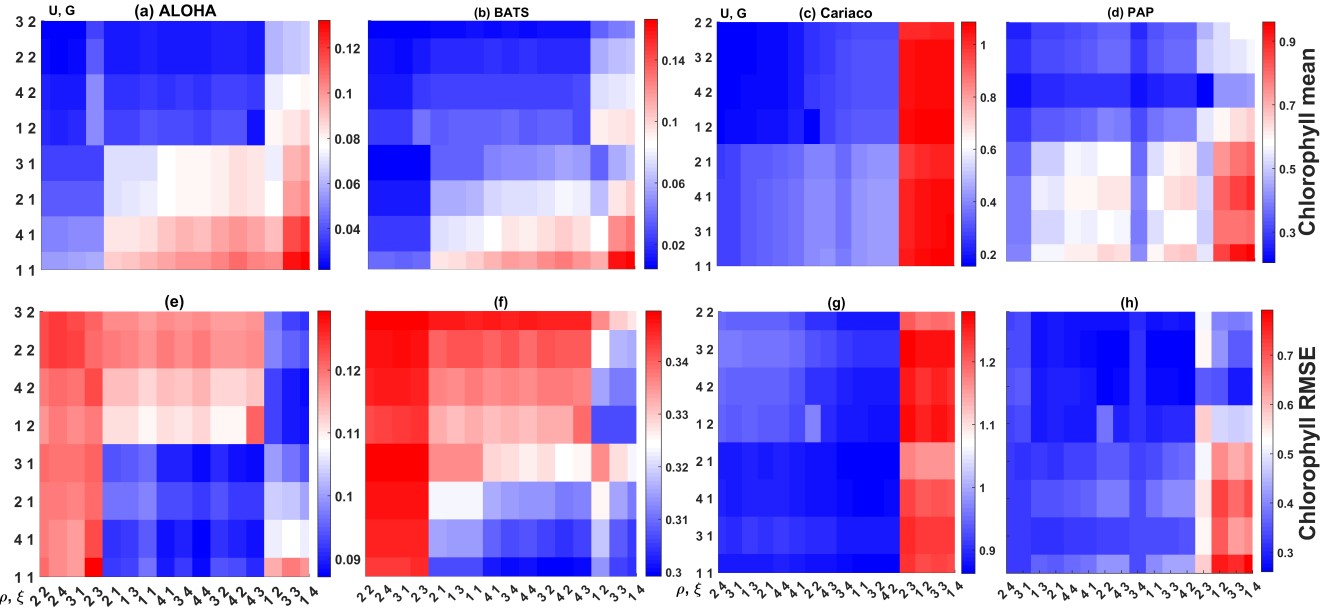

**Figure 7.** Chlorophyll profile 10-year means ((a)-(d)) and its RMSEs ((e)-(f)) at four oceanographic station from all of the ensemble members. Station L4 is not included as chlorophyll data is only taken at the surface. These are arranged by the lowest chlorophyll (top left) mean to the highest (bottom right), depending on the oceanographic regions.





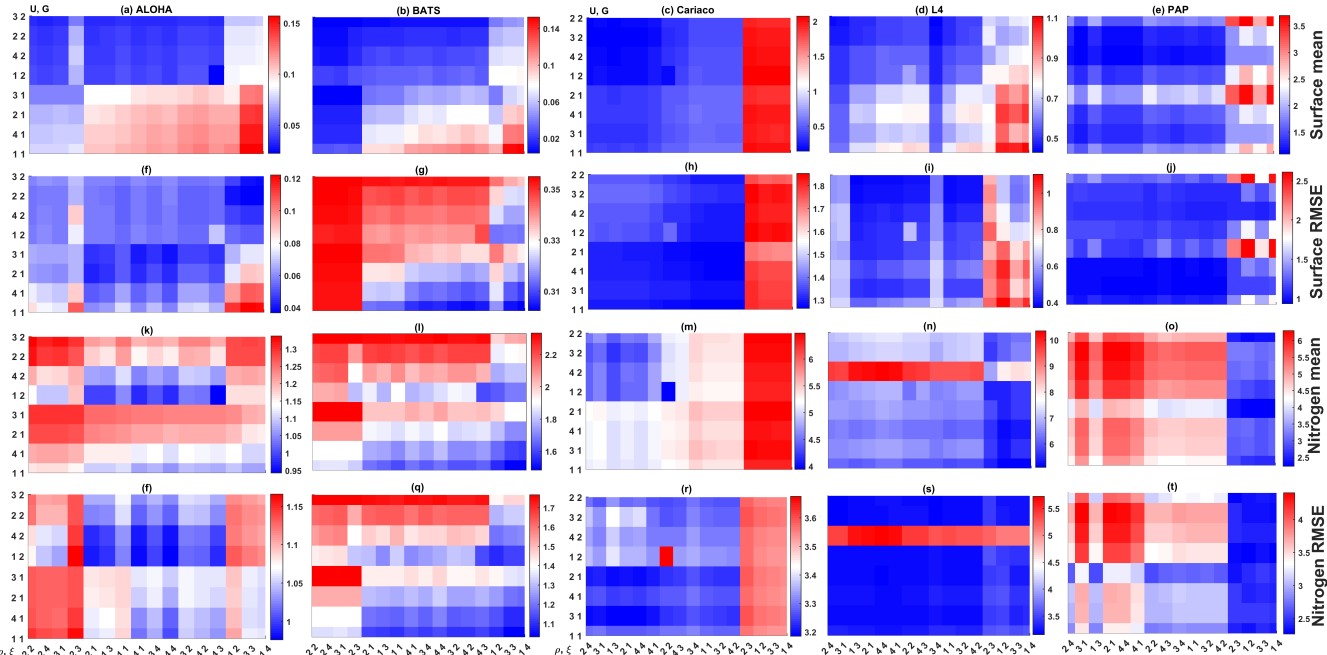

**Figure 8.** 10-year mean and RMSE of surface chlorophyll (mg m$^{-3}$), and nitrogen (mmol m$^{-3}$), at five stations from all ensemble members. The first panel ((a)-(e)) shows surface chlorophyll mean, and the third panel ((k)-(o)) shows nitrogen mean. RMSEs are shown on the second panel ((f)-(j)) and fourth ((p)-(t)) for surface chlorophyll, and nitrogen respectively. Concentrations and RMSEs are arranged by the lowest chlorophyll (top left) mean to the highest (bottom right), depending on the oceanographic regions. For station PAP, the sequence is sorted based on coastal station.



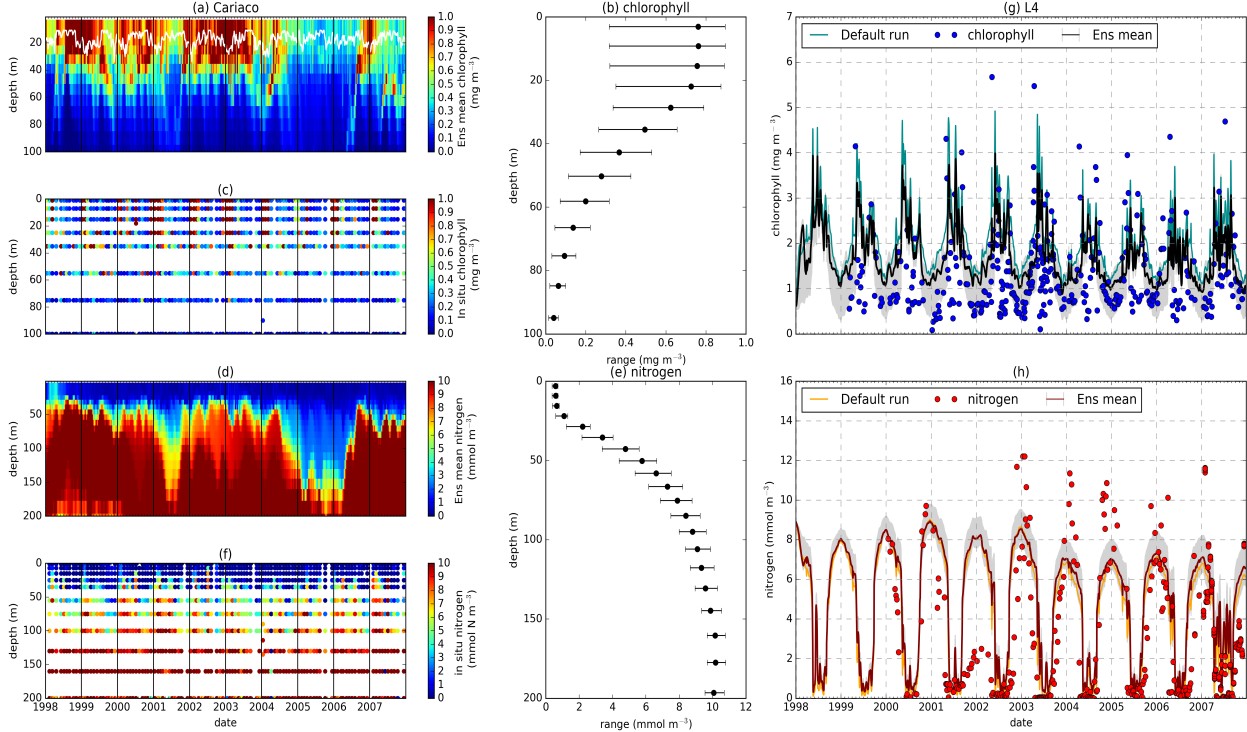

**Figure 9.** Time series of chlorophyll and nitrogen profile of ensemble mean, their range, and in situ concentrations at the coastal stations Cariaco (a-f) and L4 (g-h) from January 1998-December 2007. (a) and (d) show chlorophyll and nitrogen ensemble mean at Cariaco respectively. White solid line in (a) is the mixed layer depth. (b) and (e) shows the $75^{th}$ and $25^{th}$ percentile of chlorophyll and nitrogen concentrations at each depth. The black dots are the mean of the ensemble. These range are obtained form the 10-year mean concentrations at each depth. Since in situ chlorophyll and nitrogen were taken at the surface in station L4, only surface time series were shown in (g-h). The grey shades on chlorophyll, shown in (g), and nitrogen, shown in (h) time series show $75^{th}$ and $25^{th}$ percentile of the range. Blue and red dots are in situ concentrations for chlorophyll and nitrogen respectively.



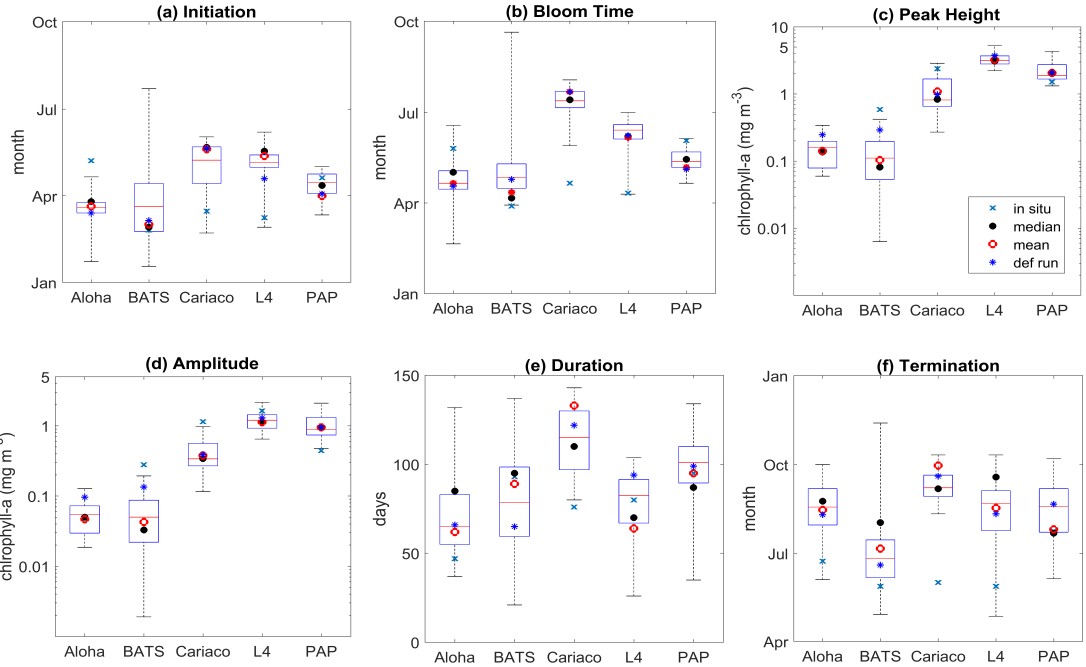

**Figure 10.** Phytoplankton phenology metrics at the five stations. Blue cross is the in situ, red, black, and blue dots are the ensemble mean, median, and the default run respectively. The timings and concentrations are averaged annually from January 1998 to December 2007.





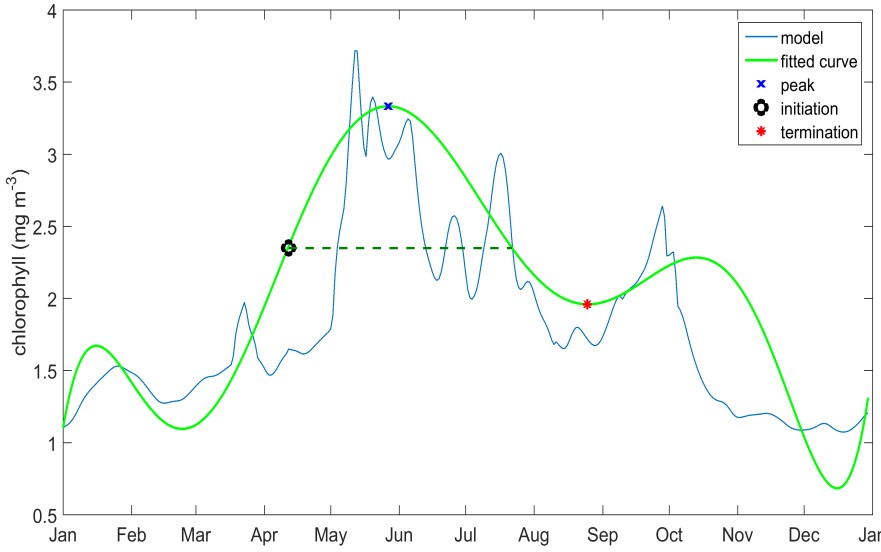

**Figure A1.** Determining phenology using a combination of threshold method and curve fit at station L4, here the initiation is when the fitted curve is above 50% of the maximum peak, however the termination is on the first valley.