# Peer review of "A perturbed biogeochemistry model ensemble evaluated against in situ and satellite observations"

_Biogeosciences, 2018_

## Referee Comment (RC1) · Anonymous Referee #1 · 19 Apr 2018

This manuscript examines the structural sensitivity (to equation formulations) of a marine ecosystem model by using ensemble analysis across various ocean sites. The topic is highly relevant and I like what the authors are trying to achieve. The protocol is well thought out and this study can potentially make a useful contribution to the literature. There are however some significant weakness in the manuscript which need to be addressed.

Major comments:

(1) My major criticism is that, in general, the model appears to show major discrepancies with the data, undermining the credibility of the whole exercise, including the

conclusions. To be effective, the default model run should show reasonable correspondence with the data but, in several instances, it appears not to do so. Just because the MEDUSA model is already parameterised and published in this regard does not save the situation here because the work involved changing the parameterisations of sinking, maximum and grazing rates (that's rather a lot; page 6, line 7). For example, I am not convinced about the new parameterisation of sinking, namely a sinking rate of 0.1 m d-1 (page 6, line 17) which seems much too low. At PAP, the blues stars (default run) are way too high relative to the blue crosses (observations) indicating a major discrepancy for chlorophyll (Figure 4). The average chlorophyll values for the oligotrophic stations look ok, but the depth plots do not look good at all in this respect (the deep chlorophyll maxima look poorly reproduced; Figure 6). I need more convincing that the model is credible at these sites. There also seem to be large discrepancies for L4 (Figure 4). The modelled vertical concentrations of nitrate at PAP look way too high compared to the data (Figure 3). Why have box and whisker plots not been produced for nitrate, comparing model and data? And why does the appendix (supplementary material) focus only on chlorophyll, and not nitrate? Overall, I am left in doubt as to whether the model, as parameterised for the default run, is credible. The authors could help the situation by looking at some other metrics, if only for the default run. For example, what is predicted primary production at the different sites and how does this compare with data (even just comparing annual average would be highly useful)?

(2) The ensemble run at each station is initialised using in situ measurements (page 6, line 31). What is needed is a stable initial condition, which will not be potentially vulnerable to initial condition instabilities. So surely what is needed is to run the first year over and over (do a spin-up) until a repeating cycle is reached, from which the run through the various years can then be undertaken.

(3) A major conclusion of the work is (page 15, line 29) that "small perturbations in model structure can produce a wide range of results". This is a very significant conclusion and I think the authors can justifiably make it. For the most part, however, the

results as shown in the Figures don't show this directly, because they involve various parameterisations acting simultaneously. There is plenty of text in the Results section to support their contention, focusing on individual parameters. I wonder if this conclusion could be better represented in the graphical representation of the results.

(4) The Introduction is generally well written, introducing the topic of model complexity nicely. The Discussion should mirror the Introduction, saying what the current study has said in context of the wider picture. Instead, the Discussion is mostly just an extended re-hash of the Results and does little to address the big picture. For example, what do the authors conclude about model sensitivity in context of complexity science and the onward drive to produce model of ever increasing complexity? A much bigger play could be made on the need to do sensitivity analysis, an important conclusion with which I fully agree. Articulate the benefits of the ensemble analysis over previous studies that have focused more narrowly on particular parameterisations. Etc. There is plenty of scope and I would say the Discussion section needs a significant overhaul in this regard. It needs re-emphasis; a few extra lines of text will not do.

Other comments:

(1) The authors articulate two types of uncertainty (page 2, line 26): "parametric, associated with the choice of parameter values; and structural, which relates to the underlying model equations". Structural uncertainty can also refer to the structure of the model itself (number of compartments, linkages, etc). This should be mentioned, stating that the authors are only looking at structural uncertainty to do with equation formulations.

(2) On page 9, line 12, there is "A selection of ensemble results are presented". A selection? On what basis?

(3) Some of the text associated with the Figures is microscopically small.

(4) Be sure to cite Le Quere, not Quere without "Le".

---

## Referee Comment (RC2) · Anonymous Referee #2 · 24 Apr 2018

The authors present a study in which an ensemble of biogeochemical 1D ocean models is generated by using different functional forms for key model pathways. The goal is to highlight the structural uncertainty in biogeochemical models as well as the benefits of the ensemble in fitting various datasets.

general comments:

The manuscript attempts to show two aspects: (1) there is a high level of structural uncertainty in biogeochemical models and (2) the uncertainty can be exploited to better fit a range of different observations. In my opinion, the authors succeed in providing evidence for first aspect but I have doubts about the second: all comparisons of the

ensemble are based on a default run that does not seem to perform very well. Other studies have shown that 1D models with the same parameter values do not perform well across multiple locations but here the same parameter values appear to be used across all stations. Have the parameters of the default run been optimized to fit the datasets used in this study? The results of the default run can have knock-on effects on the ensemble: in multiple parts of the manuscript the authors note that when there is a large bias between the model (ensemble) and the observation, that the ensemble spread is too low when really other model aspects may be to blame for the bias. In other words, problems with the parametrization, the physical model, or the 1D nature of the model cannot be explained by structural uncertainty in the biogeochemical model.

When looking at Figure 1, I noticed that the linear function in (c) provides a bad fit to the other functions and that all functions are shown on a log scale. I am wondering if a log-transformation has also been used in the function fitting exercise in Sections 2.1-2.3? If not, I would recommend that this should at least be tried as the procedure could otherwise overemphasize the fit at high tracer concentrations which may explain the slope of the linear function.

Parts of the manuscript are difficult to understand and especially the introduction contains a large number of long run-on sentences. I have highlighted some of these instances in my specific comments below but I would recommend that the authors read through the manuscript again carefully and revise some sections for clarity.

specific comments:

abstract:

l1: "mathematical structure": What exactly does this refer to? The model formulation? I would suggest to rephrase or an improved explanation.

l3: "intermediately complex BGC model" -> "BGC model of intermediate complexity"

l9: "using phytoplankton phenology (...) and other statistical measures": phytoplankton

phenology is not a statistical measure.

l11: Is this the range found in the ensemble (as opposed to e.g. different coastal stations)? Please make this explicit.

l14: "the errors are mostly reduced": This is not clear: model misfit with respect to the in situ obs is smaller for the ensemble mean/median than the model with standard parameters? I suggest to rephrase.

l15: Here a narrow spread is reported, a few lines above a "large" spread was described.

p2:

l7: This reads like the forecasting systems are having an impact on ocean biogeochemistry. The climate change aspect of the sentence reads like a repeat of sentence in line 2. Please revise for clarity.

l12: Even NPZ models represent "several" processes. Please be more precise.

l16: There can be spatial variability without iron!

l29: "only small perturbations are usually produced even with large variations in parameter values" This is a very strong statement and very much depends on what a "large variation" entails. Perhaps weaken the statement and just make the point that structural uncertainty is often larger than parametric?

p3:

l13: "linear density-dependent mortality produces the most significant differences when applied to diatoms": What exactly does this mean? Please revise.

l18: "However, not all processes give significantly different model outputs." The next sentence seems to imply that the differences maybe due to very similar inputs, can this effect thus really be attributed to the process?

l22: "However, it is still unclear what will happened if formulations of all the core processes [...] are perturbed together." The preceding sentence is very general and I would say it is quite clear that the perturbations of all core processes would also "give rise to different effects". I would suggest to rephrase.

p4:

l3: "using all possible functional combinations": Given that there can be an infinite amount of different functional forms, I would suggest to rephrase this sentence. (Later on it becomes clear that only a few functional forms are considered.)

l22: Mention right away that Table 1 contains the equations for all functions.

l29: Mention that "T" is temperature here.

l32: "the default": Is this $U_1$?

p5:

l4: "The small microzooplankton": this makes it sound like there are small and large microzooplankton. Use something like "The small zooplankton category consists of microzooplankton..."

l5: Is "non-diatoms" referring to the "smaller phytoplankton" in the previous sentence?

l8: This is the third time Michaelis-Menten and Holling type II are mentioned together.

l9: "II" -> "III"

l9: Why say "hereafter $G_1/G_2$" when "Holling type II/III" is used throughout the text?

l19: Was the shape of the curves adjusted again? If so, how?

l29: What is a "distinct trend" here?

l30: It is not clear to me how the linear function was made to match the others. Figure 1(c) seems to suggest something went wrong. Or are large values here simply

overemphasized in the fit?

p6:

l31: How long is the spin-up period for the runs?

p7:

l9: Why this lengthy comment about physical data assimilation? Is the capping done to remove the perceived negative influence of the physical data assimilation? What about rapid shifts in mixed layer depth which is also an input of the model, may also be affected by physical data assimilation and may also drastically change nutrient concentrations in the model. It is also not quite clear how the mixed layer depth influences the 1D model.

l26: It would be good to mention these locations the first time the stations are introduced.

Sec 2.5.2: Here the description is confusing, it goes from initial conditions to validation data, back to initial conditions and then to validation data.

p8:

l8: "one of MarMOT's test stations" What exactly is this test station?

p9:

l13: "These have been done at the five oceanographic stations which can be classified into three regional types:" This has been mentioned before.

l21: Mention PAP.

p11:

l4: How well does NRR work with a significant bias?

l10: "these members use functional combinations ..." The notation for the combinations
is not clear here

Table 1: It does not make sense to call \mu's the maximum rates here.

Fig 1: Use "U_1" etc. here.

Fig 7: A better description of the x and y axes are needed. Why do b,d,f and h have no y-axis? Use the same color scale across all stations. Same comment applies to Fig. 8 where the font becomes too small.

---

## Referee Comment (RC3) · Anonymous Referee #3 · 14 May 2018

This study investigates the response of a 1D version of the UK MEDUSA model to variations in the structure of equations for phytoplankton growth, zooplankton grazing and mortality of both planktonic groups. The model's response is tested at 5 different stations of varying seasonal cycle and nutrient supply.

I appreciate the attempt to investigate the sensitivity of biogeochemical models that will be, at a later stage, used for projections of climate change and other applied tasks, and I think the setup and design of the study are comprehensive, thorough, and suited to address this. However, I have some serious concerns with the way the results of the study are presented, and discussed.

[Figure]

Firstly, in the introduction (page three, line 29) the authors state that "It has been demonstrated in conventional sensitivity analyses that only small perturbations are usually produced even with large variations in parameter values, but much larger changes in system dynamics can result from changes in the structural process formulations". I am not quite sure what "conventional" means, but I do think that this statement is misleading, as it neglects previous works that indicate a large sensitivity of marine biogeochemical models to their parameters, when compared to structural sensitivity. These studies have been carried out at a local scale, across different oceanic regimes, or in 3D (see, e.g., Friedrichs et al., 2007, Jour. Geophys. Res., 112, C08001, doi:10.1029/2006JC003852; Ward et al., 2013, Prog. Oceanog.116,49–65, or Kriest et al., 2012, Glob. Biogeochem. Cyc. 26, GB2029, doi:10.1029/2011GB004072, to name just a few examples). Some of them even address the role of different functional forms, or have been applied to the BATS site (e.g., study by Ward et al., 2013). They may be helpful for presenting and discussing this current work in a wider context. Thus, more exploration about what has been found for marine biogeochemical models and their structural and parametric uncertainty can help to improve the discussion, which is currently somehow repetitive, lacks a critical discussion of the results, and how they might relate to other uncertainties (structural, parametric, physical, ...).

Secondly, I miss some discussion about the way the different functional forms have been made "equivalent to each other." (p4 line 17). As it seems, the parameters of the different equations (e.g., half saturation-constants) were fitted against the default function "so that the overall shapes are as similar as possible." (p 4, line 19), by "minimising the sum squared difference between the default and other uptake forms" (line 32ff). Obviously, when looking at Fig 1, this happened across a very wide range of potential nutrient or chlorophyll (in case of zooplankton grazing) concentrations. The upper limits are far outside the range of values for most stations simulated in this study (up to 100 uM nitrate or phytoplankton N will likely never be found at BATS or ALOHA). Thus, it seems that the different functional forms were homogenised for a range that, at many stations, is outside the expected and/or observed range. On the other hand, the

functions deviate most strongly when nutrients or phytoplankton are scarce (Fig 1a and 1b), and more representative for the simulated regimes. What would have happened, if the test functions (e.g., sigmoidal or Holling III) were made equivalent to the default functions at lower substrate levels, representative for more oligotrophic regimes? Could it be that the effects of switching to alternative forms becomes less important? Again, the paper to my opinion would benefit a lot from a more critical discussion.

Thirdly, as recommended by the second referee, I suggest that the authors read through the manuscript again carefully, revise some sections for clarity, and correct spelling and grammar.

The results section already contains a lot of detail, which is partly repeated in the discussion. I would suggest to to shorten and streamline the presentation of results, highlighting those that are common among stations (or differ), as well as the effects of different parameter combinations, and use the discussion to clarify and discuss some of the aspects mentioned above.

Some detailed comments:

p2, line 14ff: "Inclusion of ..." - As mentioned by the other referee, even the spatial variability of light, nutrient availability and mixing already induce a spatial variability of plankton concentrations.

p2, line 34ff: "However, in biogeochemical models, it is rare that a solid mechanistic basis is present, ..." But see e.g., more recent developments of adaptive models based on mechanistic approaches, such as Pahlow, et al. (2008, Prog.Oceanog., 76 (2), 151-191, doi:10.1016/j.pocean.2007.11.001) or Pahlow, and Prowe, F. (2010), Mar. Ecol. Prog. Ser., 403, 129-144, doi:10.3354/meps08466.

p3 line 5: "applying"

p3 line 9: "highly susceptible" - What does this mean?

p3 line 3: "happened"

p6 line 25: "Oschlies and Garcon, 1999" - a follow-up study by Oschlies and Schartau (2005, Jour. Mar. Res., 63, 335–358) highlighted this even more; see also the study by Friedrichs et al. mentioned above.

p7, section 2.5.1: Physical input: please indicate the vertical grid on which this model was run,including its maximum depth.

p7 section 2.5.2: Biogeochemical input and validation data: I would suggest to list all the details of the different stations (location, max depth, data source, data assimilated) in a table.

p7 section 2.5.2: Do I understand correctly, that the observations were used for initialisation as well as for model validation? If so, then the model is not validated against fully independent data (at least not at depth, given a short simulation time of just 10 years), and I would suggest to mention it here.

p7, line 13: "Simulations are made at 37 depth levels" - This formulation sounds as if simulations were done separately for each depth level.

p15 line 24: "Most current biogeochemical models are run in a deterministic, rather than a probabilistic, manner, even though data from observations contain many uncertainties, eg. in satellite-derived chlorophyll." - I think I can guess what you want to say, but in the current form this sentence is not clear.

---

## Author Comment (AC1) · 7 Jun 2018

We would like to thank the reviewers for taking the time to read the manuscript and for giving constructive comments.

Reviewers 1 and 2 have raised their major concern regarding the default run, its discrepancy with the in situ data, and the parameter values. In an initial experimental run of the model, we use similar parameters as the MEDUSA model however, when the 1D model is run in the oligotrophic regions (station ALOHA and BATS) and in L4 with these parameters the model produces too low chlorophyll, a low correlation coefficient, and a high RMSE, although in PAP the 1D model worked well. We decided to use compromise default parameters which would be the same for all stations, but then see if the ensemble structural variations would lead to improvements. We change the maximum rate of nutrient uptake and grazing at all of the stations along with sinking rate in the coastal station as they have shallower depths, in order to produce a better model run. We have not done a detailed optimisation as we recognised these parameters would not perfectly suit any one station.

Reviewer 2 and 3 also raised some concerns about the function fitting and the large range of phytoplankton and nitrogen concentrations used during the fitting exercise. This large range of phytoplankton and nitrogen concentrations (shown in Figure 1 in the original manuscript) ensures that we captured all the possible concentrations across different stations, and to be consistent on when the functions saturate. If we reduce the range, the parameter values for the particular functional forms do not change too much, and still tend to deviate most strongly when phytoplankton or nutrients are scarce.

All the reviewers agreed that the discussion section needed some major improvements. We will include the studies mentioned by reviewer 3, and we will elaborate on which aspects the ensemble was unable to capture, the benefits of using the ensemble approach compared to one model output, and the importance of conducting structural sensitivity analysis. In the revised manuscript, we will:

- Include a table describing the location, data source, and maximum depth (as suggested by reviewer 3).

- Include annual and seasonal boxplots for nitrogen, annual predicted primary production, and an additional boxplot to highlight the range obtained when changing only one process at a time (as suggested by reviewer 1).

- Change the colour scales in figure 7 and 8, splitting figure 8 into two figures so that the text won't get too small (as suggested by reviewer 2).

- Make the results more concise and the discussion more explicit.

[Figure]

Please see the attached file for our specific response to reviewer comments. The reviewer comments are included in italic, and the responses are in blue.

Please also note the supplement to this comment:
https://www.biogeosciences-discuss.net/bg-2018-136/bg-2018-136-AC1-supplement.pdf

**Supplement:**

**Response to the reviewers:**

**Reviewer #1**

Major

(1) RC: My major criticism is that, in general, the model appears to show major discrepancies with the data, undermining the credibility of the whole exercise, including the conclusions. To be effective, the default model run should show reasonable correspondence with the data but, in several instances, it appears not to do so. Just because the MEDUSA model is already parameterised and published in this regard does not save the situation here because the work involved changing the parameterisations of sinking, maximum and grazing rates (that's rather a lot; page 6, line 7). For example, I am not convinced about the new parameterisation of sinking, namely a sinking rate of 0.1 m d-1 (page 6, line 17) which seems much too low. At PAP, the blues stars (default run) are way too high relative to the blue crosses (observations) indicating a major discrepancy for chlorophyll (Figure 4). The average chlorophyll values for the oligotrophic stations look ok, but the depth plots do not look good at all in this respect (the deep chlorophyll maxima look poorly reproduced; Figure 6). I need more convincing that the model is credible at these sites. There also seem to be large discrepancies for L4 (Figure 4). The modelled vertical concentrations of nitrate at PAP look way too high compared to the data (Figure 3). Why have box and whisker plots not been produced for nitrate, comparing model and data? And why does the appendix (supplementary material) focus only on chlorophyll, and not nitrate? Overall, I am left in doubt as to whether the model, as parameterised for the default run, is credible. The authors could help the situation by looking at some other metrics, if only for the default run. For example, what is predicted primary production at the different sites and how does this compare with data (even just comparing annual average would be highly useful)?

AR: We agree with the reviewer that the default model does not represent the observations convincingly in many of the stations. However one of the objectives of this study was to see how far we can improve the default MEDUSA through structural perturbations in a consistent 1D set up across all stations and so we wanted to keep the model parameters unchanged or as similar as possible at every station. We changed one or two parameters of the default parameters from the literature to allow the default 1D run to be a compromise across all stations, before applying the ensemble. In particular we used 0.8 day-1, and 0.5 day-1 for maximum uptake rate and zooplankton grazing respectively, similar to HadOCC model; A lower sinking rate of 0.1 m d-1 was needed at the coastal stations to prevent the

Figure 1. Chlorophyll and nitrogen concentration in the water column at station L4, when sinking rate is 3 m day-1

nutrients sinking too quickly and being lost, eg. Raick et al. (2006) (a study by Ward (2013) even suggested to use 0 m d-1 for the optimum biogeochemical model). Considering station L4 is only 50m deep, using 3 m d-1 (MEDUSAs original default rate) means that all nutrients are lost from the water column after 2 years (see, the figure 1).

With the original MEDUSA default parameters the model produces too low surface chlorophyll in the oligotrophic stations, but this improves (as the reviewer observed) when the new parameters are used. But of course, the deep chlorophyll maximum it is poorly reproduced using either MEDUSA's default or the modified parameters. This also applies to station L4, where the seasonal pattern is poorly reproduced. However, the default MEDUSA parameter work better for station PAP (with NRR for surface chlorophyll and profile reduced 1.02 and 1.11, but not on nitrogen, the NRR increases to 1.35) and we will now include this in the supplementary material. Our investigations with the default parameters revealed that the large discrepancies between in situ data and the default 1D run was mostly because of the physical input data, especially the vertical velocity and vertical diffusivity coefficient as these drive the upwelling of nutrients. Since these are important to give any realistic interannual variability it is harder to tune these physical inputs in any sensible way. We will emphasise these points in the revised manuscript.

For the nitrogen in station PAP, using nitrogen from the in situ as the initial condition (available from mid-2002) instead of from the test stations (described in section 2.5.2), has improve the nitrogen run and reduced the RMSE of nitrogen (from 3.16 to 2.77), and the NRR of chlorophyll (surface from 1.29 to 0.9 and profile from 1.2 to 1.07) however the nitrogen profile NRR increases (from 1.25 to 1.38). We will include this results in the supplementary material.

In the revised version the metrics for nitrogen and primary production (as suggested by the reviewer) will be included. Further, predicted primary production at stations ALOHA and CARIACO will be included, as the in situ primary production is available only at these two stations.

(2) RC: The ensemble run at each station is initialised using in situ measurements (page6, line 31). What is needed is a stable initial condition, which will not be potentially vulnerable to initial condition instabilities. So surely what is needed is to run the first year over and over (do a spin-up) until a repeating cycle is reached, from which the run through the various years can then be undertaken.

AR: We tried to do a spin-up run for 50 years, using first year's run and the repeating cycle of chlorophyll was achieved after 17 years of run. However, the surface nitrogen kept increasing (up to 40 mmol m-3), again mainly driven by the physical model inputs, because the sum of the first year's vertical velocity is positive (upwards), continuously increasing surface nutrients with time. We decided not to use the spin up run, but instead to use in situ measurements to initialize the model. The same initialization was used for the default and ensemble run. The physical input was averaged every 5 days, controlling the biogeochemical tracers frequently. We will emphasis these points in the revised text and discuss the alternative spin up method in the supplementary material.

(3) RC: A major conclusion of the work is (page 15, line 29) that "small perturbations in model structure can produce a wide range of results". This is a very significant conclusion and I think

the authors can justifiably make it. For the most part, however, the results as shown in the Figures don't show this directly, because they involve various parameterisations acting simultaneously. There is plenty of text in the Results section to support their contention, focusing on individual parameters. I wonder if this conclusion could be better represented in the graphical representation of the results

AR: Thank you for suggesting the graphical representation of one of our main conclusions. We plan to show this in figure 7 and 8, and also using the box plots in figure 4 and 5. We can include a boxplot to show the range in chlorophyll annual means produced when changing only one process at a time thus better supporting the conclusions.

(4) RC: The Introduction is generally well written, introducing the topic of model complexity nicely. The Discussion should mirror the Introduction, saying what the current study has said in context of the wider picture. Instead, the Discussion is mostly just an extended re-hash of the Results and does little to address the big picture. For example, what do the authors conclude about model sensitivity in context of complexity science and the onward drive to produce model of ever increasing complexity? A much bigger play could be made on the need ive in benefits of the ensemble analysis over previous studies that have focused more narrowly on particular parameterisations. Etc. There is plenty of scope and I would say the Discussion section needs a significant overhaul in this regard. It needs re-emphasis; a few extra lines of text will not do.

AR: Thank you for the nice comment on the introduction, and the suggestions on the discussion. We will include these suggestions in the discussion on the revised manuscript.

**Other comments:**

(1) RC: The authors articulate two types of uncertainty (page 2, line 26): "parametric, associated with the choice of parameter values; and structural, which relates to the underlying model equations". Structural uncertainty can also refer to the structure of the model itself (number of compartments, linkages, etc). This should be mentioned, stating that the authors are only looking at structural uncertainty to do with equation formulations.

AR: Thank you for the suggestion, it will be included in the revised text with appropriate references.

(2) RC: On page 9, line 12, there is "A selection of ensemble results are presented". A selection? On what basis?

AR: The selection is based on the available in situ data for nitrogen and chlorophyll and some of the statistical measures we have done. We will rephrase this in the revised manuscript.

(3) RC: Some of the text associated with the Figures is microscopically small.

AR: Thank you for the comment, we agree that some text is too small, and we will make it larger in the revised figures, and split figure 9 into two figures to make the text clear.

(4) RC: Be sure to cite Le Quere, not Quere without "Le".

AR: Thank you, this will be included in the revised manuscript.

**Reviewer #2**

General comments:

(1) RC: The manuscript attempts to show two aspects: (1) there is a high level of structural uncertainty in biogeochemical models and (2) the uncertainty can be exploited to better fit a range of different observations. In my opinion, the authors succeed in providing evidence for first aspect but I have doubts about the second: all comparisons of the ensemble are based on a default run that does not seem to perform very well. Other studies have shown that 1D models with the same parameter values do not perform well across multiple locations but here the same parameter values appear to be used across all stations. Have the parameters of the default run been optimized to fit the datasets used in this study? The results of the default run can have knock-on effects on the ensemble: in multiple parts of the manuscript the authors note that when there is a large bias between the model (ensemble) and the observation, that the ensemble spread is too low when really other model aspects may be to blame for the bias. In other words, problems with the parametrization, the physical model, or the 1D nature of the model cannot be explained by structural uncertainty in the biogeochemical model.

AR: We have not formally optimised the parameter values for each stations. To allow this method to be applied in the 3D MEDUSA we kept the parameters as similar as possible at every station. Please also see response to Q1 from Reviewer 1 above.

(2) RC: When looking at Figure 1, I noticed that the linear function in (c) provides a bad fit to the other functions and that all functions are shown on a log scale. I am wondering if a log-transformation has also been used in the function fitting exercise in Sections 2.1-2.3? If not, I would recommend that this should at least be tried as the procedure could otherwise overemphasize the fit at high tracer concentrations which may explain the slope of the linear function.

AR: We have tried using log-transformation in the function fitting exercise, however, it does not improve the fitting - e.g., the mean absolute error between hyperbolic (the default function) and other mortality functions are larger compare to the regular nonlinear least-squares, summarised in the table 1 and figure 2 shown here. Therefore we decided to stick to a non-transformed fitting.

Table 1. Comparison between log transform and regular function fitting parameter values and its mean absolute errors.

|            | log       | mean   | non log   | mean   |
|------------|-----------|--------|-----------|--------|
| functional | transform | abs    | transform | abs    |
| form       | parameter | error  | parameter | error  |
| sigmoidal  | k = 1.019 | 0.0023 | k = 0.744 | 0.0022 |
| linear     | μ=0.085   | 0.0126 | μ= 0.097  | 0.0085 |
| quadratic  | μ= 0.023  | 0.0035 | μ= 0.050  | 0.0028 |

*Figure 2. Mortality functional forms optimised against hyperbolic function. Dotted lines are fitting without log transform and solid lines are fitting with log transform*

Specific comments:

1) 11: "mathematical structure": What exactly does this refer to? The model formulation? I would suggest to rephrase or an improved explanation

AR: Yes, this means the model formulation. We will rephrase this sentence in the revised manuscript.

2) I3: "intermediately complex BGC model" -> "BGC model of intermediate complexity"

AR: Thank you for the suggestion, we will revise this sentence accordingly.

*3) 19: "using phytoplankton phenology (...) and other statistical measures": phytoplankton phenology is not a statistical measure.*

AR: What we meant in this sentence is that we are using phytoplankton phenology as well as statistical measures (such as RMSE, annual mean, and bias) in order to quantify the impact of structural sensitivity in the ensemble mean, median, and other members. We will revise this sentence in the revised manuscript.

4) 111: Is this the range found in the ensemble (as opposed to e.g. different coastal stations)? Please make this explicit.

AR: This is the range found in the ensemble at the coastal stations. We will revise this sentence in the revised manuscript for clarity.

5) 114: "the errors are mostly reduced": This is not clear: model misfit with respect to the in situ obs is smaller for the ensemble mean/median than the model with standard parameters? I suggest to rephrase.

AR: Yes, this means the model misfit with respect to the in situ observations is smaller for the ensemble mean and median, compared to the default run (using the functional forms in MEDUSA). We will rephrase this in the revised manuscript.

6) *I15: Here a narrow spread is reported, a few lines above a "large" spread was described.*

AR: What we meant was that we do produce large spread, but not wide enough to cover the observation as measured by the NRR.

**Page 2**

7) 17: This reads like the forecasting systems are having an impact on ocean biogeochemistry. The climate change aspect of the sentence reads like a repeat of sentence in line 2. Please revise for clarity.

AR: Thank you for the suggestion, we are trying to give an example of how biogeochemical models may be applied. We will rephrase this sentence in the revised manuscript.

8) 112: Even NPZ models represent "several" processes. Please be more precise.

AR: Thank you, we will rephrase this sentence in the revised manuscript.

9) 116: There can be spatial variability without iron!

AR: We agree with this statement, we will rephrase this sentence into '...such as iron, to permit phytoplankton growth limitation regionally due to the availability of micronutrients' for clarity.

10) I29: "only small perturbations are usually produced even with large variations in parameter values" This is a very strong statement and very much depends on what a "large variation" entails. Perhaps weaken the statement and just make the point that structural uncertainty is often larger than parametric?

AR: Thank you for the suggestion, we will revise this sentence in the revised manuscript.

Page 3:

11) 113: "linear density-dependent mortality produces the most significant differences when applied to diatoms": What exactly does this mean? Please revise.

AR: We meant that the difference is more apparent, we will rephrase this sentence

12) 118: "However, not all processes give significantly different model outputs." The next sentence seems to imply that the differences maybe due to very similar inputs, can this effect thus really be attributed to the process?

AR: In this sentence, we were trying to give an example of how changing the equations of different processes (such as grazing, mortality, and photosynthesis) may give rise to different impacts on phytoplankton dynamics. Changing the equation for photosynthesis in an NPZD model gives little change in phytoplankton dynamics. However, changing the

photosynthesis function has not been tried in our study. We will paraphrase these sentences in the revised manuscript for clarity.

13) I22: "However, it is still unclear what will happened if formulations of all the core processes [...] are perturbed together." The preceding sentence is very general and I would say it is quite clear that the perturbations of all core processes would also "give rise to different effects". I would suggest to rephrase.

AR: Thank you for the suggestion, we will rephrase this sentence in the revised manuscript.

p4:

14) 13: "using all possible functional combinations": Given that there can be an infinite amount of different functional forms, I would suggest to rephrase this sentence. (Later on it becomes clear that only a few functional forms are considered.)

AR: We have rephrased this in the revised manuscript.

15) 122: Mention right away that Table 1 contains the equations for all functions.

AR: Thank you, this will be applied in the manuscript.

16) 129: Mention that "T" is temperature here.

AR: Thank you, this will be applied in the manuscript.

17) |32: "the default": Is this U\_1?

AR: Yes, this is U\_1, and we will revise this in the manuscript as U\_1 instead of default.

p5:

18) 14: "The small microzooplankton": this makes it sound like there are small and large microzooplankton. Use something like "The small zooplankton category consists of microzooplankton..."

AR: Thank you for the suggestion, we will rephrase this in the manuscript as 'The microzooplankton graze on non-diatoms and detritus' instead of using 'The small microzooplankton'

19) I5: Is "non-diatoms" referring to the "smaller phytoplankton" in the previous sentence?

AR: Yes, we will indicate this in the revised manuscript.

20) *18: This is the third time Michaelis-Menten and Holling type II are mentioned together.*

AR: We will be more consistent in the manuscript.

21) 19: "11" -> "111"

AR: We will revised this in the manuscript.

22) I9: Why say "hereafter  $G_1/G_2$ " when "Holling type II/III" is used throughout the text?

AR: We will revise this and use G\_1 and G\_2 elsewhere.

23) I19: Was the shape of the curves adjusted again? If so, how?

AR: Yes, using nonlinear least squares as explained in P4 line 17

24) I29: What is a "distinct trend" here?

AR: For clarity, we will revise this in the manuscript.

25) I30: It is not clear to me how the linear function was made to match the others. Figure 1(c) seems to suggest something went wrong. Or are large values here simply overemphasized in the fit?

AR: Linear function describe constant removal of phytoplankton or zooplankton, therefore we set the maximum rate of the linear mortality to be similar to the total loss of integrated hyperbolic over the prey range, which resulted in 0.09 day-1. We agree that the large values in the prey range may overemphasized the fit, however even after reducing the range to 10 mmol N m-3, the maximum range for the linear has not changed too much (0.086 day-1).

p6:

26) I31: How long is the spin-up period for the runs?

AR: See the answer to Q2 of Reviewer 1

**p7:**

27) *I9:* Why this lengthy comment about physical data assimilation? Is the capping done to remove the perceived negative influence of the physical data assimilation? What about rapid shifts in mixed layer depth which is also an input of the model, may also be affected by physical data assimilation and may also drastically change nutrient concentrations in the model. It is also not quite clear how the mixed layer depth influences the 1D model.

AR: We will reduce the lengthy comment on the data assimilation in the revised manuscript. We take the vertical velocity from the physical data assimilation. This vertical velocity is the most important physical property that determined the results. We also examined the sensitivity for mixed layer depth which is defined by the vertical diffusivity coefficient, using both model output and the mixed layer from the in situ data and we can't see much difference in the biogeochemical model results.

28) I26: It would be good to mention these locations the first time the stations are introduced. Sec 2.5.2: Here the description is confusing, it goes from initial conditions to validation data, back to initial conditions and then to validation data.

AR: Thank you for the suggestion, we will include this in the revised manuscript.

29) 18: "one of MarMOT's test stations" What exactly is this test station?

AR: These are stations that are available within the MarMOT software, which spans from 60° - 10° N, down 20° W in the Atlantic. These stations are used to test whether the MarMOT installation has been successful. The initial conditions are taken from the MEDUSA restart files.

**Р9*:**

- 30) 113: "These have been done at the five oceanographic stations which can be classified into three regional types:" This has been mentioned before.AR: This will be removed in the revised manuscript.
- 31) I21: Mention PAP.
  - AR: This will be included in the revised manuscript.

**p11:**

32) I4: How well does NRR work with a significant bias?

AR: NRR depends on the ratio of the time-averaged RMSE of the ensemble mean to the mean RMSE of the ensemble members. The NRR contain the bias information from the ensemble members, as seen on Table 2.

Table 2. NRR values for Surface chlorophyll at station PAP and various NRR values for different conditions

| Surface       |      |      |
|---------------|------|------|
| Chiorophyli   | INKK |      |
| Original      |      | 1.25 |
| Adding Error  |      | 1.30 |
| Removing Bias |      | 1.22 |

33) *110: "these members use functional combinations …" The notation for the combinations is not clear here*

AR: We will rephrase this sentence in the revised manuscript.

34) Table 1: It does not make sense to call \mu's the maximum rates here.

AR: In the original MEDUSA paper, the maximum loss rates are represented by \mu.

35) Fig 1: Use "U\_1" etc. here.

AR: Thank you, the figure will be revised in the manuscript.

р8

36) Fig 7: A better description of the x and y axes are needed. Why do b,d,f and h have no y-axis? Use the same color scale across all stations. Same comment applies to Fig. 8 where the font becomes too small.

AR: Thank you for the suggestions. We will add more description of the x and y-axes in figure 7 and 8 in the revised manuscript. Figure 7 b, d, and f have the same y-tick labels as a, c, and e, therefore in order to maximise the space, we decided not to put the y-tick label. In terms of colour scale, we are not quite sure whether using the same scale across all stations would be a good idea, due to the range of values between different stations and regions. For example, the chlorophyll profile RMSE at station ALOHA and BATS are on different range (ALOHA is between 0.08 and 0.15, and BATS is between 0.3 and 0.35). Therefore we will keep the colour scale on the nitrogen and chlorophyll concentrations between regions similar, and if possible also in the RMSEs.

**Reviewer #3**

Major comments:

1) Firstly, in the introduction (page three, line 29) the authors state that "It has been demonstrated in conventional sensitivity analyses that only small perturbations are usually produced even with large variations in parameter values, but much larger changes in system dynamics can result from changes in the structural process formulations". I am not quite sure what "conventional" means, but I do think that this statement is misleading, as it neglects previous works that indicate a large sensitivity of marine biogeochemical models to their parameters, when compared to structural sensitivity. These studies have been carried out at a local scale, across different oceanic regimes, or in 3D (see, e.g., Friedrichs et al., 2007, Jour. Geophys. Res., 112, C08001, doi:10.1029/2006JC003852; Ward et al., 2013, Prog. Oceanog.116,49–65, or Kriest et al., 2012, Glob. Biogeochem. Cyc. 26, GB2029, doi:10.1029/2011GB004072, to name just a few examples). Some of them even address the role of different functional forms, or have been applied to the BATS site (e.g., study by Ward et al., 2013). They may be helpful for presenting and discussing this current work in a wider context. Thus, more exploration about what has been found for marine biogeochemical models and their structural and parametric uncertainty can help to improve the discussion, which is currently somehow repetitive, lacks a critical discussion of the results, and how they might relate to other uncertainties (structural, parametric, physical, ...).

AR: Here "conventional sensitivity analysis" was referring to parameter sensitivity analysis, but not the structural sensitivity. We will clarify this in the revised version. Thank you for suggesting the relevant papers also, which will use for comparisons in our largely revised discussion section.

2) Secondly, I miss some discussion about the way the different functional forms have been made "equivalent to each other." (p4 line 17). As it seems, the parameters of the different equations (e.g., half saturation-constants) were fitted against the default function "so that the overall shapes are as similar as possible." (p 4, line 19), by "minimising the sum squared difference between the default and other uptake forms" (line 32ff). Obviously, when looking at Fig 1, this happened across a very wide range of potential nutrient or chlorophyll (in case

of zooplankton grazing) concentrations. The upper limits are far outside the range of values for most stations simulated in this study (up to 100 uM nitrate or phytoplankton N will likely never be found at BATS or ALOHA). Thus, it seems that the different functional forms were homogenised for a range that, at many stations, is outside the expected and/or observed range. On the other hand, the functions deviate most strongly when nutrients or phytoplankton are scarce (Fig 1a and 1b), and more representative for the simulated regimes. What would have happened, if the test functions (e.g., sigmoidal or Holling III) were made equivalent to the default functions at lower substrate levels, representative for more oligotrophic regimes? Could it be that the effects of switching to alternative forms becomes less important? Again, the paper to my opinion would benefit a lot from a more critical discussion.

AR: We agree that from looking at figure 1a and 1b, the functions deviate mostly when the nutrient or phytoplankton are scarce, and overfitting may occur due to the large value of nitrogen and phytoplankton. However, we are trying to capture the whole range of nutrient and phytoplankton at all the different region, and optimise the functions when both are the closest to each other (when phytoplankton and nutrient are plentiful) and within the nitrogen and chlorophyll range of all the stations. (See also response to Q2 and 25 of reviewer 2) Suppose we are optimising the nutrient uptake on the similar range of station BATS and ALOHA (with maximum nitrogen and phytoplankton concentration of 5 mmol N m- 3, shown on Figure 3, although at stations like Cariaco, PAP, and L4, we may see nitrogen larger than 5 mmol N m-3), the functions still deviate at low nitrogen and phytoplankton concentration. Additionally, the value of half saturation constant have not changed much (for nutrient, the half saturation constant for sigmoidal, exponential, and trigonometric are 0.71, 1.10, and 0.58 respectively, and for grazing the half saturation constant for Holling type II is 0.48). Therefore, the effects of switching to alternative forms will still generate a range of different model outputs. We will change Figure 1 in the manuscript to only use the range that are available in the model (between 0 - 20 mmol N m-3 for nitrogen and 0 - 10 mmol N m-3 for phytoplankton).

Figure 3. Uptake (a) and grazing (b) functions which have been optimised, with range of 0.001 to 5 mmol N m-3.

3) Thirdly, as recommended by the second referee, I suggest that the authors read through the manuscript again carefully, revise some sections for clarity, and correct spelling and grammar. The results section already contains a lot of detail, which is partly repeated in the discussion. I would suggest to to shorten and streamline the presentation of results, highlighting those that are common among stations (or differ), as well as the effects of different parameter combinations, and use the discussion to clarify and discuss some of the aspects mentioned above.

AR: Thank you for the suggestions, and also the addition of literatures which you have suggested. We have indeed revised and streamlined the new paper.

Some detailed comments:

1) p2, line 14ff: "Inclusion of ..." - As mentioned by the other referee, even the spatial variability of light, nutrient availability and mixing already induce a spatial variability of plankton concentrations.

AR: This will be rephrased in the revised manuscript

 p2, line 34ff: "However, in biogeochemical models, it is rare that a solid mechanistic basis is present, ..." But see e.g., more recent developments of adaptive models based on mechanistic approaches, such as Pahlow, et al. (2008, Prog.Oceanog., 76 (2), 151-191, doi:10.1016/j.pocean.2007.11.001) or Pahlow, and Prowe, F. (2010), Mar. Ecol. Prog. Ser., 403, 129-144, doi:10.3354/meps08466.

AR: We will remove this statement in the manuscript

3) p3 line 5: "applying"

AR: We will rephrase this to 'applied' in the manuscript

4) p3 line 9: "highly susceptible" - What does this mean?

AR: It means that biogeochemical model is likely to be structurally sensitive. We will rephrase this sentence.

5) p3 line 3: "happened"

AR: We can't find happened in p3 line 3 - if this is in line 23, we will rephrase this sentence as mentioned by reviewer #2

 p6 line 25: "Oschlies and Garcon, 1999" - a follow-up study by Oschlies and Schartau (2005, Jour. Mar. Res., 63, 335–358) highlighted this even more; see also the study by Friedrichs et al. mentioned above. AR: Thank you for the suggestions, we will include these literatures accordingly.

7) p7, section 2.5.1: Physical input: please indicate the vertical grid on which this model was run, including its maximum depth.

AR: This has been stated in the biogeochemical input but this will be revised in the manuscript.

8) p7 section 2.5.2: Biogeochemical input and validation data: I would suggest to list all the details of the different stations (location, max depth, data source, data assimilated) in a table.

AR: Thank you for the suggestion, we will include this in the new manuscript, however we do not assimilate any data into our model

9) p7 section 2.5.2: Do I understand correctly, that the observations were used for initialisation as well as for model validation? If so, then the model is not validated against fully independent data (at least not at depth, given a short simulation time of just 10 years), and I would suggest to mention it here.

AR: Indeed, we are using the observation to initialise the model (using in situ chlorophyll, nitrogen, iron, and silicate data from January 1998), but we do not use the later in situ data to force the model, so the validation data is independent.

10) p7, line 13: "Simulations are made at 37 depth levels" - This formulation sounds as if simulations were done separately for each depth level.

AR: This will be rephrased in the revised manuscript.

11) p15 line 24: "Most current biogeochemical models are run in a deterministic, rather than a probabilistic, manner, even though data from observations contain many uncertainties, eg. in satellite-derived chlorophyll." - I think I can guess what you want to say, but in the current form this sentence is not clear.

AR: This will be rephrased in the revised manuscript for clarity.

Literatures cited:

Raick, C., Soetaert, K., and Grégoire, M.: Model complexity and performance: How far can we simplify? Progress in Oceanography, 70, 27–57, https://doi.org/10.1016/j.pocean.2006.03.001, 2006.

Ward, B. A., Schartau, M., Oschlies, A., Martin, A. P., Follows, M. J., and Anderson, T. R.: When is a biogeochemical model too complex? Objective model reduction and selection for North Atlantic time-series sites, Progress in Oceanography, 116, 49–65, https://doi.org/10.1016/j.pocean.2013.06.002, 2013.

---

## Author Response (AR1)

Dear Editor,

We would like to thank the reviewers for taking the time to read the manuscript and for giving constructive comments. In the revised manuscript, we have taken into consideration all the comments and suggestions made by the referees. In particular we have:

> • Included a table (see Table 2) describing the location, data source, and maximum depth (as suggested by reviewer 3).
>
> • Included annual and seasonal boxplots for nitrogen (see Fig 6 and 7), annual predicted primary production (see Fig 10), and an additional boxplot (see Fig 8) to highlight the range obtained when changing only one process at a time (as suggested by reviewer 1).
>
> • Changed the colour scales in figure 7 and 8 (see the new Fig 11 and 12), splitting figure 8 into two figures (see Fig 12 and 13) so that the text won't get too small (as suggested by reviewer 2).
>
> • Made the results more concise (see page 10-18 in the annotated version) and the discussion (page 18-23 in the annotated version) more explicit.

In addition below is our point by point response to all comments made by the three reviewers (reviewer comment in black, response in blue, and changes to the manuscript indicated in blue bold). Please note that all line numbers in this response refer to the annotated version attached. We have further uploaded a clean version separately.

We hope that the response would be satisfactory, and we look forward to your decision.

Kind Regards,

Authors.

**Response to the reviewers:**

**Reviewer #1**

Major

> (1) *RC: My major criticism is that, in general, the model appears to show major discrepancies with the data, undermining the credibility of the whole exercise, including the conclusions. To be effective, the default model run should show reasonable correspondence with the data but, in several instances, it appears not to do so. Just because the MEDUSA model is already parameterised and published in this regard does not save the situation here because the work involved changing the parameterisations of sinking, maximum and grazing rates (that's rather a lot; page 6, line 7). For example, I am not convinced about the new parameterisation of sinking, namely a sinking rate of 0.1 m d-1 (page 6, line 17) which seems much too low. At PAP, the blues stars (default run) are way too high relative to the blue crosses (observations) indicating a major discrepancy for chlorophyll (Figure 4). The average chlorophyll values for the oligotrophic stations look ok, but the depth plots do not look good at all in this respect (the deep chlorophyll maxima look poorly reproduced; Figure 6). I need more convincing that*

*the model is credible at these sites. There also seem to be large discrepancies for L4 (Figure 4). The modelled vertical concentrations of nitrate at PAP look way too high compared to the data (Figure 3). Why have box and whisker plots not been produced for nitrate, comparing model and data? And why does the appendix (supplementary material) focus only on chlorophyll, and not nitrate? Overall, I am left in doubt as to whether the model, as parameterised for the default run, is credible. The authors could help the situation by looking at some other metrics, if only for the default run. For example, what is predicted primary production at the different sites and how does this compare with data (even just comparing annual average would be highly useful)?*

AR: We agree with the reviewer that the default model does not represent the observations convincingly in many of the stations. However one of the objectives of this study was to see how far we can improve the default MEDUSA through structural perturbations in a consistent 1D set up across all stations and so we wanted to keep the model parameters unchanged or as similar as possible at every station. We changed one or two parameters of the default parameters from the literature to allow the default 1D run to be a compromise across all stations, before applying the ensemble. In particular we used 0.8 day$^{-1}$, and 0.5 day$^{-1}$ for maximum uptake rate and zooplankton grazing respectively, similar to HadOCC model; A lower sinking rate of 0.1 m d$^{-1}$ was needed at the coastal stations to prevent the

[Figure]

Figure 1. Chlorophyll and nitrogen concentration in the water column at station L4, when sinking rate is 3 m day$^{-1}$

nutrients sinking too quickly and being lost, eg.  Raick et al. (2006) (a study by Ward (2013) even suggested to use 0 m d$^{-1}$ for the optimum biogeochemical model). Considering station L4 is only 50m deep, using 3 m d$^{-1}$ (MEDUSAs original default rate) means that all nutrients are lost from the water column after 2 years (see, the figure 1).

With the original MEDUSA default parameters the model produces too low surface chlorophyll in the oligotrophic stations, but this improves (as the reviewer observed) when the new parameters are used. But of course, the deep chlorophyll maximum it is poorly reproduced using either MEDUSA's default or the modified parameters. This also applies to station L4, where the seasonal pattern is poorly reproduced. However, the default MEDUSA parameter work better for station PAP (with NRR for surface chlorophyll and profile reduced 1.02 and 1.11 respectively, but not on nitrogen, the NRR increases to 1.35) and **we have included these experiments in the supplementary material S2.** Our investigations with the default parameters revealed that the large discrepancies between in situ data and the default 1D run was mostly because of the physical input data, especially the vertical velocity and vertical diffusivity coefficient as these drive the upwelling of nutrients. Since these are

important to give any realistic interannual variability it is harder to tune these physical inputs in any sensible way. We have emphasises these points in the revised manuscript.

For the nitrogen in station PAP, using nitrogen from the in situ as the initial condition (available from mid-2002) instead of from the test stations (described in section 2.5.2), has improve the nitrogen run and reduced the RMSE of nitrogen (from 3.16 to 2.77), and the NRR of chlorophyll (surface from 1.29 to 0.9 and profile from 1.2 to 1.07) however the nitrogen profile NRR increases (from 1.25 to 1.38). **We have included this results in the supplementary material S3.**

**In the revised version the metrics for nitrogen and primary production (as suggested by the reviewer) have been included in Fig 6 for inter-annual variability and 7 for seasonal means. Further, predicted primary production at stations ALOHA and CARIACO have been included in Fig 10, as the in situ primary production is available only at these two stations.**

(2) *RC: The ensemble run at each station is initialised using in situ measurements (page6, line 31). What is needed is a stable initial condition, which will not be potentially vulnerable to initial condition instabilities. So surely what is needed is to run the first year over and over (do a spin-up) until a repeating cycle is reached, from which the run through the various years can then be undertaken.*

AR: We tried to do a spin-up run for 50 years, using first year's run and the repeating cycle of chlorophyll was achieved after 17 years of run. However, the surface nitrogen kept increasing (up to 40 mmol m$^{-3}$), again mainly driven by the physical model inputs, because the sum of the first year's vertical velocity is positive (upwards), continuously increasing surface nutrients with time. We decided not to use the spin up run, but instead to use in situ measurements to initialize the model. The same initialization was used for the default and ensemble run. The physical input was averaged every 5 days, controlling the biogeochemical tracers frequently. **We have emphasise these points in the revised text and discuss the alternative spin up method in the supplementary material section S1.**

(3) *RC: A major conclusion of the work is (page 15, line 29) that "small perturbations in model structure can produce a wide range of results". This is a very significant conclusion and I think the authors can justifiably make it. For the most part, however, the results as shown in the Figures don't show this directly, because they involve various parameterisations acting simultaneously. There is plenty of text in the Results section to support their contention, focusing on individual parameters. I wonder if this conclusion could be better represented in the graphical representation of the results*

AR: Thank you for suggesting the graphical representation of one of our main conclusions. We plan to show this in figure 7 and 8, in the revised manuscript is now in Fig. 11, 12, and 13, and also using the box plots in figure 4 and 5. **We have include a boxplot to show the range in chlorophyll annual means produced when changing only one process at a time thus better supporting the conclusions, in Fig. 8.**

(4) *RC: The Introduction is generally well written, introducing the topic of model complexity nicely. The Discussion should mirror the Introduction, saying what the current study has said in context of the wider picture. Instead, the Discussion is mostly just an extended re-hash of*

*the Results and does little to address the big picture. For example, what do the authors conclude about model sensitivity in context of complexity science and the onward drive to produce model of ever increasing complexity? could be made on the need to do sensitivity analysis in benefits of the ensemble analysis over previous studies that have focused more narrowly on particular parameterisations. Etc. There is plenty of scope and I would say the Discussion section needs a significant overhaul in this regard. It needs re-emphasis; a few extra lines of text will not do.*

AR: Thank you for the nice comment on the introduction, and the suggestions on the discussion. **We have include these suggestions in the discussion (page 18-25)**

**Other comments:**

(1) *RC: The authors articulate two types of uncertainty (page 2, line 26): "parametric, associated with the choice of parameter values; and structural, which relates to the underlying model equations". Structural uncertainty can also refer to the structure of the model itself (number of compartments, linkages, etc). This should be mentioned, stating that the authors are only looking at structural uncertainty to do with equation formulations.*

AR: Thank you for the suggestion, **this have been included in the revised text with appropriate references (line 1-4 page 3)**

(2) *RC: On page 9, line 12, there is "A selection of ensemble results are presented". A selection? On what basis?*

AR: The selection is based on the available in situ data for nitrogen and chlorophyll and some of the statistical measures we have done. **We have removed this and make the paragraph shorter (line 12-17 page 10)**.

(3) *RC: Some of the text associated with the Figures is microscopically small.*

AR: Thank you for the comment, we agree that some text is too small, and we have make it larger in the revised figures**, and split figure 9 into two figures (Fig. 12 and 13) to make the text clear.**

(4) *RC: Be sure to cite Le Quere, not Quere without "Le".*

AR: Thank you, **this have been included in the revised manuscript.**

**Reviewer #2**

General comments:

(1) *RC: The manuscript attempts to show two aspects: (1) there is a high level of structural uncertainty in biogeochemical models and (2) the uncertainty can be exploited to better fit a range of different observations. In my opinion, the authors succeed in providing evidence for first aspect but I have doubts about the second: all comparisons of the ensemble are based on a default run that does not seem to perform very well. Other studies have shown that 1D models with the same parameter values do not perform well across multiple locations but here the same parameter values appear to be used across all stations. Have the parameters of the default run been optimized to fit the datasets used in this study? The results of the*

*default run can have knock-on effects on the ensemble: in multiple parts of the manuscript the authors note that when there is a large bias between the model (ensemble) and the observation, that the ensemble spread is too low when really other model aspects may be to blame for the bias. In other words, problems with the parametrization, the physical model, or the 1D nature of the model cannot be explained by structural uncertainty in the biogeochemical model.*

AR: We have not formally optimised the parameter values for each stations. To allow this method to be applied in the 3D MEDUSA we kept the parameters as similar as possible at every station. Please also see response to Q1 from Reviewer 1 above.

(2) *RC: When looking at Figure 1, I noticed that the linear function in (c) provides a bad fit to the other functions and that all functions are shown on a log scale. I am wondering if a log-transformation has also been used in the function fitting exercise in Sections 2.1-2.3? If not, I would recommend that this should at least be tried as the procedure could otherwise overemphasize the fit at high tracer concentrations which may explain the slope of the linear function.*

AR: We have tried using log-transformation in the function fitting exercise, however, it does not improve the fitting - e.g., the mean absolute error between hyperbolic (the default function) and other mortality functions are larger compare to the regular nonlinear least-squares, summarised in the table 1 and figure 2 shown here. Therefore we decided to stick to a non-transformed fitting.

*Table 1. Comparison between log transform and regular function fitting parameter values and its mean absolute errors.*

| functional form | log transform parameter | mean abs error | non log transform parameter | mean abs error |
|---|---|---|---|---|
| sigmoidal | k = 1.019 | 0.0023 | k = 0.744 | 0.0022 |
| linear | μ=0.085 | 0.0126 | μ= 0.097 | 0.0085 |
| quadratic | μ= 0.023 | 0.0035 | μ= 0.050 | 0.0028 |

[Figure]

*Figure 2. Mortality functional forms optimised against hyperbolic function. Dotted lines are fitting without log transform and solid lines are fitting with log transform*

Specific comments:

1) *l1: "mathematical structure": What exactly does this refer to? The model formulation? I would suggest to rephrase or an improved explanation*

   AR: Yes, this means the model formulation. **We have rephrase this sentence and change 'mathematical structure' into 'mathematical equations' (page 1 line 1-2).**

2) *l3: "intermediately complex BGC model" -> "BGC model of intermediate complexity"*

   AR: Thank you for the suggestion, **we have rephrased this sentence as suggested (page 1 line 4).**

3) *l9: "using phytoplankton phenology (...) and other statistical measures": phytoplankton phenology is not a statistical measure.*

   AR: What we meant in this sentence is that we are using phytoplankton phenology as well as statistical measures (such as RMSE, annual mean, and bias) in order to quantify the impact of structural sensitivity in the ensemble mean, median, and other members. **We have revised this sentence please see page 1 line 9-10.**

4) *l11: Is this the range found in the ensemble (as opposed to e.g. different coastal stations)? Please make this explicit.*

   AR: This is the range found in the ensemble at the coastal stations. **We have revised this sentence in the annotated manuscript for clarity (page 1 line 12-13)**.

5) *l14: "the errors are mostly reduced": This is not clear: model misfit with respect to the in situ obs is smaller for the ensemble mean/median than the model with standard parameters? I suggest to rephrase.*

   AR: Yes, this means the model misfit with respect to the in situ observations is smaller for the ensemble mean and median, compared to the default run (using the functional forms in MEDUSA). **We have rephrased this in the manuscript as RMSEs instead of errors (page 1 line 16).**

6) *l15: Here a narrow spread is reported, a few lines above a "large" spread was described.*

   AR: What we meant was that we do produce large spread, but not wide enough to cover the observation as measured by the NRR.

7) *l7: This reads like the forecasting systems are having an impact on ocean biogeochemistry. The climate change aspect of the sentence reads like a repeat of sentence in line 2. Please revise for clarity.*

AR: Thank you for the suggestion, we are trying to give an example of how biogeochemical models may be applied. **We have rephrased this sentence in the annotated manuscript into '…address and predict the impact of climate change in the ocean ecosystems…' (page 2 line 7-9).**

8) *l12: Even NPZ models represent "several" processes. Please be more precise.*

AR: Thank you, **we have rephrase this sentence in the annotated manuscript into '….More advanced biogeochemical models represent more processes and feedbacks compared to….' (page 2 line 15-16).**

9) *l16: There can be spatial variability without iron!*

AR: We agree with this statement, **we have rephrase this sentence into '…such as iron, to permit phytoplankton growth limitation' (page 2 line 18-19) for clarity.**

10) *l29: "only small perturbations are usually produced even with large variations in parameter values" This is a very strong statement and very much depends on what a "large variation" entails. Perhaps weaken the statement and just make the point that structural uncertainty is often larger than parametric?*

AR: Thank you for the suggestion, **we have revised this sentence in the annotated manuscript into '… small changes in the structural process formulation often produce larger changes in the system dynamics, compared to varying parameter values alone' (page 3 line 4-5).**

Page 3:

11) *l13: "linear density-dependent mortality produces the most significant differences when applied to diatoms": What exactly does this mean? Please revise.*

AR: We meant that the difference is more apparent, **we have rephrased this sentence in the manuscript '… linear density-dependent mortality produces the biggest difference in diatoms with concentrations at mid latitudes being twice as high…' (page 3 line 26-27).**

12) *l18: "However, not all processes give significantly different model outputs." The next sentence seems to imply that the differences maybe due to very similar inputs, can this effect thus really be attributed to the process?*

AR: In this sentence, we were trying to give an example of how changing the equations of different processes (such as grazing, mortality, and photosynthesis) may give rise to different impacts on phytoplankton dynamics. Changing the equation for photosynthesis in an NPZD model gives little change in phytoplankton dynamics. However, changing the photosynthesis function has not been tried in our study. **We have shortened these sentences in the revised manuscript for clarity (page 3 line 31-35).**

13) *l22: "However, it is still unclear what will happened if formulations of all the core processes [...] are perturbed together." The preceding sentence is very general and I would say it is*

*quite clear that the perturbations of all core processes would also "give rise to different effects". I would suggest to rephrase.*

AR: Thank you for the suggestion, we have removed this in the revised manuscript **(page 4 line 3).**

p4:

*14)* l3: *"using all possible functional combinations": Given that there can be an infinite amount of different functional forms, I would suggest to rephrase this sentence. (Later on it becomes clear that only a few functional forms are considered.)*

AR: We have rephrased this in the revised manuscript into **'… using possible functional form combinations within the NPZ compartments…' (page 4 line 18)**

*15)* l22: *Mention right away that Table 1 contains the equations for all functions.*

AR: Thank you, **this have been applied in the manuscript (page 5 line 2)**.

*16)* l29: *Mention that "T" is temperature here.*

AR: Thank you, **this have been applied in the manuscript (page 5 line 11)**.

*17)* l32: *"the default": Is this U_1?*

AR: Yes, this is U_1, and we **have revise this in the manuscript as U_1 instead of default (page 5 line 14).**

p5:

*18)* l4: *"The small microzooplankton": this makes it sound like there are small and large microzooplankton. Use something like "The small zooplankton category consists of microzooplankton..."*

AR: Thank you for the suggestion, we have rephrased this sentence into **'The small zooplankton, represented by the microzooplankton, graze on small phytoplankton, non-diatoms, and detritus …' (page 5 line 18-19)**

19) l5: *Is "non-diatoms" referring to the "smaller phytoplankton" in the previous sentence?*

AR: Yes, **we have indicate this in the revised manuscript (page 5 line 19)**.

*20)* l8: *This is the third time Michaelis-Menten and Holling type II are mentioned together.*

AR: We have changed this throughout the manuscript.

*21)* l9: *"II" -> "III"*

AR: We have revised this in the manuscript **(page 5 line 23)**

22) *l9: Why say "hereafter G_1/G_2" when "Holling type II/III" is used throughout the text?*

AR: **We have revised this and use G_1 and G_2 elsewhere.**

23) *l19: Was the shape of the curves adjusted again? If so, how?*

AR: Yes, **using nonlinear least squares as explained in page 4 line 31-34 in the annotated manuscript**

24) *l29: What is a "distinct trend" here?*

*AR:* For clarity, **we have revised this in the manuscript (page 6 line 12-13).**

25) *l30: It is not clear to me how the linear function was made to match the others. Figure 1(c) seems to suggest something went wrong. Or are large values here simply overemphasized in the fit?*

AR: Linear function describe constant removal of phytoplankton or zooplankton, therefore we set the maximum rate of the linear mortality to be similar to the total loss of integrated hyperbolic over the prey range, which resulted in 0.09 day$^{-1}$. We agree that the large values in the prey range may overemphasized the fit, however even after reducing the range to 10 mmol N m$^{-3}$, the maximum range for the linear has not changed too much (0.086 day$^{-1}$).

p6:

26) *l31: How long is the spin-up period for the runs?*

AR: See the answer to Q2 of Reviewer 1

p7:

27) *l9: Why this lengthy comment about physical data assimilation? Is the capping done to remove the perceived negative influence of the physical data assimilation? What about rapid shifts in mixed layer depth which is also an input of the model, may also be affected by physical data assimilation and may also drastically change nutrient concentrations in the model. It is also not quite clear how the mixed layer depth influences the 1D model.*

AR: We take the vertical velocity from the physical data assimilation. This vertical velocity is the most important physical property that determined the results. We also examined the sensitivity for mixed layer depth which is defined by the vertical diffusivity coefficient, using both model output and the mixed layer from the in situ data and we can't see much difference in the biogeochemical model results. **We have reduce the lengthy comment on the data assimilation in the revised manuscript, it's now on page 7 line 26-33 in the annotated manuscript**.

28) *l26: It would be good to mention these locations the first time the stations are introduced. Sec 2.5.2: Here the description is confusing, it goes from initial conditions to validation data, back to initial conditions and then to validation data.*

AR: Thank you for the suggestion, **we have now revise this description of the station at the start of section 2.5.2 (page 8 line 13-15).**

p8

29) l8: "one of MarMOT's test stations" What exactly is this test station?

AR: These are stations that are available within the MarMOT software, which spans from 60° - 10° N, down 20° W in the Atlantic. These stations are used to test whether the MarMOT installation has been successful. The initial conditions are taken from the MEDUSA restart files.

P9:

30) l13: "These have been done at the five oceanographic stations which can be classified into three regional types:" This has been mentioned before.

AR: **We have removed this in the annotated manuscript (please see the start of the Results section on page 10 line 12-17).**

31) l21: Mention PAP.

AR: **We have mentioned this in the annotated manuscript (page 10 line 19)**

p11:

32) l4: How well does NRR work with a significant bias?

AR: NRR depends on the ratio of the time-averaged RMSE of the ensemble mean to the mean RMSE of the ensemble members. The NRR contain the bias information from the ensemble members, as seen on Table 2.

*Table 2. NRR values for Surface chlorophyll at station PAP and various NRR values for different conditions*

| Surface Chlorophyll | NRR |
|---|---|
| Original | 1.25 |
| Adding Error | 1.30 |
| Removing Bias | 1.22 |

33) l10: "these members use functional combinations ..." The notation for the combinations is not clear here

AR: We have rephrase this in the manuscript into **'… show that ensemble members with…' (Page 12 line 33)**

34) Table 1: It does not make sense to call \mu's the maximum rates here.

AR: In the original MEDUSA paper, the maximum loss rates are represented by \mu.

35) *Fig 1: Use "U_1" etc. here.*

AR: **We have included this in the manuscript, please see Fig. 1**

36) *Fig 7: A better description of the x and y axes are needed. Why do b,d,f and h have no y-axis? Use the same color scale across all stations. Same comment applies to Fig. 8 where the font becomes too small.*

AR: Thank you for the suggestions. **We have added more description of the x and y-axes in figure 7 and 8 in the revised manuscript (now Fig. 11-13).** Figure 7 b, d, and f have the same y-tick labels as a, c, and e, therefore in order to maximise the space, we decided not to put the y-tick label. In terms of colour scale, we are not quite sure whether using the same scale across all stations would be a good idea, due to the range of values between different stations and regions. For example, the chlorophyll profile RMSE at station ALOHA and BATS are on different range (ALOHA is between 0.08 and 0.15, and BATS is between 0.3 and 0.35). Therefore we will keep the colour scale on the nitrogen and chlorophyll concentrations between regions similar if possible, and if possible also in the RMSEs.

**Reviewer #3**

*Major comments:*

1) *Firstly, in the introduction (page three, line 29) the authors state that "It has been demonstrated in conventional sensitivity analyses that only small perturbations are usually produced even with large variations in parameter values, but much larger changes in system dynamics can result from changes in the structural process formulations". I am not quite sure what "conventional" means, but I do think that this statement is misleading, as it neglects previous works that indicate a large sensitivity of marine biogeochemical models to their parameters, when compared to structural sensitivity. These studies have been carried out at a local scale, across different oceanic regimes, or in 3D (see, e.g., Friedrichs et al., 2007, Jour. Geophys. Res., 112, C08001, doi:10.1029/2006JC003852; Ward et al., 2013, Prog. Oceanog.116,49–65, or Kriest et al., 2012, Glob. Biogeochem. Cyc. 26, GB2029, doi:10.1029/2011GB004072, to name just a few examples). Some of them even address the role of different functional forms, or have been applied to the BATS site (e.g., study by Ward et al., 2013). They may be helpful for presenting and discussing this current work in a wider context. Thus, more exploration about what has been found for marine biogeochemical models and their structural and parametric uncertainty can help to improve the discussion, which is currently somehow repetitive, lacks a critical discussion of the results, and how they might relate to other uncertainties (structural, parametric, physical, ...).*

AR: Here "conventional sensitivity analysis" was referring to parameter sensitivity analysis, but not the structural sensitivity. **We have removed this statement in the revised version, and paraphrased it (page 3 line 3-4).** Thank you for suggesting the relevant papers also,

which have used these literatures for comparisons in our largely revised discussion section **(page 23, starting line 20-25)**.

2) *Secondly, I miss some discussion about the way the different functional forms have been made "equivalent to each other." (p4 line 17). As it seems, the parameters of the different equations (e.g., half saturation-constants) were fitted against the default function "so that the overall shapes are as similar as possible." (p 4, line 19), by "minimising the sum squared difference between the default and other uptake forms" (line 32ff). Obviously, when looking at Fig 1, this happened across a very wide range of potential nutrient or chlorophyll (in case of zooplankton grazing) concentrations. The upper limits are far outside the range of values for most stations simulated in this study (up to 100 uM nitrate or phytoplankton N will likely never be found at BATS or ALOHA). Thus, it seems that the different functional forms were homogenised for a range that, at many stations, is outside the expected and/or observed range. On the other hand, the functions deviate most strongly when nutrients or phytoplankton are scarce (Fig 1a and 1b), and more representative for the simulated regimes. What would have happened, if the test functions (e.g., sigmoidal or Holling III) were made equivalent to the default functions at lower substrate levels, representative for more oligotrophic regimes? Could it be that the effects of switching to alternative forms becomes less important? Again, the paper to my opinion would benefit a lot from a more critical discussion.*

AR: We agree that from looking at figure 1a and 1b, the functions deviate mostly when the nutrient or phytoplankton are scarce, and overfitting may occur due to the large value of nitrogen and phytoplankton. However, we are trying to capture the whole range of nutrient and phytoplankton at all the different region, and optimise the functions when both are the closest to each other (when phytoplankton and nutrient are plentiful) and within the nitrogen and chlorophyll range of all the stations. (See also response to Q2 and 25 of reviewer 2) Suppose we are optimising the nutrient uptake on the similar range of station BATS and ALOHA (with maximum nitrogen and phytoplankton concentration of 5 mmol N m$^{-3}$, shown on Figure 3, although at stations like Cariaco, PAP, and L4, we may see nitrogen larger than 5 mmol N m$^{-3}$), the functions still deviate at low nitrogen and phytoplankton concentration. Additionally, the value of half saturation constant have not changed much (for nutrient, the half saturation constant for sigmoidal, exponential, and trigonometric are 0.71, 1.10, and 0.58 respectively, and for grazing the half saturation constant for Holling type II is 0.48). Therefore, the effects of switching to alternative forms will still generate a range of different model outputs. **We have changed Figure 1 in the manuscript to only use the range that are available in the model (between 0 – 20 mmol N m$^{-3}$ for nitrogen and 0 – 10 mmol N m$^{-3}$ for phytoplankton).**

[Figure]

*Figure 3. Uptake (a) and grazing (b) functions which have been optimised, with range of 0.001 to 5 mmol N m⁻³.*

3) *Thirdly, as recommended by the second referee, I suggest that the authors read through the manuscript again carefully, revise some sections for clarity, and correct spelling and grammar. The results section already contains a lot of detail, which is partly repeated in the discussion. I would suggest to to shorten and streamline the presentation of results, highlighting those that are common among stations (or differ), as well as the effects of different parameter combinations, and use the discussion to clarify and discuss some of the aspects mentioned above.*

AR: Thank you for the suggestions, and also the addition of literatures which you have suggested. We have indeed revised and streamlined the new paper.

Some detailed comments:

1) *p2, line 14ff: "Inclusion of ..." - As mentioned by the other referee, even the spatial variability of light, nutrient availability and mixing already induce a spatial variability of plankton concentrations.*

AR: We have rephrased this on the main manuscript, **please see reviewer 2's answer no 9**

2) *p2, line 34ff: "However, in biogeochemical models, it is rare that a solid mechanistic basis is present, ..." But see e.g., more recent developments of adaptive models based on mechanistic approaches, such as Pahlow, et al. (2008, Prog.Oceanog., 76 (2), 151- 191, doi:10.1016/j.pocean.2007.11.001) or Pahlow, and Prowe, F. (2010), Mar. Ecol. Prog. Ser., 403, 129-144, doi:10.3354/meps08466.*

**AR: We have removed this statement in the manuscript.**

3) *p3 line 5: "applying"*

AR: We have removed this **please see page 3 line 16**.

4) *p3 line 9: "highly susceptible" - What does this mean?*

AR: It means that biogeochemical model is likely to be structurally sensitive. **We have rephrased this sentence to: 'These discrepancies from simple interaction suggest that complex biogeochemical models need to be tested by altering their default functional forms…' in the revised manuscript (page 3 line 21-23)**

5) *p3 line 3: "happened"*

AR: We can't find happened in p3 line 3 – **if this is in line 23, we have rephrase this sentence as mentioned by reviewer #2**

6) *p6 line 25: "Oschlies and Garcon, 1999" - a follow-up study by Oschlies and Schartau (2005, Jour. Mar. Res., 63, 335–358) highlighted this even more; see also the study by Friedrichs et al. mentioned above.*

AR: Thank you for the suggestions, **we have added these literatures accordingly in the revised manuscript (page 7 line 11).**

7) *p7, section 2.5.1: Physical input: please indicate the vertical grid on which this model was run, including its maximum depth.*

AR: This has been stated in the biogeochemical input, for clarity **this have been revised in the annotated manuscript, in page 8 line 8-11.**

8) *p7 section 2.5.2: Biogeochemical input and validation data: I would suggest to list all the details of the different stations (location, max depth, data source, data assimilated) in a table.*

AR: Thank you for the suggestion, **we have included this in the new manuscript (please see table 2 in the annotated manuscript)** however we do not assimilate any data into our model

9) *p7 section 2.5.2: Do I understand correctly, that the observations were used for initialisation as well as for model validation? If so, then the model is not validated against fully independent data (at least not at depth, given a short simulation time of just 10 years), and I would suggest to mention it here.*

AR: Indeed, we are using the observation to initialise the model (using in situ chlorophyll, nitrogen, iron, and silicate data from January 1998), but we do not use the later in situ data to force the model, so the validation data is independent.

10) *P8, line 13: "Simulations are made at 37 depth levels" - This formulation sounds as if simulations were done separately for each depth level.*

AR: This have been rephrased **to 'the model is simulated at 37 depth levels…' in the revised manuscript (page 9 line 5)**

11) *p15 line 24: "Most current biogeochemical models are run in a deterministic, rather than a probabilistic, manner, even though data from observations contain many uncertainties, eg. in satellite-derived chlorophyll." - I think I can guess what you want to say, but in the current form this sentence is not clear.*

AR: **This have been removed from the manuscript**

Literatures cited:

[revised manuscript text omitted]

---

## Author Response (AR2)

**Dear Editor,**

We would like to thank the reviewers for agreeing to review this paper again, and for their constructive criticisms. We would also like to thank Reviewer #1 for accepting the revised manuscript in its current form.

Reviewers 2 and 3 have raised their concerns about the function fitting. We have now added more details on the fitting in section 2.1, 2.2, and 2.3 (page 5-6) to ensure that the fitting process, which is crucial to this study, is clear. We have further done another fitting using narrower concentration ranges; between 0.001 to 20 mmol m-3, for dissolved inorganic nitrogen (DIN) and 0.001 to 10 mmol m-3, for phytoplankton and zooplankton concentrations, and confirmed that there are no significant changes in parameters when the range of DIN and phytoplankton concentrations have been changed (please see Table 1 and Reviewer #2's comment for p4 l29). We have also discussed (see annotated manuscript page 16 line 10-11 and 26-27) how the fitting still maintains the phenomenological similarity (such as when fitting nutrient uptake with DIN in the lower concentration range (up to 5 mmol m-3), we can still see that at lower concentrations the function deviates).

Below is our point by point response to all comments made by two reviewers. The reviewer comments are included in italic and responses are in bold. Please note that all the line numbers and pages in this response refer to the annotated version attached.

We hope that the response would be satisfactory, and we look forward to your decision.

**Kind Regards,**

**Authors.**

**Response to the reviewers:**

*Reviewer #2*

*The updated manuscript shows some improvement in places and the authors have added some useful analyses. However, some of my main criticisms of the manuscript still apply.*

*General comments*

1) *In my first review I wrote that "The results of the default run can have knock-on effects on the ensemble: in multiple parts of the manuscript the authors note that when there is a large bias between the model (ensemble) and the observation, that the ensemble spread is too low when really other model aspects may be to blame for the bias. In other words, problems with the parametrization, the physical model, or the 1D nature of the model cannot be explained by structural uncertainty in the biogeochemical model." To make it more explicit: if the ensemble does not envelop the observations, this does not necessarily imply that the ensemble spread is too low. A bad model (physical model, unresolved BGC*

*processes) with a large bias may be to blame. This is now touched on in the discussion section but ignored before, and phrases like "the observation is outside the ensemble range ... making the ensemble spread too narrow" appear too often, especially in the phytoplankton phenology section 3.4 (blooms are often controlled by physical processes).*

**AR: We agree that the large bias, especially in the physical model, may be to blame for the ensemble not covering the in situ observations. We have updated the text to emphasise this (page 16 line 35 to page 17 line 2). Also, we have replaced 'ensemble spread too wide/narrow' with appropriate text to convey that the observations are outside the ensemble range.**

2) *With regard to my earlier comment (reviewer 2, general comment (2)): The plot in the response shows little difference between the regular and the log-transformed fit. This is surprising to me. The linear fit produces a somewhat good match to the sigmoidal function only for high phytoplankton concentrations. This can be easily explained as the linear scale emphasizes high phytoplankton values. For the log-scale fit, what I suggested was to fit the linear function across the range of phytoplankton concentrations shown in the figure using the same log-scaling. I would expect a much lower mortality value in this case, much closer to 0.05 than the 0.085 that were obtained. This may yield significantly different results. The best idea would probably be to let the data determine the fit, i.e. minimize the misfit for the values occurring in the model output (or the observations if these are available). In any case, since many of the results hinge on which functional form is used, the authors need to include more detail on how exactly the fit was obtained. This should include the range of phytoplankton concentrations considered in the fit.*

**AR: Perhaps the closer value to the linear fit occurred because we previously used a slightly higher range of phytoplankton between 0.01 – 50 mmol N m-3. We have now done the fitting on linear scale using phytoplankton concentration between 0.001 – 10 mmol N m-3, to capture a more realistic range based on the model outputs. The fitted parameters within this refined range remained very similar to our initial linear fit using 0.001- 100 mmol N m-3, ensuring robustness of the linear fit in the case. This have also been observed in nutrient uptake and grazing. However, if we log transform phytoplankton concentration before the fit (which we avoided earlier to ensure that we use the definition of functional forms, which are defined for actual concentrations, and not for log-transformed concentrations), the maximum mortality value for 'linear functional form' changes to 0.04. But, even in this case, the mean absolute errors (measures of goodness of fit) between the hyperbolic (default) function and other functions show lower misfits for all the linear fittings compared with the log-transformed fittings, as summarised on Table 1. Therefore, we have decided to stick to linear fitting in this study. But following the suggestion of the referee, we have added further details on fitting in the method section ( section 2.1 page 5, line 1-4, section 2.2 page 5, line 16-19, section 2.3, page 6, line 1-2 and 5-7), and in the discussion section (page 16, line 17-23 on using log transform parameter (0.04) for the linear mortality functional form).**

*Table 1. Comparison between log transform and linear-scale function fitting parameter values, with phytoplankton concentration ranging from 0.001-10 mmol N m-3, and its mean absolute errors. The numbers in brackets are the original fit (concentration ranging from 0.001-50 mmol N m-3).*

| functional form | log transform parameter | mean abs error | non log transform parameter | mean abs error |
|---|---|---|---|---|
| sigmoidal | k = 1.20 (1.10) | 0.007 (0.002) | k = 0.74 (0.74) | 0.007 (0.002) |
| linear | μ=0.04 (0.085) | 0.05 (0.13) | μ= 0.094 (0.097) | 0.009 (0.008) |
| quadratic | μ= 0.04 (0.02) | 0.01 (0.003) | μ= 0.05 (0.05) | 0.01 (0.003) |

[Figure]

*Figure 1. Mortality functional forms optimised against hyperbolic function. The range of phytoplankton used was 0.001 to 10 mmol N m-3*

*At the risk of sounding pedantic: there are still many sentences in the manuscript that are not clearly formulated and that can be interpreted in different ways. Readers familiar with the topic will likely know what is meant but others will not. I have pointed out some of these instances below but I would recommend that the authors go through the manuscript again carefully.*

**AR: Thank you for this suggestion. We have included all the suggestions, and have also gone through the manuscript for further textual improvements.**

*Specific comments:*

p1

*l3: "We assessed the impact of structural sensitivity ... by modelling the chlorophyll and nitrogen concentrations at five different oceanographic stations spanning three different regimes": This appears to imply (incorrectly) that structural sensitivity is linked to the model location.*

**AR: Thank you for noticing this, we did not mean that structural sensitivity is linked to the model location, but we are testing to observe the effect on structural sensitivity at different oceanographic regions. We have rephrased this in the revised manuscript to '….. by modelling the chlorophyll and DIN concentrations. The model is run at five different oceanographic stations**

**spanning three different regimes, over a 10-year timescale to observe the effect in different regions' (page 1, line 4-6).**

*l3: "nitrogen": Do you mean "nitrate"? This comment applies to later mentions of nitrogen as well.*

**AR: We use the combination of inorganic nitrate, nitrite, and ammonia as initial condition for dissolved inorganic nitrogen. We have rephrased nitrogen into dissolved inorganic nitrogen (DIN).**

*l6: Agreed, though there are other applications for BGC models than climate change assessments.*

**AR: We have added assessing the impact of anthropogenic input on biogeochemical cycles in the marine ecosystem, and producing decadal reanalyses (page 2, line 9-10)**

*p2*

*l24: Mention some causes of mortality here.*
**AR: These have been added to the revised manuscript. We added '…, due to diseases or implicit higher trophic levels…' (page 2, line 29).**

*p3*

*l14 "demonstrated ... that linear density-dependent mortality produces the biggest difference in diatoms". As opposed to other phytoplankton types? This is not clear.*
**AR: The most notable difference has been observed in diatoms. We have rephrased this sentence in the manuscript to make it clearer (page 3, line 21).**

*p4*

*l29: "The difference in shape of the optimised functional forms are more obvious at low nutrient concentrations." Likely due to the way the fitting is performed, see general comment (1).*

**AR: See our response to fitting above. We have changed the range and still observed that at lower DIN concentrations (below 1 mmol m-3) the functions still deviate the most. We have however changed the range of DIN concentration (between 0.001 – 20 mmol m-3) we used to fit the function. This reduction in range also has not changed the shape-defining parameters for other functional forms (U2= 0.74, U3= 1.12, U4= 0.60). We have added some text to mention these: ' The fit is done within the nutrient concentrations of 0.001 – 20 mmol N m-3 and are discretised into 1000 intervals. The difference in shape of the optimised functional forms are more obvious between 0.1 to 1 mmol m-3' (page 5, line 2-3).**

*p5*

*l3: "graze on small phytoplankton, non-diatoms, and detritus": This is confusing, as small phytoplankton and non-diatoms are the same variable.*

**AR: This has been rephrased in the manuscript. We have removed small phytoplankton and keeping non-diatoms (page 5, line 7).**

*l13: "hyperbolic": at this point it is not clear what the hyperbolic refers to, please use "hyperbolic" and "sigmoidal" when G_1 and G_2 are introduced initially, in line 6.*
**AR:  This has been rephrased in the manuscript.**

*l28: Here it is very important to note if the area below the function is computed in log-space or not and over what range of concentrations. Figure 1 makes it look like a log transformation was used and concentrations between 0.001 and 10 mmol N m^-3 were considered in the fitting.*

**AR: The area below the function is not computed in log space, and the range of concentrations were between 0.001 and 100 mmol N m-3. In the new fit, (see the general response 2), we have used 0.001 and 10 mmol N m-3 instead. We have added text to detail the fitting (page 6, line 1-2 'calculated as the area below the function …. *In linear terms between 0.001 to 10 mmol m-3*' and line 5-7 for the detail about concentration range).**

*p6*

*l7: "as" -> "is"*

**AR: This has been changed in the revised manuscript (page 6, line 14)**

*l30: "mortalities" -> "mortality formulations"*

**AR: This has been changed in the revised manuscript (page 7, line 6)**

*p7*

*l18: There are several 1D models now, do they all use the same depth levels (at every station)?*

**AR: We only use one 1D biogeochemical model, but we use several NEMO outputs for the physical input. Our 1D biogeochemical model has the same level thickness resolution for all the stations, however shallower stations have fewer levels, such as Cariaco and L4.**

*l21: The station abbreviations should be mentioned earlier when the stations are first mentioned (section 2.5).*

**AR: This has been changed in the revised manuscript (please see page 6, line 30-33).**

*p8*

*l3: I don't quite understand the reasoning behind using the same concentration at each depth level. Are the initial conditions quickly forgotten? Mention this. A short model spinup could have been useful.*

**AR: Since station L4 is only 50m deep, we assume that the concentration is similar at all depths during the winter start of the run (1st January 1998). We have mentioned that L4 is a shallow station and therefore using similar biogeochemical input at all depth levels. We have now added 'Since the maximum depth in this station is only 50 m deep….' (page 8, line 10).**

*l17: "for to" something is missing here.*

**AR: We have corrected this mistake. What we meant was for the seasonal variations.**

*p10*

*l9: How does the ensemble range not cover the whole ensemble and what exactly is meant by "if we only allow one process function at a time to change"?*

**AR: Allowing one process function at a time means that we only change functional form in one process, whilst keeping other processes the default function, therefore the new smaller ensemble (11 members) range is different from that of the old larger ensemble (128 members). We have rephrased this to be clearer (page 10, line 18-19)**

*p13*

*l23: "phenology metrics" -> "observed phenology metrics" or "phenology metrics obtained from observations"*

**AR: Thank you, this has been changed in the revised manuscript (page 14, line 1).**

*l27: How can a range be late; rephrase?*

**AR: We meant that the dates within the inter-quartile range are later than for in situ observations. This has been revised in the revised manuscript (page 14, line 6).**

*l28 and again l30: "For initiation...": add "bloom"*

**AR: This has been added in the revised manuscript (page 14, line 7).**

*p17*

*l23: This long sentence mentions "active prey selection" twice.*

**AR: The first active prey selection has been removed from the sentence.**

*Fig 1: Should it be "nitrate" in the axis label in (a)? Same comment applies to other figures.*

**AR: We have changed this to DIN**

*Fig 3: missing units in (b) and (e)*

**AR: We have added the units in (b) and (e)**

*Fig 7: "(d)" is used twice*

**AR: We have changed (d) to (e)**

*Fig 9: The order in which panels are labeled is confusing and inconsistent. It lead to a problem in the description as well. White line is in panel (a) not (b).*

**AR: We have changed the labelling, please also see figure 3 and 14.**

*Fig 10: Use same y-axis across all panels.*

**AR: We have changed this**

*Fig 11: "(e)-(f)" should be "(e)-(h)"*

**AR: We have changed this**

*Reviewer #3*

*As already noted in my first review, I really appreciate the very thorough and exhaustive attempt to investigate the sensitivity of a biogeochemical model that will be, at a later stage, used for projections of climate change and other applied tasks. This paper could provide important information on global biogeochemical model sensitivity with respect to structural uncertainty, even for users not applying this particular model. However, so far, and for this purpose, I find the discussion of results somehow incomplete. I think discussing the outcome of these experiments before the background of earlier studies that also deal with global model uncertainty and performance would enhance the paper's impact, and put this work into a broader context.*

1) *For example, I am not convinced that the results indicate a larger importance of structural sensitivity compared to uncertainties related to physical or parametric uncertainty. To my impression, so far there is little (with respect to the metrics applied) evidence that mean or median perform much better than the default model (see below, point B). I think this should*

*be discussed more, e.g. before the background of the study by Kwiatkowski et al. (2014), who showed that "No model is shown to consistently outperform all other models across all metrics." when comparing six models (among them MEDUSA) against surface tracers such as DIN and Chl. Likewise, Galbraith et al. (2016) or Kriest (2017) found that models of varying complexity performed quite similar with respect to metrics such as RMSE etc. The importance of parametric uncertainty was addressed in several studies, even at a global scale. There are also many studies that deal with the uncertainty due to physics (starting with the "classic" study by Najjar et al., 2007), and show a large impact of circulation or phenomena at smaller (meso) scales.*

**AR: Thank you for acknowledging the importance of our study. We have revised in parts to highlight this importance. However, we do not think that structural sensitivity has larger importance compared to physical input or parametric uncertainty. We are trying to emphasise that using similarly shaped functional forms may yield different plankton dynamics, and such particular study have not been done as much as for parametric uncertainty, or physical uncertainty. We have also noted in the discussion section that 'a coupled biogeochemical model is only as good as its physical model', meaning that this input may have larger importance than structural sensitivity (page 16 line 33 to page 17 line 2) and how unresolved biogeochemical processes also affect the model output especially at the coastal and oligotrophic regions (page 17, line 26-29 and page 18, line 23-27). There are also no metrics to determine whether one sensitivity is more important than the others. We are trying to generate a range of model results from perturbing the model equations alone, with fixed parameters (apart from the shape defining parameters, such as half saturation coefficient) and quantify them.**
**Further, it should be noted from the results table (Table 3 in the manuscript), that only at station BATS where there is a large bias, the RMSE for default run is better than the ensemble mean, and we have stated this in the result section. But in all other stations (ALOHA (ensemble mean RMSE 0.095 < default 0.097 for chlorophyll profile), L4, PAP, Cariaco), the ensemble mean and median have lower RMSEs than the default run. We have now emphasised this important result within the discussion section (see page 17, line 11-21).**

2) *Secondly, different equations, as applied in this study, necessarily result in different nutrient or food affinities of plankton, as briefly discussed by the authors; therefore, to some extent the structural sensitivity also includes some parametric sensitivity. Again, this might also be worthwhile discussing (see below, point A, and specific comments for p15).*
   *I would thus recommend an overhaul of the discussion section, as already suggested by Reviewer #1, to give this paper a wider impact and "to address the big picture".*

   a. *Authors response to my comment 2 on fitting procedure of different structural forms. "However, we are trying to capture the whole range of nutrient and phytoplankton at all the different region, [...]". As shown in Figures 3, 9 and 14, the full range of observed and simulated nutrient concentration is between ~0-10 mmol/m3, and the range of Chl ~0-1 (PAP, Cariaco), 0-0.3 (ALOHA and BATS) and ~0-6 (L4), i.e. more than an order of magnitude lower than used for curve fitting. I would suggest to give the range over which the functions were fitted in the paper (method section).*

      **AR: Thank you for the suggestion, we have updated the discussion in places to**

**include these points. Also, we have tried fitting the function between 0.001-10 mmol m^-3 and we found that there are no significant changes in the parameter values if we use linear fitting (please see the response for Reviewer #2 general comments 2, table 1). We have now included further details on fitting (method section 2.1 page 5, line 1-4, section 2.2 page 5, line 16-19, section 2.3, page 6, line 1-2 and 5-7). The information about the range has been included in the manuscript. We also agree that structural sensitivity is also dependent on parametric sensitivity, especially in linear mortality (please see the response to Reviewer #2 general comments 2), and we have included this in the manuscript (please see discussion section, page 16, line 17-22).**

*Further, in their response the authors state that*

*"Suppose we are optimising the nutrient uptake on the similar range of station BATS and ALOHA (with maximum nitrogen and phytoplankton concentration of 5 mmol N m- 3, shown on Figure 3, although at stations like Cariaco, PAP, and L4, we may see nitrogen larger than 5 mmol N m-3), the functions still deviate at low nitrogen and phytoplankton concentration."*

*This is very important information, and I would suggest to mention this in the paper, together with a full description on how the fitting was carried out. (Was the range between 0-100 discretized, and if so, into how many bins? Were the discrete values equally distributed over the range?)*

*I still think this issue (fitting different functional forms, the side effects on affinities and rates in different oceanic regimes) might merit more in depth discussion and detailed presentation.*

**AR: We have mentioned this in the method section in the manuscript, and we have now noted clearly that we are fitting in the range between 0.001-20 mmol m-3 for DIN (nutrient uptake) and 0.001-10 mmol m-3 (zooplankton grazing and plankton mortality) for phytoplankton and zooplankton, with 1000 intervals, similar to the observations. We found that there isn't much change in the shape defining parameter values (please also see the response for Reviewer #2 general comment no 2 and for p4 l29).**

b. *I find the analysis of model performance for the different structural types (ensemble mean or median vs default) somewhat biased and incomplete: For example, Table 3 shows, that out of the 51 criteria listed in that table for r, RMSE and Bias for DIN and Chl (profile, surface, integrated), the default model is the best model w.r.t. to the bias and also r, compared to ensemble mean or median. It also outperforms mean and median at BATS, and, all three criteria combined, is as good as the ensemble mean at ALOHA. Also, for RMSE some of the values do not seem to provide a clear evidence that the ensemble mean or median is better: for example, at ALOHA the RMSE for DIN and Chl (profile or surface) is very similar among the different models.*

**AR: We have stated in the results section that all of the statistical metrics at BATS are better in the default run. But in all other stations the results are promising. For example, in ALOHA, for the surface cholorophyll, the RMSE of ensemble mean (0.05) is better than the default run (0.07), and chlorophyll profile RMSE in ensemble mean is also lower (0.97) than the default run (0.10). In all other stations, chlorophyll profiles' RMSEs are lower than default run: Cariaco, ens mean= 0.83 < default= 0.87, PAP ens mean =0.06 < default= 0.18, and L4, ens mean = 0.42 < default= 0.83. In terms of correlation, in nitrogen, the ensemble mean in stations that are not BATS show either better or similar correlation to the default run: ALOHA, ens mean= 0.77 = default, Cariaco, ens mean= 0.78 > default= 0.76, PAP ens mean =0.23 > default= 0.21, and L4, ens mean = 0.70 > default= 0.52. Nonetheless, we agree that the model bias didn't improve, and we have explained the additional cause of model bias (see page 16, line 33- page 17, line 2 for physical input bias and page 17, line 24-29 and page 18, line 23-27 for unresolved biogeochemical process), and further discussed why the default run produces lower bias these in page 17, line 13-17).**

*Specific comments:*

*p1, line 6-7: "that describe the key biogeochemical processes" - I would suggest to rephrase this to "that describe some key biogeochemical processes", as, at a larger scale, phytoplankton light affinity, recycling and sinking organic matter may also be regarded as key processes.*

**AR: This has been changed accordingly in the revised manuscript (page 1, line7)**

*p1, line 13: "Changing mortality" - which mortality: both zooplankton and phytoplankton?*

**AR: Yes both zooplankton and phytoplankton, and this has been included in the abstract (page 1, line 14)**

*p1, line 14-15: "The RMSEs between in situ observations and the ensemble mean and median are mostly reduced compared to the default model output." - See above, point B. Why not address the bias or correlation coefficient here, and in the discussion?*

**AR: We have added the statement and further explanation in the discussion section (page 17, line 13-19).**

*p2, Introduction: This is probably just a minor point, but usually global models are NPZD models, with detritus (and/or DOP) being an important component in the recycling and vertical redistribution of nutrients.*

**AR: NPZD is now referred to (page 2, line 7).**

*p2, line 17ff: "Moreover, in order to investigate the effect of global climate change and anthropogenic activities in the ocean, marine biogeochemical models are now being embedded into earth system models." - Simple marine biogeochemical models were first embedded into GCMs in the*

*90ies, and into ESMs possibly around 2005 (e.g., UVic, Schmittner et al., 2005). Perhaps it would be better to rephrase this as "[...] activities in the ocean, MORE COMPLEX marine biogeochemical models are now being embedded into earth system models."*

**AR: This has been changed accordingly, thank you for the suggestion (page 2, line 21)**

*p3, line 14: "A few studies have investigated the effects of biogeochemical process formulations, e.g. Yool et al. (2011) has demonstrated [...]" – Perhaps better "A few studies have investigated the effects of biogeochemical process formulations. For example, Yool et al. (2011) have demonstrated [...]" ?*

**AR: This has been changed accordingly, thank you for the suggestion (page 3, line 19)**

*p4, section 2.1: I suggest to give range of nutrient concentrations over which the functions were fitted, and some details of the fitting procedure here (see A).*

**AR: We have added the range of nutrients in this section (0.001 – 20 mmol m-3).**

*p5, line 5 "which are high quality food sources" - In the model? In reality? Is this of importance here, and if so, how is it embedded in the model?*

**AR: meaning that it is more nutritious compared to detritus and non-diatoms. This has been rephrased (page 5, line 9-10)**

*p5, line 12: "These functions both become constant at a maximum grazing rate." - Perhaps better: "These functions approach a maximum grazing rate at high concentrations of prey."*

**AR: This has been rephrased accordingly, thank you for the suggestion (page 5, line 16-17).**

*p5, line 25: "as shown on Fig. 1(b)." - Fig 1(c)?*

**AR: This has been changed, thank you for noticing (page 5, line 20).**

*p5, line 27ff "integrated over the range of prey density" - Give range here (see p4, section 2.1)*

**AR: We have stated the range of phytoplankton concentration, and changed prey density into phytoplankton concentrations (page 6, line 1-2).**

*P6, line 5: "From a previous 3-D MEDUSA run, in the oligotrophic regions show" – skip "in".*

**AR: We have removed this from the sentence. Thank you for the suggestion (page 6, line 12).**

*p6, line 9: "MEDUSA also contains both slow and fast detritus sinking factors." - What does this mean? Sinking speeds? Remineralisation length scales? Perhaps better "MEDUSA also parameterises slow and fast sinking detritus"?*

**AR: Thank you for the suggestion, we have rephrased this sentence accordingly (page 6, line 16).**

*p6, line 14f: "We chose a lower sinking rate of 0.1 m day to prevent depletion of state variables particularly at the shallower stations." - Does this mean sinking organic matter is buried at the sea floor? If so, I would mention it here.*

**AR: Yes, the remineralisation rate is slower than the sinking rate, and therefore loss occurs to the seafloor. We have mentioned this in the revised manuscript (page 6, line 19).**

*p7, line 18f: "Our 1D model uses these same 63 depth levels vertical resolution." - The same as the global (MEDUSA) model? Then, on p8, line 9: "The model is simulated at 37 depth levels, [...]" - I am a bit confused - How many levels did the 1D setup have - 63 or 37?*

**AR: The model uses 37 depth levels, but the resolution is similar to the 63 depth levels vertical resolution, instead of the 75 mentioned earlier (please see page 7, line 22-26)**

*p7, line 18f: "we use the integrated nitrogen over 200 m (integrated nitrogen / depth)" - is this meant to be "dissolved inorganic nitrogen averaged over the upper 200 m"? I would suggest to refer to "averaged" when appropriate (also in some of the figure captions); Additionally, if nutrients are mean it should be "nitrate" or DIN or "dissolved inorganic nitrogen", because "nitrogen" alone can also include the organic forms.*

**AR: Thank you for the suggestion. What we meant was averaged over the upper 200m, and dissolved inorganic nitrogen and this has been changed accordingly.**

*p 9, line 25: "Chlorophyll and nitrogen profile 10 year means"? What does this mean? "Observed mean profiles of chlorophyll and DIN"?*

**AR: Yes. We have rephrased this sentence, thank you for the suggestion (page 10, line 1).**

*p15, lines 15ff: "In order to maintain phenomenological similarity, these functions are calibrated using non-linear least squares, while keeping the maximum process rates fixed." - To me it seems as if the phenomenological similarity is maintained with respect to the rates integrated over the nutrient and Chlorophyll range considered; i.e. between 0-100 mmol/m3 or mg/m3. This could downweigh the affinity at very low nutrient or chlorophyll concentrations. (See above, A.) Thus, phenomenological similarity is maintained with regard to a certain criterion. As stated by the authors in their reply, fitting the curves over a narrower range results [0-5] in almost the same curves, which would strengthen the above statement of phenomenological similarity. I would suggest to add a few sentences about this here.*

**AR: Thank you, we have included a few sentences about this in the discussion section and methods (methods page 5, line 2-3 and line 18-19, and in discussion page 16, line 10-11 and line 26-27).**

*p15, lines 19ff: "Applying structural sensitivity in the 1-D framework has also allowed a large parameter space of concurrent variations to be explored for several different oceanographic regions, and with minimal computational cost." - How is the parameter space explored, particularly when considering the above statement about phenomenological similarity?*

**AR: What we meant was large range of process variability, we have rephrased this in the revised manuscript (page 16, line 2-3).**

*p15, line 24: "have" -> has*

**AR: We have changed this in the revised manuscript**

*p15, line 25: "This is expected as at low concentrations, using the G2 function would graze more phytoplankton, as shown on Fig. 1(b)." - Again, here the effect of the fitting procedure for different functions shows up: perhaps adjusting the k's of the different function such that they become similar at very low concentrations might have resulted in similar results at the oligotrophic stations BATS and ALOHA? Thus, the effect of different functional forms seems to be mingled with the effects of parametric uncertainty. Of course, this cannot be avoided, but I think I think it could be discussed more.*

**AR: We have also tried fitting it using a lower range of phytoplankton concentrations, but it seems that the functions still deviate greatly in the lower concentration range (please see page 16 line 10-11). Also see our response to Referee #2, page 2.**

*p16, lines 19ff: "At most of the stations, the ensemble mean produced lower RMSE compared to the default run, suggesting that the structural ensemble with a wide range of predictions covering the in situ observations, is likely to produce a mean field closer to the observation, than a single-structure model." - I am still not fully convinced about this - see my above comment B.*

**AR: We have included the discussion on model bias and correlation in the discussion section (page 17, line 13-19). However, higher correlations in ensemble mean and median have been observed in nitrogen correlation in all of the stations but BATS (please see table 3 in the manuscript).**

*p18, line 8: "widely used in the community" - In which community?*

**AR: We have removed this sentence in the manuscript**

[revised manuscript text omitted]

---

## Author Response (AR3)

Dear Editor,

We would like to thank Reviewer #2 for agreeing to review this paper again, and for the constructive criticism.

As suggested by Reviewer #2, we have now changed the functions from $U_1$, $U_2$, $U_3$, and $U_4$ into $U_h$, $U_s$, $U_e$, and $U_t$ for uptake, and similarly to the mortality functions. We have also re-structured section 3, whereby the observations are described first, followed by the ensemble simulations, their statistics, and NRR. The paragraphs and sentences in the Summary and Discussion section have been made shorter.

Below is our point by point response to all comments made by Reviewer #2. Please note that the line and page numbers corresponds to the annotated version attached. Reviewer comments (RC) are included in normal text and responses (AR) are in bold.

We hope that the response would be satisfactory, and we look forward to your decision.

Kind Regards,

Authors.

**Response to Reviewer #2:**

RC: The updated manuscript has improved in many aspects. I have no more issues with methods and results that are presented but the way they are presented needs improvement before the manuscript can be published.

**AR: Thank you reviewing the paper again and acknowledging the improvement of the current manuscript.**

General comments:

RC: I found the results sections were difficult to read and not well structured. For example, section 3.1 starts out by describing the data. The second paragraph covers model DIN seasonality but in the last sentence also starts the model-data comparison. In the third paragraph, it is initially not clear if the chlorophyll data is described or the ensemble. The fourth paragraph then launches the model data comparison and mentions the same result from the second paragraph again (NRR=1.25 for DIN). This back and forth is confusing to the reader. I suggest to reorganize each results section: start out by describing the ensemble characteristics (which are the focus here) and then compare them to the data.

**AR: Thank you for the suggestion. The data description at the beginning of section 3.1 is included because the abyssal station has a shorter data timespan compared to other stations. We find that it is easier to start with in situ observations then briefly explain the match or miss-match between the ensemble and the in situ. We therefore have re-structured the paragraphs in the results section by first describing the observations, followed by the ensemble simulation. Then we describe ensemble characteristics, along with the RMSEs, NRR, and other error statistics. These are**

**followed by the ensemble results when we only change one functional term in each member, keeping the others in the default forms. (Please see the revised section 3.1 (page 9-11), 3.2 (page 11-14), 3.3 (page 14-16), and the Phytoplankton Phenology in section 3.4 (page 17-19).**

RC: Section 4 is very dense and difficult to read. Again there is a lack of structure and the first few paragraphs each cover a range of different aspects without much consistency. Some run-on sentences are almost impossible to follow (for example, p18, l15). The manuscript would greatly benefit from tidying up this summary section and a clear focus on the key results.

**AR: Thank you for the suggestion, we have now made the sentences and paragraphs shorter, for example, for p18, l15, we have split it into two sentences, and make the sentences a bit shorter (page 24, line 17-22).**

RC: In order to improve readability, I'd suggest to rename the functions that are used throughout the manuscript, so it is easier for the reader to keep track. For example, new names for the uptake functions could be:

U_1: U_hyp

U_2: U_sig

U_3: U_exp

U_4: U_tri

The same naming scheme can easily be applied to the other functions.

**AR: We have explained which functional forms corresponds to the abbreviations (U_1, U_2, G_1, etc) in Table 1. We have now changed the function abbreviations to U_h, U_s, U_e etc. For grazing function, we keep Holling type II function as G_2, and the default Holling type III as G_1.**

RC: Specific comments:

p2

l6: "NPZ-D": "NPZD" is much more commonly used.

**AR: We have changed this to NPZD (page 2 line 6).**

l31: Change "parameters" to "biological parameters".

**AR: We have changed this to biological parameters (page 2 line 31).**

p3

l23: Anderson et al. (2015) is cited twice in the same sentence.

AR: **The second Anderson et al. (2015) has been removed (page 3 line 26).**

p4

l10: Here the manuscript shifts briefly from present into past tense. Use on consistently.

**AR: We have changed 'assessed' to assess (page 4 line 11).**

p6

l11: "has" -> "have"

**AR: This has been changed (page 6 line 11).**

l18: "study" -> "studies"

**AR: This has been changed (page 6 line 18).**

l18: 0.1 is not lower than zero, rephrase to something like "towards the lower end of the range of literature values"

**AR: This sentence has been rephrased (page 6 line 19).**

l26: "the 3-D models' predictive skill" -> "the predictive skill of 3-D models"

**AR: This has been changed (page 6 like 27).**

l33: "ensemble model" -> "model ensemble"

**AR: This has been changed (page 7 line 5).**

p7

l2: "This provides a total number of 128 ..." This sentence should probably precede the previous one. Altogether, the ensemble generation is needs a bit more explanation.

**AR: This has been changed, we have added more explanation before this sentence (page 7 line 4-6).**

l9: What is "in situ satellite SST"?

**AR: We have rephrased this to satellite derived sea surface temperature (page 7 line 13).**

l15: "there is no" rephrase to "is zero"

**AR: This has been changed (page 7 line 20).**

p8

l12: This paragraph is very difficult to understand. How does a simulation from 6-1200m minimize the computational cost? Is the maximum depth of the model at Cariaco also at 500m? What does "The level thickness is similar to that in 63 depth levels." mean? Do the 2 stations with reduced depth have fewer layers? You do not need to answer these questions in your reply, please rephrase the paragraph so that the reader knows.

**AR: Running the model from 6-1200 instead of 6-5800 minimise the computational cost, and this have been added in the paragraph. We have also added the maximum sampling depth for Cariaco changed this sentence to make it clearer and shows all the boundaries for the depth levels (please see page 8 line 16 – 22).**

l19: "We use ..." The sentence does not say what the metrics are used for.

**AR: We have added what the metrics are used for (page 8 line 26-27).**

p9

l9: I don't this paragraph is very helpful. Instead of explaining in what order the metrics will be used (the order is not followed anyway: in 3.1, NRR is mentioned before RMSE), it could explain the order the following subsections will be in.

**AR: We have changed the content of this paragraph and included the order the following subsection will be in (page 9 line 15-22).**

p10

l11: "range": Is this the ensemble range? Make this explicit.

**AR: Yes this means ensemble range, and we have changed this to ensemble range (page 11 line 9)**

l11: "If we only allow one process function at a time to change": I still don't think the new explanation helps much to understand what is going on. The "at a time" seems to imply that something is changing over time. I would suggest: "by generating an ensemble in which the functional form of only one of the processes is modified"

**AR: We have changed the explanation to: 'However if we only allow one process function to change in each ensemble member, keeping the other processes with their default functions' (page 11 line 9-12).**

l13: "covering 84% of the all ensemble members": what does this mean, please explain better.

**AR: 'all ensemble members' has been changed to 'the full ensemble (128 members)' (page 11 line 12).**

p11

l5: "A DCM occurs when lower chlorophyll is detected at the surface": Not sure if a description of DCM is needed but this one is more confusing than helpful.

**AR: We have rephrase this to 'Another feature of the oligotrophic ocean is a deep chlorophyll maximum (DCM) that occurs below the mixed layer, when the chlorophyll concentration in the surface is low {Fennel2003, Letelier2004}' (page 11 line 33 – page 12 line 1).**

p13

l3: "the in situ" add "data"

**AR: This has been removed from the manuscript.**

l30: "At most stations, the observed phenology metrics are covered by the ensemble range.": "covered by" does not make it clear what you are trying to say, I would suggest to rephrase using "fall within".

**AR: This sentence has been removed from the manuscript.**

p15

l26: "Through this approach, we provide a perturbed biology ensemble conditioned upon process structural uncertainties." This is a very dense sentence

**AR: We have changed this sentence to 'Through this approach, we provide a perturbed biology ensemble conditioned upon structural uncertainties in model formulation' to the sentence (page 20 line 9-10).**

p16

l2: "Even though fitting..." This sentence is difficult to understand, please rephrase.

[revised manuscript text omitted]